# Structured Semantics Meet Uncertain Visuals: A Unified Approach to Calibrated Test-Time Prompt Tuning

## Abstract

Large vision-language models (VLMs) generalize well zero-shot but become overconfident and poorly calibrated under distribution shifts. Existing test-time prompt tuning (TPT) methods largely apply uniform entropy minimization with fixed geometric regularizers, ignoring instance-wise uncertainty and domain-specific visual cues. We propose **Uncertainty-Calibrated Test-Time Prompt Tuning (UC-TPT)**, a label-free TPT framework targeted at improving reliability rather than solely maximizing accuracy. UC-TPT consists of three theoretically motivated components: (i) lightweight visual-to-text conditioning that injects *shallow* visual statistics—where shift is most pronounced—into prompts, yielding domain-conditioned predictions; (ii) an *uncertainty-tempered* entropy objective that adaptively controls distribution sharpness to curb overconfidence; and (iii) a topology-aware prompt regularizer that approximately preserves the pairwise semantic relations of manual prompts, stabilizing adaptation in the pretrained embedding space. Experiments on CLIP, Remote-CLIP, and BiomedCLIP across diverse benchmarks demonstrate that UC-TPT consistently outperforms existing TPT methods in calibration robustness, yielding significant reductions in Expected Calibration Error (ECE) across a wide range of distribution shifts at a marginal cost to classification accuracy. [1]

## 1 Introduction

Despite the strong generalization of large-scale Vision–Language Models (VLMs) Radford et al. (2021a); Jia et al. (2021); Yang et al. (2022), their reliability can degrade sharply under real-world distribution shifts (e.g., imaging conditions, sensors, environments), limiting deployment in safety-critical settings such as autonomous driving and medical diagnosis. *Test-time adaptation (TTA)* Liang et al. (2025); Xiao & Snoek (2024) mitigates this by adapting at inference using only unlabeled target data, avoiding retraining and supervision. For VLMs, *Test-Time Prompt Tuning (TPT)* Shu et al. (2022) is especially attractive, as it updates only a small set of prompt tokens while keeping pretrained encoders frozen.

However, existing TPT pipelines expose a key research gap: they primarily adapt *confidence geometry* without explicitly controlling *when* and *where* updates should occur. Early TPT Shu et al. (2022) relies on entropy minimization, which encourages peaky posteriors and can amplify overconfidence on ambiguous or out-of-distribution (OOD) inputs, harming *calibration* Guo et al. (2017). Later methods add global geometric regularizers—C-TPT Yoon et al. (2024) (isotropic dispersion), O-TPT Sharifdeen et al. (2025) (orthogonality), and A-TPT Ahamed et al. (2026) (angular diversity)—to stabilize predictions, but these constraints remain *static* and *sample-agnostic*: they apply identical update pressure regardless of instance uncertainty and are blind to the evolving target-domain visual distribution. Moreover, the pretrained CLIP space encodes a meaningful *semantic topology*, where distances reflect linguistic relatedness (e.g., "cat"–"tiger"–"leopard"). Rigid constraints such as orthogonality Sharifdeen et al. (2025) or aggressive dispersion Yoon et al. (2024); Ahamed et al. (2026) can distort this structure during adaptation, decoupling semantic similarity from logit

---

[1]Code will be released upon acceptance.

[2]**ECE**: Expected Calibration Error, **SCE**: Static Calibration Error, **MCE**: Maximum Calibration Error, **ACE**: Adaptive Calibration Error, **Brier score**: Mean squared probability error, **NLL**: Negative Log-Likelihood. See **Appendix E** for further details, All results in Fig. 1 are averaged over 9 image-classification datasets, excluding ImageNet with CLIP-ViT B/16. See Table 1 for details.

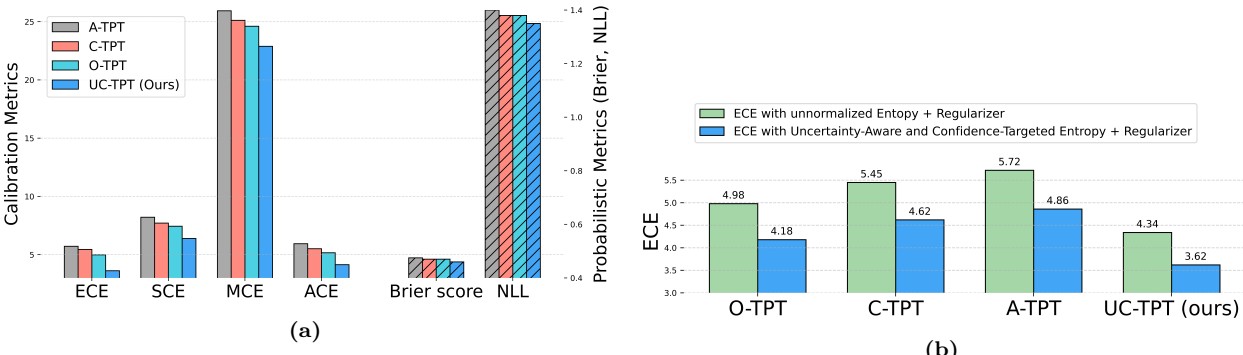

**Figure 1:** (a) **UC-TPT** consistently achieves superior calibration across multiple metrics $(\downarrow)^2$ surpassing TPT variants C-TPT Yoon et al. (2024), O-TPT Sharifdeen et al. (2025), and A-TPT Ahamed et al. (2026). (b) Replacing the unnormalized entropy in prior methods with our adaptive, uncertainty weighted formulation (Section 3.3.1) along with their respective regularizer markedly reduces miscalibration, showing that ignoring representational uncertainty hinders TTA. Adding our topology-weighted prompt regularizer (Section 3.3.2) further yields the lowest ECE $(\downarrow)$ in our case.

geometry and leading to unstable logit scaling and calibration drift under shift (Fig. 1a–b, Fig. 5). These observations suggest that reliable TPT needs (i) *uncertainty-aware* updates (to avoid sharpening unreliable predictions), (ii) *visual grounding* (to condition prompts on target-domain evidence), and (iii) *topology preservation* (to prevent semantic neighborhood collapse).

**Our Proposal.** We present **UC-TPT**, a unified framework that improves test-time prompt tuning with a *primary goal of reliability and calibration* (rather than maximizing accuracy) along three theoretically justified axes: (i) *visual-to-textual shallow conditioning*, (ii) *adaptive uncertainty-guided entropy optimization*, and (iii) *topology-weighted prompt regularization*.

First, UC-TPT introduces **shallow visual-to-text prompt conditioning**—a minimal, domain-grounded mechanism that injects early-layer visual shift statistics into prompt tokens via a lightweight cross-modal projection, yielding *instance-conditioned* prompts while keeping both encoders frozen (Fig. 7).

Second, UC-TPT proposes an **uncertainty-calibrated entropy objective** that *replaces uniform sharpening* with *risk-aware* test-time optimization. The core of this objective is a dynamic uncertainty gate that acts as a critical safety brake, attenuating destructive gradients from ambiguous or out-of-distribution samples to prevent representation collapse. Operating synergistically under this protection, a confidence-shaped target safely refines the predictive sharpness of well-calibrated samples, ensuring exceptional calibration stability without compromising the underlying semantic topology (Fig. 1a–b).

Third, UC-TPT introduces a **topology-weighted prompt diversity regularizer** that *preserves CLIP's semantic neighborhood structure* during adaptation by scaling repulsion according to manual-prompt semantic distances. Unlike hard orthogonality Sharifdeen et al. (2025) or uniform angular dispersion Ahamed et al. (2026), this soft cosine-weighted constraint improves separability *without* topology-breaking drift, stabilizing inter-class logit geometry and reducing gradient volatility (Fig. 1b, Fig. 4b, Fig. 5).

Together, these contributions provide a **principled decomposition of test-time prompt tuning for calibration**: visual conditioning supplies *domain evidence*, uncertainty calibration decides *update magnitude per instance*, and topology weighting constrains *update direction* to respect pretrained semantics. This synergy yields stable test-time dynamics and consistently better calibration across datasets and backbones (Fig. 1a–b).

**Our major contributions are:**

- We propose **UC-TPT**, an uncertainty-calibrated test-time prompt tuning framework that targets *reliable confidence* under distribution shifts.

- UC-TPT combines *shallow visual conditioning* for domain-aware grounding, an *uncertainty-guided entropy objective* for selective confidence updates, and a *topology-weighted regularizer* that preserves semantic geometry while encouraging diversity.

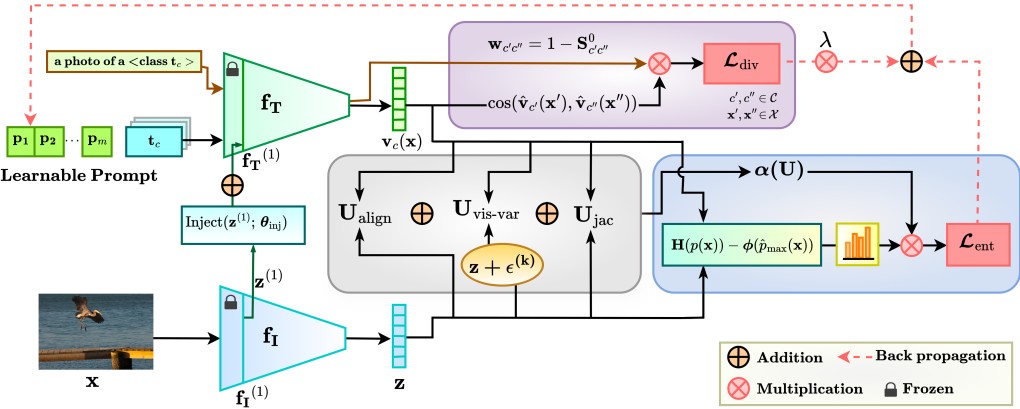

**Figure 2:** Overview of the **UC-TPT** framework. Given an unseen test sample $\mathbf{x}$, UC-TPT performs instance-wise adaptation of textual prompts $\mathbf{s_c(P)}$ through uncertainty-guided updates without requiring labeled data. The frozen CLIP encoders with visual conditioning (Inject$(\mathbf{z}; \boldsymbol{\theta}_{\text{inj}})$) extract visual $\mathbf{z}$ and textual $\mathbf{v}_c(\mathbf{x})$ embeddings. An uncertainty module estimates alignment-based ($\mathbf{U}_{\text{align}}$), variance-based ($\mathbf{U}_{\text{vis-var}}$), and Jacobian-based ($\mathbf{U}_{\text{jac}}$) uncertainty components, fused via a gating function $\boldsymbol{\alpha}(\cdot)$ to modulate the adaptive confidence-aware entropy loss $\mathcal{L}_{\text{ent}}$. Simultaneously, the semantics preserving diversity loss $\mathcal{L}_{\text{div}}$ enforces discriminative consistency among prompts. Together, these components yield uncertainty-aware, topology-stable prompt adaptation that enhances calibration.

- Extensive evaluations across multiple architectures and distribution shifts demonstrate that our integrated UC-TPT recipe consistently anchors the high-reliability end of the Accuracy-ECE Pareto frontier. By securing substantial calibration improvements with only a highly efficient, marginal trade-off in accuracy, it offers a safer, risk-aware paradigm for real-world multimodal adaptation.

## 2 Related Works

**(a) Test-Time Tuning.** Deep neural networks achieve impressive accuracy yet remain vulnerable under distribution shifts Shimodaira (2000). TTA addresses this by allowing pretrained models to adapt online using unlabeled target data, without retraining. Prior TTA research has largely focused on entropy minimization Wang et al. (2020); Han et al. (2025) and consistency regularization Wang et al. (2022); Liu et al. (2023) or other confidence estimation for standard classifiers Lee et al. (2024). Within this paradigm, TPT Shu et al. (2022) optimizes prompt tokens with entropy minimization and view augmentation, achieving lightweight adaptation for VLMs and inspiring several extensions Xiao et al. (2025); Sheng et al. (2025); Feng et al. (2023). Recently, the reliability of these online adaptations has gained critical attention. For instance, NTTA Cao et al. (2025) addresses open-set noisy streams by training a decoupled auxiliary detection module, while CLIPTTA Lafon et al. (2025) abandons entropy minimization entirely in favor of a contrastive objective. While effective, these broader TTA approaches either introduce structural overhead or alter the fundamental learning mechanism. In contrast, our work strictly adheres to the lightweight TPT paradigm. We demonstrate that miscalibration and semantic distortion can be prevented natively within the prompt space by mathematically correcting the flaw in standard entropy minimization via an uncertainty gate, avoiding the need for auxiliary networks or contrastive pair tracking.

**(b) Uncertainty Estimation.** Uncertainty quantification in deep models has been approached via Bayesian networks and approximate inference, e.g., Monte Carlo dropout for epistemic uncertainty Gal & Ghahramani (2016), or ensemble-based methods capturing predictive variance Lakshminarayanan et al. (2017). Predictive entropy provides a scalar confidence proxy Depeweg et al. (2017), while perturbation-based methods estimate both aleatoric and epistemic uncertainty Ovadia et al. (2019). In VLMs, alignment confidence between image–text embeddings serves as a natural uncertainty cue Shu et al. (2022), and prompt sensitivity has been shown to correlate with calibration quality Yoon et al. (2024). Bayesian prompt ensembles Tonolini et al. (2024); Daneshfar et al. (2025), stochastic embedding models Pautsch et al. (2023); Erick et al. (2024), and Bayesian adapters Alvarez et al. (2024) further enhance reliability. However, most of these approaches

remain offline, with limited exploration of dynamic, sample-level uncertainty modeling designed specifically to gate online prompt tuning under distribution shifts.

**(c) Calibration in VLMs.** CLIP-based models exhibit strong zero-shot generalization but often become miscalibrated under domain shifts, where prediction confidence diverges from correctness. Previous works address this via temperature scaling using text–class distances Wang et al. (2024), unseen-label augmentation Wang et al. (2025), dominant-dimension suppression Han & Hwang (2025), or covariance-regularized tuning Oh et al. (2024). Within TPT frameworks, calibration is commonly enforced through geometric constraints such as dispersion Yoon et al. (2024); Ahamed et al. (2026) or orthogonality Sharifdeen et al. (2025), while cross-regularized tuning Li et al. (2023) and logit-range normalization Murugesan et al. (2024) further stabilize adaptation. Concurrently, Semantic Orthogonal Calibration (SoC) Fillioux et al. (2026) mitigates miscalibration by enforcing semantic orthogonality based on class proximity. However, many of these methods apply uniform regularization across all samples, ignoring instance-specific predictive risk. Furthermore, rigid geometric constraints (e.g., angular orthogonality) can artificially distort CLIP's pretrained semantic topology. In contrast, we position UC-TPT as an **integrated calibration recipe**: it synthesizes the preservation of exact zero-shot semantic topology with a dynamic, sample-level uncertainty gate, preventing catastrophic overconfidence without overriding the model's foundational inter-class relationships.

## 2.1 Problem Formulation and Background

We consider a pretrained vision–language model (VLM), such as CLIP Radford et al. (2021c), consisting of a frozen image encoder $\mathbf{f}_I : \mathcal{X} \to \mathbb{R}^d$ and a frozen text encoder $\mathbf{f}_T : \mathcal{T} \to \mathbb{R}^d$, where $\mathcal{X}$ and $\mathcal{T}$ denote the image and text spaces, respectively. Both encoders map inputs to a shared $d$-dimensional embedding space. Let $\mathcal{C} = \{1, \ldots, C\}$ denote the set of semantic classes. Each class $c \in \mathcal{C}$ is associated with a textual template $\mathbf{t}_c \in \mathcal{T}$.

**Prompt Tokens and Class Representations.** Prompt tuning augments each class template with a sequence of $m$ learnable prompt tokens $\mathbf{P} = [\mathbf{p}_1, \ldots, \mathbf{p}_m] \in \mathbb{R}^{m \times d}$. The resulting class-specific prompt is defined as

$$\mathbf{s}_c(\mathbf{P}) = [\mathbf{p}_1, \ldots, \mathbf{p}_m, \mathbf{t}_c], \tag{1}$$

which is encoded by the text encoder to produce a class embedding

$$\mathbf{v}_c = \mathbf{f}_T\big(\mathbf{s}_c(\mathbf{P})\big) \in \mathbb{R}^d. \tag{2}$$

**Prediction Model.** Given an image $\mathbf{x} \in \mathcal{X}$, the image encoder outputs a normalized visual embedding $\mathbf{z} = \mathbf{f}_I(\mathbf{x}) \in \mathbb{R}^d$. The model predicts class probabilities via a temperature-scaled softmax over cosine similarities:

$$p_c(\mathbf{x}; \mathbf{P}) = \frac{\exp\big(\mathbf{z}^\top \mathbf{v}_c / \tau\big)}{\sum_{c'=1}^{C} \exp(\mathbf{z}^\top \mathbf{v}_{c'} / \tau)}, \tag{3}$$

where $\tau > 0$ denotes a temperature parameter.

**Test-Time Adaptation (TTA).** At test time, the model is exposed to unlabeled target samples $\mathcal{D}_t = \{\mathbf{x}_j\}_{j=1}^{n}$ drawn from an unknown target distribution $\mathcal{P}_t$. Following the test-time prompt tuning (TPT) paradigm Shu et al. (2022); Sharifdeen et al. (2025), adaptation is performed by optimizing only the prompt parameters $\mathbf{P}$, while keeping both encoders frozen. The test-time adaptation objective is formulated as

$$\min_{\mathbf{P}} \ \mathbb{E}_{\mathbf{x} \sim \mathcal{P}_t}\big[\mathcal{L}_{\text{TTA}}(\mathbf{x}; \mathbf{P})\big], \tag{4}$$

where $\mathcal{L}_{\text{TTA}}$ is a self-supervised loss, typically based on entropy minimization or its regularized variants.

**Calibration and Confidence Reliability.** Calibration measures the alignment between predictive confidence and empirical accuracy Guo et al. (2017). In practice, it is quantified using Expected Calibration Error (ECE), which compares accuracy and confidence across confidence bins (see Appendix E for the formal derivation). Poor calibration is particularly harmful in test-time prompt tuning, where entropy minimization can over-sharpen uncertain predictions under domain shift, leading to overconfident errors.

# 3 Taking through the UC-TPT Framework

Our UC-TPT is a unified, calibration-aware prompt tuning framework that performs uncertainty-guided adaptation of vision–language models at test time (Fig. 2). It couples instance-aware visual conditioning (Section 3.1), multi-signal uncertainty estimation (Section 3.2), and adaptive entropy optimization with a prompt regularizer that preserves linguistic geometry (Section 3.3). The subsequent sections describe each component in detail.

## 3.1 Shallow Visual-to-Textual Conditioning

Conventional TPT methods Shu et al. (2022); Sharifdeen et al. (2025) adapt prompts with a *single, sample-agnostic* parameterization, applying identical updates to all test images. Under domain shift, this is limiting: the mismatch largely resides in *visual* features, yet adaptation occurs only in prompt space without injecting target-domain evidence into text representations.

A natural remedy is stronger cross-modal coupling. MaPLe Khattak et al. (2023), for example, couples text and image branches across multiple layers, including text→image pathways. While effective for supervised prompt learning, such deep bi-directional coupling is less suited for *test-time* tuning: it increases update capacity and lets transient test-time signals steer visual features, potentially distorting CLIP's pretrained geometry and destabilizing calibration (higher ECE in Table 15).

UC-TPT instead adopts *minimal conditioning*: make prompts input-dependent via a *one-way, shallow* visual→text injection while keeping both encoders frozen. By injecting early-layer shift statistics—where shift is strongest and least entangled with class semantics—UC-TPT grounds adaptation without deep coupling. Fig. 3 shows shallow injection improves calibration over deeper injection and no-conditioning, yielding a favorable stability–adaptivity trade-off.

**Definition 3.1** (Shallow Visual-to-Textual Injection). Let $\mathbf{f_I}^{(1)}$ and $\mathbf{f_T}^{(1)}$ denote the first layers of the frozen image and text encoders. Given an image $\mathbf{x}$, the first-layer visual embedding is

$$\mathbf{z}^{(1)} = \mathbf{f_I}^{(1)}(\mathbf{x}) \in \mathbb{R}^d. \tag{5}$$

A lightweight MLP projection $\text{Inject}(\cdot; \boldsymbol{\theta}_{\text{inj}})$ with `GELU` (details in **Appendix B**) maps $\mathbf{z}^{(1)}$ to

$$\mathbf{q_x} = \text{Inject}(\mathbf{z}^{(1)}; \boldsymbol{\theta}_{\text{inj}}) \in \mathbb{R}^{1 \times m}. \tag{6}$$

Let $\mathbf{P} = \{\mathbf{p}_i\}_{i=1}^m$ be base prompt tokens and $\mathbf{P}^{(1)} = \mathbf{f_T}^{(1)}([\mathbf{p}_1, \ldots, \mathbf{p}_m])$ their first-layer embeddings. The conditioned prompt is

$$\tilde{\mathbf{P}}^{(1)}(\mathbf{x}) = [\mathbf{P}_1^{(1)} + \mathbf{q_{x1}}, \ldots, \mathbf{P}_m^{(1)} + \mathbf{q_{x}}_m]. \tag{7}$$

For class template $\mathbf{t}_c$, the conditioned class embedding is

$$\mathbf{v}_c(\mathbf{x}) = \mathbf{f_T}^{(>1)}(\mathbf{s}_c(\tilde{\mathbf{P}}^{(1)}(\mathbf{x}))), \tag{8}$$

where $\mathbf{s}_c$ is defined in Eq. 1; prediction follows Eq. 3.

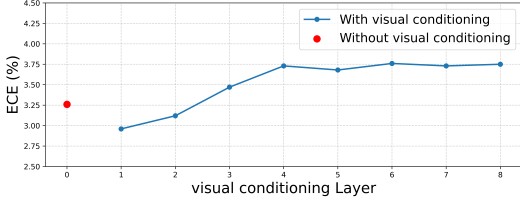

**Figure 3:** Effect of visual conditioning depth on ECE for Caltech101 Fei-Fei et al. (2004) using CLIP–ViT/B/16, comparing shallow conditioning to deeper and unconditioned variants.

**Remark.** UC-TPT transforms *static* prompts into *instance-conditioned* ones through lightweight visual conditioning, incurring negligible computational cost ($< 0.1\%$) while keeping the visual backbone and deeper text layers frozen. By computing a single gradient step over augmented views, UC-TPT extracts a robust visual bias for the test time update without the instability typically associated with extended optimization. This design ensures a fast, lightweight adaptation that preserves the calibrated nature of the pretrained embedding space.

## 3.2 Per-Sample Uncertainty Quantification

Prior TPT methods Shu et al. (2022); Ahamed et al. (2026); Sharifdeen et al. (2025) apply *uniform* adaptation, implicitly assuming equal reliability across test samples. Under distribution shift, instances vary widely in ambiguity and local stability; uniform entropy minimization indiscriminately sharpens posteriors, often *amplifying overconfidence* on hard or OOD inputs and degrading calibration.

UC-TPT instead estimates *per-sample uncertainty* to control test-time updates. Rather than using heavy post-hoc Bayesian tools or many stochastic forward passes Brahma & Rai (2023); Tan et al. (2024), we design an *on-the-fly*, label-free estimator that is (i) *multimodal* (image–text agreement and predictive stability), (ii) *optimization-aware* (prompt-update fragility), and (iii) *lightweight* (negligible overhead beyond standard TPT).

**Why uncertainty, and what should it measure?**   The key failure mode under shift is *misplaced confidence*: confident errors arise when inputs are off-manifold, image–text alignment is weak, or predictions are unstable to small perturbations. Accordingly, UC-TPT models three complementary "adaptation risks": (i) *semantic* risk (weak alignment to any class text prototype), (ii) *predictive* risk (output variance under small visual changes), and (iii) *optimization* risk (sensitivity to prompt updates). These cues act at representation, prediction, and update levels, providing non-redundant uncertainty for selective adaptation.

**Unified uncertainty score.**   Given a test sample $\mathbf{x} \in \mathcal{X}$ with visual embedding $\mathbf{z}$ and class text embeddings $\mathbf{v}_c(\mathbf{x})$ (Eq. 8), we define the total uncertainty $\mathbf{U}(\mathbf{x})$ as the sum of three efficiently computable components.

**Definition 3.2** (Image–Text Alignment Uncertainty)**.** If the visual embedding $\mathbf{z}$ is poorly aligned with *all* class text embeddings $\{\mathbf{v}_c(\mathbf{x})\}$, the sample is likely off-manifold or semantically ambiguous. We define

$$\mathbf{U}_{\text{align}}(\mathbf{x}) = 1 - \max_{c \in \mathcal{C}} \cos(\mathbf{z}, \mathbf{v}_c(\mathbf{x})), \tag{9}$$

where larger values indicate weaker multimodal agreement (higher semantic risk).

**Definition 3.3** (Visual Perturbation Variance)**.** A reliable predictor should be locally stable: small perturbations in the visual representation should not cause large changes in the output distribution. We inject Gaussian noise into $\mathbf{z}$ to obtain perturbed embeddings

$$\mathbf{z}^{(k)} = \mathbf{z} + \boldsymbol{\epsilon}^{(k)}, \quad \boldsymbol{\epsilon}^{(k)} \sim \mathcal{N}(\mathbf{0}, \sigma^2 \mathbf{I}), \tag{10}$$

and compute

$$p^{(k)} = \text{softmax}\big( \cos(\mathbf{z}^{(k)}, \{\mathbf{v}_c(\mathbf{x})\}_{c=1}^{C}) \big). \tag{11}$$

Let $\bar{p}_v = \frac{1}{K} \sum_{k=1}^{K} p^{(k)}$. The visual variance uncertainty is

$$\mathbf{U}_{\text{vis-var}}(\mathbf{x}) = \frac{1}{K} \sum_{k=1}^{K} \|p^{(k)} - \bar{p}_v\|_2^2. \tag{12}$$

This term captures local predictive instability (predictive risk).

**Definition 3.4** (Prompt Sensitivity)**.** Even when predictions are sharp, test-time updates can be brittle: if the loss landscape is steep w.r.t. prompts, small prompt steps can induce large distributional shifts. We quantify this *optimization risk* via the prompt Jacobian:

$$\mathbf{U}_{\text{jac}}(\mathbf{x}) = \sum_{c \in \mathcal{C}} \big(1 - \cos(\mathbf{z}, \mathbf{v}_c(\mathbf{x}))\big) \Big\| \frac{\partial p_c}{\partial \mathbf{P}} \Big\|_2, \tag{13}$$

where the alignment term emphasizes gradients along semantically unreliable directions.

**Computational Complexity.**   To ensure test-time efficiency, UC-TPT employs several lightweight approximations. $\mathbf{U}_{\text{align}}$ reuses standard CLIP cosine similarities, adding no overhead. $\mathbf{U}_{\text{vis-var}}$ operates directly in the low-dimensional embedding space via $K$ noise samples ($K = 5$). Naïvely, $\mathbf{U}_{\text{jac}}$ would require explicit Jacobian trace computation scaling with the total number of classes $C$; instead, we restrict it to the top-$\tilde{C}$ classes that capture $> 99.9\%$ of the probability mass and approximate the trace with a Hutchinson estimator Avron & Toledo (2011). As a result, the asymptotic complexity is bounded as follows:

**Theorem 3.5** (UC-TPT Complexity). *Let $N$ denote the batch size, $C_B \ll C$ the active classes in a batch, $d$ the embedding dimension, and $K$ the number of perturbations. The total computational complexity of UC-TPT is:*

$$\mathcal{C}_{\text{UC-TPT(H)}} = \mathcal{O}(KN\tilde{C} + C_B^2 d + Nd)$$

By leveraging $\tilde{C} \ll C$ and $C_B \ll C$, this formulation entirely bypasses the strict $\mathcal{O}(C^2 d)$ bottleneck of prior class-interaction regularizers (e.g., A-TPT, O-TPT). A full formal derivation (including intermediate lemmas) and empirical validation demonstrating that UC-TPT maintains peak memory and runtime comparable to baselines while significantly improving calibration are provided in Appendix H.

**Total uncertainty.** The final per-sample uncertainty is

$$\mathbf{U}(\mathbf{x}) = \mathbf{U}_{\text{align}}(\mathbf{x}) + \mathbf{U}_{\text{vis-var}}(\mathbf{x}) + \mathbf{U}_{\text{jac}}(\mathbf{x}). \tag{14}$$

**Theoretical insight and fusion strategy.** Eq. 14 bounds three complementary failure modes: weak multimodal agreement ($\mathbf{U}_{\text{align}}$), local output sensitivity ($\mathbf{U}_{\text{vis-var}}$), and prompt-space optimization instability ($\mathbf{U}_{\text{jac}}$). Together, $\mathbf{U}(\mathbf{x})$ acts as a lightweight surrogate for *expected calibration risk*, gating entropy updates when any stability metric fails. We intentionally adopt unweighted additive fusion. Because these metrics track independent risks, summation acts as a continuous "OR" gate: if any single risk spikes, total uncertainty scales to protect the model. This cleanly avoids the manual tuning of weighted alternatives while maintaining robust cross-domain performance (ablation in Appendix F).

### 3.3 Proposed Loss Functions

UC-TPT updates prompts at test time using two complementary objectives: an *Adaptive Uncertainty-Aware Entropy Loss* (Eq. 18) and a *Topology-Weighted Prompt Diversity Regularizer* (Sec. 3.3.2). The first controls *when* and *how strongly* to sharpen predictions (sample-wise reliability), while the second constrains *where* prompts are allowed to move (semantic geometry), yielding stable adaptation by bounding logit variance and stabilizing gradients (Fig. 4(b)).

#### 3.3.1 Adaptive Uncertainty-Aware Entropy Loss

Standard entropy minimization sharpens predictions but can harm calibration by treating all samples equally and propagating gradients from uncertain/OOD inputs. UC-TPT instead formulates test-time entropy optimization as a *heteroscedastic* objective: both the entropy target and the update weight depend on confidence and per-sample uncertainty, preventing indiscriminate over-sharpening.

For $\mathbf{x} \in \mathcal{X}$ with probabilities $p_c(\mathbf{x})$ (Eq. 3), the predictive entropy is

$$\mathbf{H}(p(\mathbf{x})) = -\sum_{c=1}^{C} p_c(\mathbf{x}) \log(p_c(\mathbf{x}) + \nu), \tag{15}$$

where $\nu$ is a stability constant and $\hat{p}_{\max} = \max_c p(y{=}c \mid \mathbf{x}; \mathbf{P})$ denotes confidence. Prior TPT methods Shu et al. (2022); Ahamed et al. (2026); Sharifdeen et al. (2025) directly minimize $\mathbf{H}$, which implicitly drives all samples toward near-delta posteriors. Under shift, this increases the risk of *confident mistakes*. UC-TPT makes two key changes.

**(i) Confidence-shaped entropy target.** We set

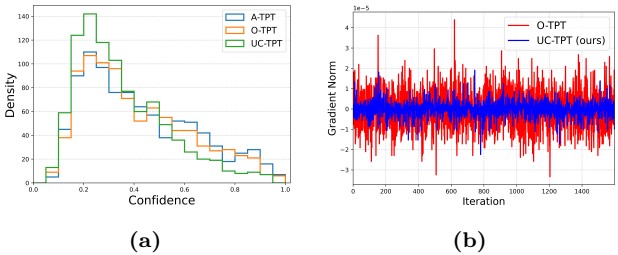

**(a)**             **(b)**

**Figure 4:** On DTD Cimpoi et al. (2014) with CLIP–ViT-B/16: (a) Max confidence of misclassified samples: UC-TPT shifts errors to lower confidence than Sharifdeen et al. (2025); Ahamed et al. (2026). (b) Gradient-norm variance: $\mathcal{L}_{\text{div}}$ has tighter variance than O-TPT Sharifdeen et al. (2025).

$$\phi(\hat{p}_{\max}(\mathbf{x})) = \log C \cdot (1 - \hat{p}_{\max}(\mathbf{x})), \tag{16}$$

which lies in $[0, \log C]$. Intuitively, $\phi$ establishes a soft, confidence-aligned theoretical target: for confident predictions, $\phi$ approaches zero, encouraging necessary sharpening. For low-confidence samples, $\phi$ remains large, defining a mathematically conservative boundary.

However, enforcing this quadratic target uniformly across noisy domains can still induce gradient instability. Therefore, this *data-dependent prior* is explicitly designed to operate synergistically with our dynamic uncertainty gate (introduced next). While $\phi$ sets the ideal posterior sharpness, it is the uncertainty gate that provides the critical gradient attenuation necessary to safely operationalize this target and definitively prevent representation collapse.

**(ii) Uncertainty-gated update weight.** Let $\mathbf{U}(\mathbf{x})$ be the unified uncertainty (Eq. 14) and $\mathbf{U}_{\mathrm{norm}} = \mathrm{norm}(\mathbf{U}(\mathbf{x}))$ **min-max normalization, (Appendix F).** We define

$$\boldsymbol{\alpha}(\mathbf{U}_{\mathrm{norm}}) = \frac{1}{1 + \exp\big(k_0(\mathbf{U}_{\mathrm{norm}} - u_0)\big)}, \tag{17}$$

where $k_0$ controls the slope and $u_0$ the midpoint. The sigmoid yields smooth, monotone attenuation: reliable samples receive stronger updates and uncertain/OOD samples are down-weighted **(Appendix K).** The smoothness is important for test-time stability: it avoids hard thresholding that can cause abrupt regime switches across nearby samples.

The final objective is

$$\mathcal{L}_{\mathrm{ent}}(\mathbf{x}) = \boldsymbol{\alpha}(\mathbf{U}_{\mathrm{norm}}(\mathbf{x}))\big(\mathbf{H}(p(\mathbf{x})) - \boldsymbol{\phi}(\hat{p}_{\mathrm{max}}(\mathbf{x}))\big)^2. \tag{18}$$

**Theoretical insight (safe sharpening as inverse-variance weighting).** Eq. 18 adopts a heteroscedastic least-squares form: the dynamic gate $\boldsymbol{\alpha}(\mathbf{U})$ serves as a critical *inverse-variance weight*, suppressing destructive gradients from samples with high uncertainty (high estimated noise/instability). Protected by this attenuation, $\boldsymbol{\phi}(\hat{p}_{\mathrm{max}})$ safely operationalizes a confidence-aligned target entropy. This replaces *uniform* entropy minimization with *risk-aware* sharpening: the target $\boldsymbol{\phi}$ is only permitted to drive adaptation on samples that the gate $\boldsymbol{\alpha}$ explicitly deems stable. Consequently, UC-TPT shifts misclassified samples toward lower confidence and yields smoother confidence evolution (Fig. 4(a), Fig. 15(a,b)). Further theoretical insight is in **Appendix I.**

### 3.3.2 Topology-Weighted Prompt Diversity Regularizer

CLIP's manual prompts induce a semantic topology where cosine proximity reflects linguistic relatedness, providing a strong prior for zero-shot generalization. Uniform diversity constraints can violate this prior: strict orthogonality Sharifdeen et al. (2025) forces *equal* separation even for semantically close classes, while uniform angular inflation Ahamed et al. (2026) can over-repel dense semantic neighborhoods, distorting inter-class logit ratios and harming calibration. UC-TPT therefore enforces *topology-aware* diversity: separation strength is proportional to pretrained semantic distance, improving discriminability without breaking local neighborhoods (Qualitative analysis is reported in Appendix U).

**Definition.** Let $\hat{\mathbf{v}}(\mathbf{x}) \in \mathbb{R}^d$ be the $\ell_2$-normalized *conditioned prompt embedding* and $\tilde{c} = \arg\max_c p(c \mid \mathbf{x})$ the pseudo-label. Define

$$\mathbf{w}_{ij} = 1 - \mathbf{S}_{\tilde{c}_i \tilde{c}_j}^0, \qquad \mathbf{S}_{\tilde{c}_i \tilde{c}_j}^0 = \cos(\mathbf{v}_{\tilde{c}_i}^0, \mathbf{v}_{\tilde{c}_j}^0), \tag{19}$$

where $\mathbf{v}_c^0$ is the frozen manual-prompt embedding (`"a photo of a <class>"`). For a minibatch $\{\mathbf{x}_i\}_{i=1}^B$,

$$\mathcal{L}_{\mathrm{div}} = \sum_{1 \leq i < j \leq B} \mathbf{w}_{ij} \cos\big(\hat{\mathbf{v}}(\mathbf{x}_i), \hat{\mathbf{v}}(\mathbf{x}_j)\big). \tag{20}$$

**Theoretical insight (graph-based stabilization).** Eq. 20 can be interpreted as a soft semantic-graph constraint: $\mathbf{S}^0$ defines a target similarity graph from pretrained manual prompts, and the weighting $\mathbf{w}_{ij}$ prevents collapsing edges that should remain close while promoting separation where the prior indicates dissimilarity. This reduces topology-breaking prompt drift and stabilizes the inter-class logit geometry during test-time optimization. Empirically, UC-TPT increases dispersion of pairwise similarities with minimal mean shift (Fig. 5), indicating improved separability without erasing the pretrained semantic scaffold **(Appendix J).**

**Overall Objective.** The final test-time adaptation objective combines uncertainty-aware entropy minimization with topology-weighted prompt diversity, where $\lambda$ balances entropy minimization and prompt diversity:

$$\mathcal{L}_{\text{TTA}} = \mathcal{L}_{\text{ent}} + \lambda\,\mathcal{L}_{\text{div}}. \tag{21}$$

Fig. 1(b) shows consistent ECE reduction when $\mathcal{L}_{\text{ent}}$ is applied across baseline methods (see Table 13), while combining $\mathcal{L}_{\text{ent}}$ & $\mathcal{L}_{\text{div}}$ yields further gains for UC-TPT, highlighting the broader applicability of this principle in TPT.

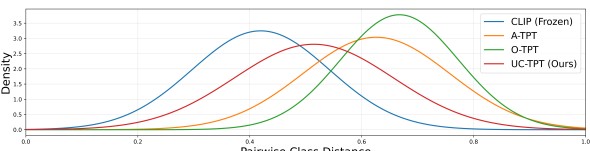

**Figure 5:** On Pets Parkhi et al. (2012) with CLIP–ViT/B-16: fitted Gaussians of pairwise prompt similarities. UC-TPT increases dispersion with minimal mean shift, improving separation while preserving semantics.

## 4 Experimental Evaluations

**Datasets.** We evaluate on a comprehensive suite of datasets encompassing general, fine-grained, robustness, medical, Remote sensing, and cross-domain benchmarks using CLIP and BiomedCLIP. Notably, this includes natural distribution shifts evaluated through ImageNet variants as well as specialized remote sensing and texture tasks. Full dataset descriptions and configuration details are provided in Appendix C.

**Implementation Details, Comparative Methods, and Evaluation Protocol.** We evaluate UC-TPT with CLIP-RN50 and CLIP-ViT-B/16 under a unified test-time prompt tuning (TPT) setup, comparing against calibration-aware and TPT-based baselines under identical settings. All methods use a single AdamW update (lr $= 5 \times 10^{-3}$), hard prompt initialization, and a frozen CLIP backbone. For UC-TPT, following A-TPT, we set $\lambda$ in Eq. 21 to 80, and $k_0$ and $u_0$ in Eq. 17 to 10 and 0.01. Implementation details, hyperparameters, and evaluation protocols are provided in **Appendix D** & **K.** Further analysis of different prompt learning frameworks and different prompt initializations, different combination of regularizers, Batch size analysis, adaptation step analysis, and additional ablations reported in **Appendix**.

### 4.1 Main Results

**Table 1: ECE** ($\downarrow$) with CLIP-ViT-B/16 backbone across ten datasets. The overall best-performing result is in **bold**, and the second best is underlined. All baselines are reimplemented under reported settings for fair comparison. Standard deviation information across multiple seeds is provided in the Appendix L.

| Method | Metric | INet | DTD | FLW | Food | Air. | Pets | C101 | UCF | SAT | Car | Avg. |
|---|---|---|---|---|---|---|---|---|---|---|---|---|
| Zero-Shot | Acc. | 66.8 | 44.5 | 67.4 | 83.8 | 23.8 | 88.1 | 92.9 | 65.0 | 41.3 | 65.4 | 63.9 |
| | ECE | 2.03 | 8.28 | 2.61 | 2.32 | 5.17 | 4.44 | 5.50 | 3.26 | 7.51 | 4.26 | 4.54 |
| TPT | Acc. | 69.0 | 46.7 | 69.1 | 84.6 | 23.9 | 87.1 | 93.8 | 67.4 | 42.5 | 66.2 | 65.0 |
| | ECE | 10.54 | 21.30 | 13.23 | 3.94 | 16.37 | 5.42 | 4.52 | 12.56 | 21.73 | 5.36 | 11.50 |
| R-TPT | Acc. | 66.7 | 44.5 | 67.2 | 83.8 | 23.9 | 88.1 | 92.9 | 64.9 | 41.4 | 65.3 | 63.9 |
| | ECE | 10.30 | 18.80 | 10.80 | 3.30 | 12.60 | 5.40 | 3.60 | 12.10 | 22.00 | 1.93 | 10.10 |
| C-TPT | Acc. | 68.2 | 46.0 | 69.7 | 83.3 | 23.9 | 88.2 | 93.4 | 65.3 | 43.2 | 65.5 | 64.7 |
| | ECE | 3.19 | 12.50 | 5.13 | 3.72 | 4.33 | 1.83 | 4.34 | 2.40 | 13.30 | 1.56 | 5.22 |
| A-TPT | Acc. | 67.7 | 45.9 | 65.9 | 83.2 | 23.1 | 84.1 | 92.0 | 63.3 | 42.9 | 62.9 | 63.1 |
| | ECE | 2.26 | 6.15 | 4.19 | 7.08 | 6.32 | 5.85 | 10.10 | 3.63 | 4.46 | 5.52 | 5.55 |
| O-TPT | Acc. | 67.3 | 45.6 | 69.2 | 82.8 | 23.3 | 88.0 | 93.1 | 64.2 | 43.3 | 64.9 | 64.2 |
| | ECE | 1.98 | 8.08 | 3.87 | 4.63 | 3.97 | 1.96 | 4.64 | 2.28 | 13.80 | 1.61 | 4.68 |
| **UC-TPT (Ours)** | **Acc.** | 68.2 | 44.3 | 67.8 | 83.3 | 24.0 | 88.4 | 93.4 | 63.7 | 42.4 | 64.4 | 64.0 |
| | **ECE** | **1.53** | **4.74** | 3.97 | 3.23 | **2.65** | 2.65 | **2.96** | 2.70 | 8.36 | **1.35** | **3.41** |

**Cross-Domain Performance (ViT-B/16).** Table 1 reports calibration across ten cross-domain benchmarks using CLIP-ViT-B/16. UC-TPT attains the lowest mean ECE (3.41) while maintaining competitive

**Figure 6: Comparison of ECE (↓) on 11 BioMed datasets** using BioMedCLIP Zhang et al. (2024) as the backbone. UC-TPT achieves the lowest average ECE across all baselines.

accuracy (64.0%) relative to Zero-Shot (63.9%) and baseline adaptation methods. Compared to O-TPT (4.68 ECE), UC-TPT reduces ECE by 1.27 points. Improvements are notable on high-variance datasets like DTD (8.08 to 4.74) and SAT (13.80 to 8.36), where uniform regularizers in O-TPT and A-TPT can over-constrain the prompt manifold. By down-weighting risky samples, UC-TPT better preserves the pre-trained semantic topology.

**Cross-Domain Performance (RN50).** Table 2 demonstrates these benefits transfer to the CNN-based CLIP-RN50 backbone. Across the same datasets, UC-TPT achieves an average ECE of 4.76, improving upon O-TPT (5.84) by 1.08 points. Concurrently, UC-TPT maintains an average accuracy of 56.81%, remaining highly competitive with baselines while enhancing calibration reliability.

**Table 2: Comparison of average acc. / ECE performance** (↓) with CLIP-RN50 backbone on ten datasets as per Table 1.

| Method | Avg. Acc./ECE |
|---|---|
| Zero-Shot | 55.65 / 5.60 |
| TPT | 57.44 / 11.88 |
| R-TPT | 55.81 / 11.71 |
| C-TPT | 57.25 / 6.66 |
| A-TPT | 55.10 / 7.50 |
| O-TPT | 56.92 / 5.84 |
| **UC-TPT (Ours)** | 56.81 / **4.76** |

**Table 3: Average acc. / ECE performance** (↓) across CLIP-RN50 and ViT-B/16 backbones in natural distribution shift datasets (ImageNet-A, V2, R, S).

| Method | RN50 (Acc./ECE) | ViT-B/16 (Acc./ECE) |
|---|---|---|
| Zero-Shot | 40.60 / 7.18 | 57.16 / 4.90 |
| TPT | 43.53 / 16.75 | 60.20 / 11.70 |
| R-TPT | 40.60 / 16.57 | 57.16 / 11.02 |
| C-TPT | 41.70 / 8.92 | 58.37 / 5.26 |
| A-TPT | 42.93 / 14.22 | 59.34 / 7.95 |
| O-TPT | 41.72 / 8.93 | 58.49 / 5.06 |
| **UC-TPT (Ours)** | 41.88 / **6.88** | 57.16 / **4.63** |

**Natural Distribution Shifts.** Table 3 evaluates natural distribution shifts (ImageNet-A, -V2, -R, -S). UC-TPT achieves the lowest average ECE on both backbones while preserving comparable accuracy. For RN50, UC-TPT yields an average ECE of 6.88, compared to C-TPT (8.92) and O-TPT (8.93). For ViT-B/16, UC-TPT reaches 4.63 ECE (vs. O-TPT's 5.06). Extended analyses, including per-dataset breakdowns and additional shifts (PACS, DomainNet), are detailed in Appendices L and M.

**Biomedical Domain Robustness.** On biomedical benchmarks utilizing BioMedCLIP Zhang et al. (2024) (Fig. 6), UC-TPT achieves the lowest average ECE (45.10) relative to A-TPT (48.30) and other baselines (> 58.00). This calibration improvement coincides with an accuracy gain, achieving 34.43% versus standard TPT (33.02%) and A-TPT (33.59%). While high absolute error rates confirm that clinical deployment remains challenging, reducing overconfidence improves reliability in such high-risk domains. Extended analysis of RemoteCLIP Liu et al. (2024) across 7 remote sensing datasets, including full accuracy tables for both benchmarks, is provided in Appendix R.

**Accuracy vs. ECE trade-off.** Improving calibration often entails a minor decrease in peak accuracy Kumar et al. (2018); Karandikar et al. (2021); Yoon et al. (2023). Standard TPT methods uniformly update all samples, which can force marginally higher accuracy but risks severe overconfidence under domain shifts due to noisy self-training signals. UC-TPT addresses this through *risk-aware adaptation*—selectively gating uncertain samples while safely sharpening reliable ones—and topology-aware regularization to constrain prompt drift. Consequently, UC-TPT demonstrates an efficient Pareto trade-off (analysis shown in Appendix X), accepting a fractional accuracy reduction to significantly mitigate overconfident errors.

## 5 Ablation Analysis

We analyze the contribution of each component of UC-TPT and its individual impact on optimizing calibration (ECE) and accuracy. All ablation experiments are averaged over nine datasets from Table 1, excluding ImageNet, using the CLIP ViT-B/16 backbone unless otherwise specified.

**(i) Contribution of Each Uncertainty.** Table 4 analyzes the effect of individual uncertainty terms i.e. $\mathbf{U}_{\text{align}}$, $\mathbf{U}_{\text{vis-var}}$, and $\mathbf{U}_{\text{jac}}$. Using only $\mathbf{U}_{\text{align}}$ yields moderate calibration, while adding $\mathbf{U}_{\text{vis-var}}$ improves both metrics by stabilizing predictions under perturbations. Further including $\mathbf{U}_{\text{jac}}$, achieves the best trade-off, confirming that the uncertainties capture complementary cues, semantic consistency, prediction stability, and prompt sensitivity, respectively.

**Table 4: Ablation on uncertainty components.** Green ticks (✓) and red crosses (✗) indicate the inclusion or exclusion of each term.

| $\mathbf{U}_{\text{align}}$ | $\mathbf{U}_{\text{vis-var}}$ | $\mathbf{U}_{\text{jac}}$ | Avg. Acc. / ECE |
|:---:|:---:|:---:|:---:|
| ✓ | ✗ | ✗ | 63.04 / 4.10 |
| ✗ | ✓ | ✗ | 63.25 / 4.35 |
| ✗ | ✗ | ✓ | 63.01 / 4.08 |
| ✓ | ✓ | ✗ | 63.19 / 3.96 |
| ✓ | ✗ | ✓ | 63.08 / 3.86 |
| ✗ | ✓ | ✓ | 63.16 / 3.91 |
| ✓ | ✓ | ✓ | **63.51 / 3.62** |

**(ii) Impact of Model Components.** We ablate the effects of the topology-preserving regularizer $\mathcal{L}_{\text{div}}$, visual conditioning $\mathbf{q_x}$, and confidence-shaped target $\phi(\cdot)$. As shown in Fig. 7, incorporating these modules consistently improves both accuracy and calibration. Visual conditioning contributes the largest accuracy gain, while $\mathcal{L}_{\text{div}}$ most effectively lowers ECE by respecting the semantic topology of the frozen model.

**(iii) Addressing Under- and Over-Confidence.** Fig. 8 compares the reliability of A-TPT, O-TPT, and UC-TPT. The diagonal line represents perfect calibration. A-TPT exhibits underconfidence in the low-confidence region and overconfidence in the mid-confidence region. O-TPT remains largely overconfident, particularly around the mid-confidence levels. In contrast, UC-TPT closely follows the diagonal, demonstrating balanced confidence and improved reliability, with notably reduced overconfidence.

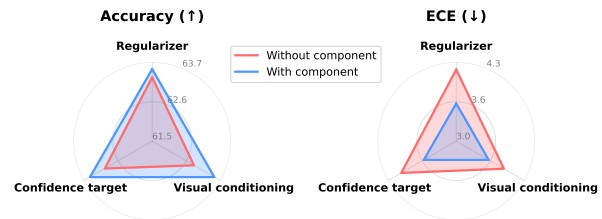

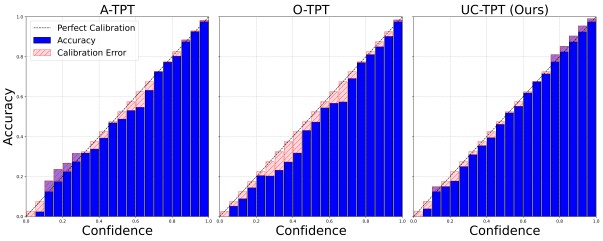

**Figure 7: Component-wise ablations** illustrating their impact.

**Figure 8: Reliability plots**, averaged over datasets, showing better calibration with UC-TPT.

**(iv) Isolating Uncertainty Modeling vs. Loss Shape.** To explicitly decouple the effects of the uncertainty-gated weight $\boldsymbol{\alpha}(\mathbf{U}_{\text{norm}})$ and the confidence-shaped target formulation $(\mathbf{H}(p) - \phi)^2$, we provide a controlled ablation study across six distinct datasets (DTD, Pets, Caltech, UCF, Flowers, Aircraft). Modifying both the weight and the loss shape simultaneously could be perceived as lacking a strictly controlled variable setting; this ablation isolates their individual contributions. To maintain a perfectly controlled environment, our topology-weighted prompt diversity regularizer ($\mathcal{L}_{\text{div}}$) is kept active across all four settings below, ensuring that the *only* variable changing is the formulation of the entropy objective ($\mathcal{L}_{\text{ent}}$). As shown in Table 5, applying the modified loss shape alone (Shape-Only) causes a severe representation collapse, with average

accuracy dropping to 43.76%. Conversely, integrating the uncertainty gate with the traditional entropy loss (Uncertainty-Only) stabilizes the representations and substantially reduces the ECE. This empirically demonstrates the synergistic relationship between the two components: the dynamic gate $\boldsymbol{\alpha}(\mathbf{U}_{\mathrm{norm}})$ functions as a critical *gradient attenuator*. Rather than acting as a hard stop, it smoothly down-weights the magnitude of the entropy penalty for highly ambiguous samples. This soft attenuation suppresses the aggressive gradient oscillations that would otherwise be induced by the quadratic target, allowing the shape $\boldsymbol{\phi}$ to safely refine the predictive boundaries without distorting the underlying prompt topology.

**Table 5:** Ablation decoupling the loss shape from the uncertainty modeling. Averaged across 6 datasets (DTD, Pets, Caltech, UCF, Flowers, Aircraft). The diversity regularizer $\mathcal{L}_{\mathrm{div}}$ is active in all settings to ensure a strictly controlled environment (A detailed dataset-level analysis is provided in Appendix Y).

| Setting | Entropy Objective ($\mathcal{L}_{\mathbf{ent}}$) | Avg. Acc ($\uparrow$) | Avg. ECE ($\downarrow$) |
|---|---|---|---|
| Traditional Loss (Base) | $\mathbf{H}(p)$ | 64.88 | 8.30 |
| Shape-Only | $(\mathbf{H}(p) - \boldsymbol{\phi}(\hat{p}_{\mathrm{max}}))^2$ | 43.76 | 6.18 |
| Uncertainty-Only | $\boldsymbol{\alpha}(\mathbf{U}_{\mathrm{norm}}) \cdot \mathbf{H}(p)$ | 63.19 | 3.58 |
| Full UC-TPT | $\boldsymbol{\alpha}(\mathbf{U}_{\mathrm{norm}}) \cdot (\mathbf{H}(p) - \boldsymbol{\phi}(\hat{p}_{\mathrm{max}}))^2$ | **63.60** | **3.28** |

**(v) Preventing Catastrophic Overconfidence.** While aggregate ECE improvements demonstrate overall calibration gains, the true danger of domain shift lies in *confidently wrong* predictions. To evaluate whether our framework successfully prevents logit inflation on these dangerous samples, we isolate top-1 incorrect predictions with high initial confidence ($\hat{p}_{\mathrm{max}} > 0.8$). As shown in Table 6, traditional baselines like O-TPT often exacerbate this issue, worsening up to 69.6% of these errors by uniformly sharpening them. UC-TPT effectively interrupts this cycle. By correctly identifying local fragility, the uncertainty gate heavily attenuates the learning signal on EuroSAT, allowing the model to soften 100% of these severe errors and reducing the subset ECE by 27.16 points compared to O-TPT. This empirically validates that our dynamic gate successfully acts as a targeted *gradient attenuator*, safely scaling down destructive updates exactly when the model is most vulnerable.

**Table 6: Diagnostic on confidently wrong samples.** Evaluated on top-1 incorrect predictions with pre-adaptation confidence $\hat{p}_{\mathrm{max}} > 0.8$. Standard entropy minimization baselines (O-TPT, A-TPT) blindly over-sharpen these incorrect predictions, exacerbating miscalibration. In contrast, UC-TPT dynamically detects predictive fragility via the gating coefficient $\boldsymbol{\alpha}(\mathbf{U}_{\mathrm{norm}})$, suppressing harmful updates and drastically improving subset calibration.

| Dataset | Method | Pre-Adapt Conf. | Post-Adapt Conf. ($\downarrow$) | Mean Gate $\boldsymbol{\alpha}(\mathbf{U}_{\mathrm{norm}})$ | % Sharpened (Worsened $\downarrow$) | % Softened (Improved $\uparrow$) | Subset ECE ($\downarrow$) |
|---|---|---|---|---|---|---|---|
| | O-TPT | 0.89 | 0.90 | 1.000 | 65.30% | 34.70% | 89.70 |
| ImageNet-A | A-TPT | 0.89 | 0.89 | 1.000 | 54.80% | 45.20% | 88.90 |
| | **UC-TPT (Ours)** | **0.89** | **0.85** | **0.018** | **25.60%** | **74.40%** | **84.80** |
| | O-TPT | 0.87 | 0.84 | 1.000 | 69.60% | 30.40% | 83.96 |
| EuroSAT | A-TPT | 0.87 | 0.88 | 1.000 | 53.30% | 46.70% | 87.70 |
| | **UC-TPT (Ours)** | **0.87** | **0.57** | **0.013** | **0.00%** | **100.00%** | **56.80** |

# 6 Takeaways

We introduced UC-TPT, an **integrated test-time prompt tuning recipe** designed to improve VLM reliability under distribution shifts. UC-TPT synthesizes visual-to-textual conditioning with a unified uncertainty metric—combining alignment confidence, prediction variance, and prompt sensitivity—to enable selective, risk-aware updates. This dynamic gating safely sharpens reliable predictions while actively attenuating updates for ambiguous samples. Concurrently, a topology-weighted diversity regularizer preserves CLIP's foundational semantic geometry. Across diverse benchmarks, UC-TPT consistently anchors the high-reliability end of the Accuracy-ECE Pareto frontier. By delivering substantial calibration improvements alongside a highly efficient, marginal trade-off in peak accuracy, UC-TPT provides a robust and mathematically sound foundation for safe, real-world multimodal adaptation.

**Broader Impact Statement**

This paper presents work whose goal is to advance the field of Machine Learning. The proposed methods aim to improve the robustness and calibration of test-time adaptation in vision–language models. While more reliable adaptive inference can support safer deployment of machine learning systems under distribution shift, the techniques introduced in this work do not introduce new ethical risks beyond those commonly associated with pretrained models. We do not foresee immediate negative societal consequences specific to this work.

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

# A  Appendix

In this Appendix, we provide the following components to support and extend the main paper:

1. Details of Visual-to-Textual Injection Module and design choice (Sec. B).

2. Full details of the datasets used for evaluation of UC-TPT and comparative methods (Sec. C).

3. Full implementation and experimental protocols of UC-TPT, Details of baseline methods and their implementation details for reproducibility (Sec. D).

4. Formal definitions, derivations, and complete results for all calibration metrics (Sec. E).

5. Details on the uncertainty normalization strategy (Sec. F).

6. Details on the use of Hutchinson's estimator for Jacobian-based uncertainty (Sec. G).

7. Analysis and breakdown of the computational overhead introduced by UC-TPT (Sec. H).

8. Theoretical insight of safe sharpening as inverse-variance weighting (Sec. I).

9. Interpretive analysis on Topology-Aware Prompt Geometry regularizer (Sec. J).

10. Analysis on selection of key hyperparameters used in our study (Sec. K).

11. Additional and full detailed results across datasets (Sec. L).

12. Extended evaluation on natural distribution shift datasets (Sec. M).

13. Test-time prompt tuning results using different prompt-learning backbones (Sec. N).

14. Dataset-level reliability and robustness analysis and comparison with baseline methods on challenging benchmarks (Sec. O).

15. Evaluation of different combinations of regularizers of baselines along with our framework (Sec. P).

16. Comparison of different prompt initialization strategies with existing methods (Sec. Q).

17. Full Results on Specialized Domains (RemoteCLIP and BioMedCLIP) (Sec. R).

18. Batch size and adaptation step ablations (Sec. S).

19. Isolation of representational adaptation benefits compared to a post-hoc Temperature Scaling baseline (Sec. T).

20. Qualitative failure analysis demonstrating the mitigation of catastrophic overconfidence on highly ambiguous samples (Sec. U).

21. Empirical validation and statistical stratification analysis of the adaptation risk proxy $U(x)$ (Sec. V).

22. Comprehensive empirical comparison with the concurrent Semantic Orthogonal Calibration (SoC) framework (Sec. W).

23. Accuracy-ECE Pareto frontiers quantifying the optimal trade-off between robust calibration and adaptation accuracy (Sec. X).

24. Dataset-level ablation decoupling the synergistic effects of the uncertainty gate and the shaped entropy objective (Sec. Y).

# B  Details of Visual-to-Textual Injection Module

This appendix provides implementation details for the visual-to-textual conditioning module introduced in Section 3.1.

**Definition B.1** (Visual Injection Mapper). The visual-to-textual conditioning function $\text{Inject}(\cdot; \boldsymbol{\theta}_{inj})$ is implemented as a lightweight two-layer multilayer perceptron (MLP) that maps first-layer visual features to prompt-level modulation coefficients. Specifically,

$$\text{Inject}(\mathbf{z}^{(1)}) = \mathbf{W}_2 \, \sigma(\mathbf{W}_1 \mathbf{z}^{(1)}), \tag{22}$$

where $\mathbf{z}^{(1)} \in \mathbb{R}^d$ is the first-layer visual embedding, $\mathbf{W}_1 \in \mathbb{R}^{w \times d}$ and $\mathbf{W}_2 \in \mathbb{R}^{m \times w}$ are learnable parameters, $\sigma(\cdot)$ denotes the GELU nonlinearity, $w$ is the text transformer width, and $m$ is the number of prompt tokens.

The output of $\text{Inject}(\cdot)$ is interpreted as a token-wise modulation vector $\mathbf{q_x} \in \mathbb{R}^m$, which is broadcast across the corresponding prompt embeddings and injected into the first layer of the text encoder as defined in Eq. 7. This design introduces no additional depth, attention, or parameter updates beyond the prompt tokens and preserves the frozen CLIP backbone.

*Remark* B.2 (Shallow Injection Design Choice). We restrict visual-to-text conditioning to the first layer of the text encoder. This design is motivated by stability and calibration considerations rather than expressive capacity. Empirically, deeper-layer injection or richer coupling mechanisms (e.g., cross-attention) were observed to introduce larger confidence distortions and increased expected calibration error (ECE), without yielding consistent improvements in accuracy. Early-layer injection allows image-conditioned prompt modulation while largely preserving the pretrained semantic structure of CLIP, aligning with the goal of label-free and calibration-aware test-time adaptation.

# C  Datasets

We conduct a comprehensive empirical evaluation of UC-TPT across a broad and diverse collection of benchmarks, encompassing generic image classification, fine-grained recognition, robustness to distribution shifts, remote sensing, and medical image understanding. This evaluation protocol is designed to rigorously assess the generalization capabilities, robustness, and adaptability of the model across heterogeneous visual domains, semantic granularities, and imaging modalities.

**Generic image classification benchmarks.** We evaluate standard object recognition performance on ImageNet Deng et al. (2009), a large-scale dataset comprising over one million images across 1,000 object categories, and Caltech101 Fei-Fei et al. (2004), which consists of 101 object categories characterized by substantial intra-class variability. These datasets serve as canonical benchmarks for assessing general-purpose visual recognition performance.

**Fine-grained & Cross-domain recognition benchmarks.** Fine-grained and cross-domain generalization is evaluated on a diverse set of datasets designed to capture subtle inter-class variations across multiple visual attributes. DTD Cimpoi et al. (2014) evaluates texture recognition under varying material properties and illumination conditions. FLW Nilsback & Zisserman (2008) focuses on fine-grained flower classification, where class distinctions are often defined by subtle visual cues. Food101 Bossard et al. (2014) assesses recognition of food categories exhibiting significant appearance diversity. UCF101 Soomro et al. (2012) is employed to evaluate action recognition from still frames extracted from video sequences. We further include StanfordCars Maji et al. (2013a) and FGVC-Aircraft Maji et al. (2013b) to assess fine-grained vehicle recognition. OxfordPets Parkhi et al. (2012) evaluates fine-grained animal recognition across cat and dog breeds. Finally, EuroSAT Helber et al. (2019) is included in this suite to evaluate cross-domain adaptation to multi-spectral satellite imagery using standard vision-language backbones.

**Robustness under distribution shifts.** To evaluate robustness to distributional shifts and naturally occurring perturbations, we conduct experiments on several ImageNet variants. ImageNet-A Hendrycks et al. (2021b) contains naturally adversarial examples that pose significant challenges to standard image classifiers. ImageNet-V2 and ImageNet-R Hendrycks et al. (2021a) introduce distribution shifts through changes in

image sources and artistic renditions, respectively. ImageNet-S Wang et al. (2019) evaluates robustness with an emphasis on shape-based recognition through segmentation-derived subsets.

**Remote sensing benchmarks.** For a deeper analysis into specialized aerial and Earth observation imagery, we evaluate our method using the domain-specific RemoteCLIP backbone Liu et al. (2024) across seven remote sensing datasets. This includes high-resolution aerial scene classification benchmarks such as MLRSNet Qi et al. (2020), PatternNet Zhou et al. (2018), and RESISC45 Cheng et al. (2017). We also include AID Xia et al. (2017) and the UC Merced Land Use Dataset (UCM) Yang & Newsam (2010) for complex land-use classification, alongside RSICD Lu et al. (2017) to capture rich spatial and semantic earth-observation structures. Finally, we re-evaluate EuroSAT Helber et al. (2019) within this dedicated remote sensing suite. These datasets present unique challenges, such as arbitrary visual orientations, varying spatial resolutions, and extreme domain shifts.

**Medical image classification benchmarks.** For medical image understanding, we adopt Biomed-CLIP Zhang et al. (2024) and evaluate performance across 11 datasets spanning 10 anatomical regions and 9 imaging modalities, thereby covering a wide range of clinically relevant imaging scenarios. Specifically, computed tomography (CT) imaging is evaluated using CTKidney Islam et al. (2022). Dermatoscopic skin lesion classification is assessed using DermaMNIST Codella et al. (2018); Tschandl et al. (2018). Endoscopic image classification is evaluated on the Kvasir dataset Pogorelov et al. (2017), while retinal disease classification is assessed using fundus images from RETINA Köhler et al. (2013); Porwal et al. (2018). Histopathological image classification is evaluated using LC25000 Borkowski et al. (2019) and CHMNIST Kather et al. (2016), which capture cellular- and tissue-level variations. Magnetic resonance imaging (MRI)–based brain tumor classification is assessed using BTMRI Nickparvar (2021). Optical coherence tomography (OCT)–based retinal imaging is evaluated using OCTMNIST Kermany et al. (2018). Ultrasound-based breast lesion classification is assessed using BUSI Al-Dhabyani et al. (2020). Finally, X-ray imaging is evaluated using COVID-QU-Ex Tahir et al. (2021) and KneeXray Chen (2018), covering thoracic and musculoskeletal imaging tasks, respectively.

## D Implementation details and comparative methods

This section describes the implementation details, optimization settings, and comparative evaluation protocols used in our experiments, with the goal of ensuring clarity, fairness, and reproducibility.

We adopt CLIP models with both ResNet-50 (CLIP-RN50) and Vision Transformer (CLIP ViT-B/16) backbones as the foundational architectures. Test-time prompt tuning (TPT) Shu et al. (2022) serves as the primary baseline framework, upon which all comparative methods are implemented. Additional experiments involving prompt-learning–based initializations, such as CoOp Zhou et al. (2022) and MaPLe Khattak et al. (2023).

All baseline calibration methods—including C-TPT Yoon et al. (2024), O-TPT Sharifdeen et al. (2025), and A-TPT Ahamed et al. (2026)—are implemented on top of the TPT framework, following the configurations and design choices reported in their respective publications. For prompt optimization, all methods employ a single-step update using the AdamW optimizer Loshchilov & Hutter (2019) with a learning rate of $5 \times 10^{-3}$ and a batch size of 64, comprising the original image and 63 augmented images. Following prior TPT-based methods, a batch size of 64 is used solely for stable entropy minimization during optimization; predictions remain label-free and instance-wise. Prompt embeddings are initialized as hard prompts (e.g., `"a photo of a <class>"`) for all methods, consistent with the setup in C-TPT and related works.

**Optimization Dynamics and Stop-Gradient Protocol.** While the adapted context embeddings (prompts) are the sole learnable parameters in UC-TPT, it is critical to define the gradient flow through the uncertainty and gating mechanisms. To ensure stable optimization and preserve our heteroscedastic inverse-variance formulation, the uncertainty components ($u_{\text{align}}$, $u_{\text{vis-var}}$, $u_{\text{jac}}$), the dynamic gate ($\alpha$), and the confidence-shaped target ($\phi$) are treated strictly as **stop-gradient quantities** during the backward pass.

If gradients were permitted to flow backward through the computation of the $\boldsymbol{\alpha}$ gate, the optimizer could exploit a degenerate learning path: it could artificially maximize the model's internal uncertainty to force the gate weight to zero ($\boldsymbol{\alpha} \to 0$), thereby trivially minimizing the total loss without learning meaningful visual-semantic representations.

To prevent this, these components are computed dynamically at each step and then explicitly detached from the computational graph. Consequently, the prompts are optimized exclusively through the gradients of the shaped entropy loss, $\nabla_{\text{prompt}}(\mathbf{H}(p) - \boldsymbol{\phi})^2$, which is then statically scaled by the detached $\boldsymbol{\alpha}$ coefficient. This explicitly enforces the role of $\boldsymbol{\alpha}$ as a safe, targeted gradient attenuator, ensuring that the variance estimate dictates the update magnitude without becoming an adversarial optimization target itself.

Regularization hyperparameters are set in accordance with the original implementations of the respective methods. Specifically, C-TPT fixes the regularization weight to $\lambda = 50$ across all experiments. O-TPT sets $\lambda = 18$ for standard experimental settings and reduces it to $\lambda = 2$ when evaluating natural domain shift scenarios. A-TPT employs $\lambda = 80$ for standard experiments and $\lambda = 10$ for natural domain shift evaluations. For our method, we follow A-TPT and set $\lambda$ in Eq. 21 to 80 in all standard experiments. The $K$ in Eq. 10 is set to 5, The $k_0$ and $u_0$ in Eq. 17 is set to 10 and 0.01 respectively. For computing the visual-variance uncertainty $\mathbf{U}_{\text{vis-var}}(\mathbf{x})$ (Eq. 12), we inject isotropic Gaussian noise with fixed standard deviation $\sigma = 0.01$ into $\ell_2$-normalized visual embeddings. This choice induces small local perturbations on the feature manifold and is kept constant across all datasets and methods. The stability constant $\nu$ in Eq.15 is set to 0.1. The parameter $\tilde{C}$ in section 3.2, which determines the subset of classes contributing to the softmax density mass, is chosen in a way that more than 99.9% cumulative probability mass for each dataset, as this proportion consistently captures the majority of the probability mass.

For comparative evaluation, we benchmark our approach against CLIP Radford et al. (2021b), TPT Shu et al. (2022), C-TPT Yoon et al. (2024), O-TPT Sharifdeen et al. (2025), A-TPT Ahamed et al. (2026), and R-TPT Sheng et al. (2025). To ensure fairness and reproducibility, all methods are re-implemented and evaluated under reported experimental settings.

Across all experiments, we report classification accuracy and Expected Calibration Error (ECE) as the primary evaluation metrics. All experiments are conducted on a single NVIDIA RTX Pro 6000 GPU.

## E  Calibration Metrics

**Notation.** Let $n$ be the total number of samples, $C$ the number of classes, and $B$ the number of confidence bins. $\mathbf{S}_b$ denotes the set of samples falling into bin $b$, and $\mathbf{S}_b^{(c)}$ the samples of class $c$ in bin $b$. $n_c = |\{i : y_i = c\}|$ is the number of samples of class $c$. $\text{acc}(\cdot)$ and $\text{conf}(\cdot)$ denote empirical accuracy and mean confidence in a set, respectively. $p_{i,c}$ is the predicted probability for class $c$ on sample $i$, and $p_{i,y_i}$ the probability assigned to the true class. $\mathbb{1}[\cdot]$ is the indicator function.

**Expected Calibration Error (ECE).** ECE measures the discrepancy between predicted confidence and empirical accuracy over fixed bins:

$$\text{ECE} = \sum_{b=1}^{B} \frac{|\mathbf{S}_b|}{n} \left| \text{acc}(\mathbf{S}_b) - \text{conf}(\mathbf{S}_b) \right|. \tag{23}$$

**Static Calibration Error (SCE).** We use a simplified, class-agnostic SCE computed over $B$ fixed (equal-width) confidence bins. Let $\mathbf{S}_b$ be the set of samples falling in bin $b$, $\text{acc}(\mathbf{S}_b)$ the empirical accuracy in that bin, and $\text{conf}(\mathbf{S}_b)$ the average confidence in the bin. Then

$$\text{SCE} = \frac{1}{B} \sum_{b=1}^{B} \left| \text{acc}(\mathbf{S}_b) - \text{conf}(\mathbf{S}_b) \right|. \tag{24}$$

**Maximum Calibration Error (MCE).** MCE reports the worst-case calibration gap across bins:

$$\text{MCE} = \max_{b \in \{1,\dots,B\}} \left| \text{acc}(\mathbf{S}_b) - \text{conf}(\mathbf{S}_b) \right|. \tag{25}$$

**Table 7: Calibration metrics across datasets** for OTPT, CTPT, ATPT, and our method on CLIP ViT-B/16. Lower is better (↓).

| Method | Metric | DTD | FLW | Food | Aircraft | Pets | Caltech | UCF | EuroSAT | Cars | INet | Avg |
|---|---|---|---|---|---|---|---|---|---|---|---|---|
| O-TPT | ECE | 8.08 | 3.87 | 4.63 | 3.97 | 1.96 | 4.64 | 2.28 | 13.80 | 1.61 | 1.98 | 4.682 |
| | SCE | 8.26 | 5.45 | 5.88 | 5.58 | 5.22 | 18.95 | 2.68 | 12.34 | 2.56 | 2.98 | 6.990 |
| | MCE | 19.48 | 13.43 | 11.13 | 13.23 | 25.13 | 85.92 | 11.01 | 29.17 | 13.00 | 9.18 | 23.068 |
| | ACE | 7.86 | 4.48 | 4.85 | 4.46 | 2.00 | 4.73 | 2.32 | 13.80 | 1.90 | 2.38 | 4.878 |
| | Brier | 0.703 | 0.413 | 0.260 | 0.862 | 0.174 | 0.114 | 0.480 | 0.760 | 0.460 | 0.450 | 0.468 |
| | NLL | 2.20 | 1.65 | 0.69 | 3.00 | 0.40 | 0.25 | 1.33 | 1.85 | 1.03 | 1.23 | 1.363 |
| C-TPT | ECE | 12.45 | 5.13 | 3.72 | 4.33 | 1.83 | 4.34 | 2.40 | 13.25 | 1.56 | 3.19 | 5.220 |
| | SCE | 11.67 | 7.13 | 4.62 | 5.41 | 5.19 | 18.55 | 3.38 | 10.75 | 2.73 | 3.51 | 7.294 |
| | MCE | 25.45 | 17.26 | 8.86 | 12.17 | 24.72 | 82.18 | 13.16 | 28.79 | 13.49 | 6.36 | 23.244 |
| | ACE | 12.40 | 5.33 | 3.90 | 4.66 | 1.73 | 4.48 | 2.12 | 13.26 | 1.71 | 3.16 | 5.275 |
| | Brier | 0.720 | 0.440 | 0.250 | 0.860 | 0.173 | 0.110 | 0.460 | 0.750 | 0.450 | 0.440 | 0.465 |
| | NLL | 2.26 | 1.68 | 0.68 | 3.01 | 0.41 | 0.26 | 1.28 | 1.83 | 1.01 | 1.23 | 1.365 |
| A-TPT | ECE | 6.15 | 4.19 | 7.08 | 6.32 | 5.85 | 10.1 | 3.63 | 4.46 | 5.52 | 2.26 | 5.550 |
| | SCE | 9.13 | 6.53 | 4.65 | 7.46 | 5.65 | 22.59 | 2.75 | 12.92 | 2.25 | 3.44 | 7.737 |
| | MCE | 32.31 | 16.18 | 9.38 | 18.11 | 24.70 | 86.93 | 7.44 | 30.61 | 7.80 | 8.17 | 24.163 |
| | ACE | 9.04 | 4.51 | 3.67 | 7.54 | 1.89 | 5.55 | 2.00 | 17.51 | 1.76 | 2.46 | 5.593 |
| | Brier | 0.710 | 0.410 | 0.250 | 0.870 | 0.178 | 0.120 | 0.480 | 0.800 | 0.460 | 0.450 | 0.473 |
| | NLL | 2.23 | 1.67 | 0.69 | 3.05 | 0.43 | 0.26 | 1.34 | 1.91 | 1.03 | 1.24 | 1.385 |
| UC-TPT (Ours) | ECE | 4.74 | 3.97 | 3.23 | 2.65 | 2.65 | 2.96 | 2.70 | 8.36 | 1.35 | 1.53 | 3.414 |
| | SCE | 4.65 | 5.60 | 4.88 | 6.31 | 5.52 | 17.49 | 3.36 | 7.32 | 2.40 | 2.25 | 5.978 |
| | MCE | 13.81 | 18.35 | 8.94 | 13.92 | 18.71 | 85.62 | 11.17 | 23.16 | 12.38 | 8.15 | 21.421 |
| | ACE | 5.18 | 4.74 | 3.61 | 4.24 | 2.66 | 3.30 | 2.55 | 9.25 | 1.72 | 1.78 | 3.903 |
| | Brier | 0.690 | 0.420 | 0.240 | 0.860 | 0.170 | 0.110 | 0.470 | 0.750 | 0.450 | 0.450 | 0.461 |
| | NLL | 2.19 | 1.61 | 0.67 | 3.00 | 0.38 | 0.23 | 1.32 | 1.76 | 1.03 | 1.24 | 1.343 |

**Adaptive Calibration Error (ACE).** ACE uses adaptive bins with equal sample counts ($|\mathbf{S}_b| = n/B$):

$$\text{ACE} = \sum_{b=1}^{B} \frac{|\mathbf{S}_b|}{n} \left| \text{acc}(\mathbf{S}_b) - \text{conf}(\mathbf{S}_b) \right|. \tag{26}$$

**Brier Score.** A mean squared error between predicted probabilities and one-hot labels:

$$\text{BS} = \frac{1}{n} \sum_{i=1}^{n} \sum_{c=1}^{C} \left( p_{i,c} - \mathbb{1}[y_i = c] \right)^2. \tag{27}$$

**Negative Log-Likelihood (NLL).** A likelihood-based calibration measure penalizing confident wrong predictions:

$$\text{NLL} = -\frac{1}{n} \sum_{i=1}^{n} \log p_{i,y_i}. \tag{28}$$

We presented the results of all the calibration metrics across datasets in Table 7. Metrics such as ECE, SCE, MCE, and ACE are reported in percentage form, whereas Brier Score and NLL are presented in probabilistic values. We can clearly see that UC-TPT (Ours) consistently performs better across all metrics compared to Sharifdeen et al. (2025); Ahamed et al. (2026); Yoon et al. (2024).

## F Composite Uncertainty Aggregation and Normalization

Recall that UC-TPT combines three complementary per-sample uncertainty signals—image–text alignment, perturbation variance, and Jacobian-based prompt sensitivity—into a unified scalar score:

$$\mathbf{U}(\mathbf{x}) = \mathbf{U}_{\text{align}}(\mathbf{x}) + \mathbf{U}_{\text{vis-var}}(\mathbf{x}) + \mathbf{U}_{\text{jac}}(\mathbf{x}), \tag{29}$$

where $\mathbf{U}_{\text{align}}(\mathbf{x})$, $\mathbf{U}_{\text{vis-var}}(\mathbf{x})$, and $\mathbf{U}_{\text{jac}}(\mathbf{x})$ are defined in Eqs. (9)–(13) of the main paper. These terms have heterogeneous ranges and units: $\mathbf{U}_{\text{align}}$ is derived from cosine similarities, $\mathbf{U}_{\text{vis-var}}$ from prediction variance

under perturbations, and $\mathbf{U}_{\text{jac}}$ from Jacobian norms. Directly feeding $\mathbf{U}(\mathbf{x})$ into the gating function $\alpha(\cdot)$ in Eq. 17 would make the update strength sensitive to arbitrary scale differences between these components and to dataset- or batch-specific magnitudes.

**Justification for additive fusion.** We intentionally adopt a direct, unweighted additive fusion for Eq. 29 rather than multiplicative or weighted alternatives. Conceptually, because $\mathbf{U}_{\text{align}}$, $\mathbf{U}_{\text{vis-var}}$, and $\mathbf{U}_{\text{jac}}$ track independent failure modes, direct summation acts as a continuous logical "OR" gate. For instance, if a sample exhibits a massive semantic gap but has perfectly stable visual features, the total uncertainty must still spike to protect the model from a harmful update. A multiplicative approach would dangerously suppress this warning if any single term approached zero.

To empirically validate this design, we compared our unweighted additive formulation against two intuitive alternatives:

- **Weighted Fusion:** Prioritizes specific risks by assigning uneven weights (e.g., emphasizing semantic risk via $\mathbf{U}_{\text{w}} = 0.6\mathbf{U}_{\text{align}} + 0.2\mathbf{U}_{\text{vis-var}} + 0.2\mathbf{U}_{\text{jac}}$).

- **Rank-Based Aggregation:** Sums intra-batch rankings to eliminate sensitivity to extreme outliers: $\mathbf{U}_{\text{rank}} = \text{Rank}(\mathbf{U}_{\text{align}}) + \text{Rank}(\mathbf{U}_{\text{vis-var}}) + \text{Rank}(\mathbf{U}_{\text{jac}})$.

As shown in Table 8, simple unweighted addition remains the most robust validation-free approach. Rank aggregation loses critical absolute magnitude information (which dictates the actual severity of the risk), while weighted fusion requires dataset-specific tuning and generally yields inferior average calibration across varied domains.

To obtain a robust and comparable scale across samples after additive fusion, we normalize the composite uncertainty within each mini-batch $B$ using a batch-wise min–max transformation:

**Table 8: Ablation on uncertainty aggregation strategies.** Average accuracy and ECE across benchmarks using the ViT-B/16 backbone.

| Aggregation Strategy | Avg. Acc ($\uparrow$) | Avg. ECE ($\downarrow$) |
|---|---|---|
| Weighted (0.6 : 0.2 : 0.2) | 63.42 | 3.68 |
| Weighted (0.2 : 0.6 : 0.2) | **63.55** | 3.71 |
| Rank-Based Aggregation | 63.52 | 3.64 |
| **Additive (Ours: 1:1:1)** | 63.51 | **3.62** |

$$\mathbf{U}_{\text{norm}}(\mathbf{x}) = \frac{\mathbf{U}(\mathbf{x}) - \min_{\mathbf{x}' \in B} \mathbf{U}(\mathbf{x}')}{\max_{\mathbf{x}' \in B} \mathbf{U}(\mathbf{x}') - \min_{\mathbf{x}' \in B} \mathbf{U}(\mathbf{x}') + \epsilon}, \quad \mathbf{x} \in B, \tag{30}$$

where $\epsilon > 0$ is a small constant for numerical stability. This ensures that $\mathbf{U}_{\text{norm}}(\mathbf{x}) \in [0, 1]$ for all samples in the batch.

**Why batch-wise normalization?** Our entropy objective uses $\mathbf{U}_{\text{norm}}(\mathbf{x})$ only through the bounded gating function $\alpha(\mathbf{U}_{\text{norm}}(\mathbf{x}))$ in Eq. 17,

$$\alpha\big(\mathbf{U}_{\text{norm}}(\mathbf{x})\big) = \frac{1}{1 + \exp\big(k_0(\mathbf{U}_{\text{norm}}(\mathbf{x}) - u_0)\big)}, \tag{31}$$

and the adaptive loss

$$L_{\text{ent}}(\mathbf{x}) = \alpha\big(\mathbf{U}_{\text{norm}}(\mathbf{x})\big)\big(\mathbf{H}(p(\mathbf{x})) - \phi(\hat{p}_{\max}(\mathbf{x}))\big)^2. \tag{32}$$

In this formulation, only the *relative* ordering of uncertainties within a batch matters: samples with larger $\mathbf{U}(\mathbf{x})$ should receive smaller updates (smaller $\alpha$), while more reliable samples with lower $\mathbf{U}(\mathbf{x})$ should be adapted more aggressively (larger $\alpha$). The

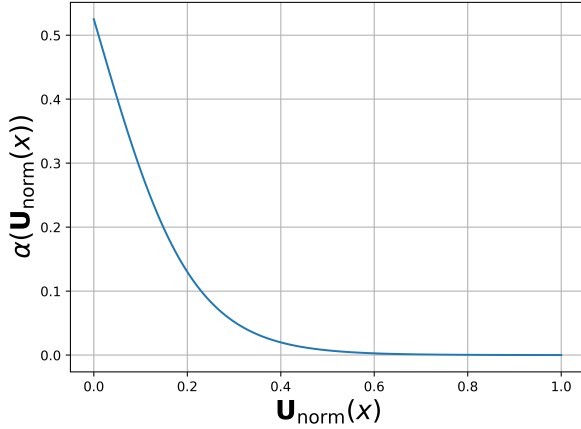

**Figure 9:** Typical behavior of the sigmoid gate $\alpha(\mathbf{U})$ used in our adaptive entropy objective, shown for $k_0 = 10$ and $u_0 = 0.01$ over the normalized uncertainty range $\mathbf{U} \in [0, 1]$.

min–max normalization in Eq. (30) enforces a consistent dynamic range $[0, 1]$ for $\mathbf{U}_{\mathrm{norm}}(\mathbf{x})$ across batches, making the slope parameter $k_0$ and midpoint $u_0$ interpretable and stable across datasets and backbones. The typical behaviour of the gating function $\alpha(u)$ over the normalized range $u \in [0, 1]$ is illustrated in Fig. 9, showing a smooth monotonic decay that naturally suppresses updates for high-uncertainty samples.

Empirically, we observe that although $\mathbf{U}_{\mathrm{align}}(\mathbf{x})$, $\mathbf{U}_{\mathrm{vis\text{-}var}}(\mathbf{x})$, and $\mathbf{U}_{\mathrm{jac}}(\mathbf{x})$ can differ in absolute scale, their *relative* ordering within a batch is informative and fairly consistent across domains. Batch-wise min–max normalization therefore acts as a monotone reparameterization that preserves this ordering while removing arbitrary scale and offset effects.

In fact, on the EuroSAT dataset, we find that the raw combined uncertainty $\mathbf{U}(\mathbf{x})$ has a very large absolute magnitude (e.g., batch-level mean 242.109 and standard deviation 3.28), making its scale highly dataset-dependent and unsuitable for direct gating. After applying batch-wise min–max normalization, the corresponding $\mathbf{U}_{\mathrm{norm}}(\mathbf{x})$ values lie in a stable and interpretable range with mean 0.38 and standard deviation 0.202. Since $\alpha(\cdot)$ is monotonic, this normalization preserves the adaptation ordering while ensuring numerical stability and allowing $(k_0, u_0)$ to behave consistently across datasets.

**Comparison to alternative normalizations.** Beyond alternative aggregation strategies, we also considered other normalization strategies, such as: (i) z-score normalization of $\mathbf{U}(\mathbf{x})$ over the batch and (ii) independent normalization of each component $\mathbf{U}_{\mathrm{align}}$, $\mathbf{U}_{\mathrm{vis\text{-}var}}$, $\mathbf{U}_{\mathrm{jac}}$ before aggregation. In practice, these alternatives either produced unbounded values (requiring additional clipping) or made the gating overly sensitive to outliers in one component. By contrast, the min–max normalization in Eq. (30) guarantees $\mathbf{U}_{\mathrm{norm}}(\mathbf{x}) \in [0, 1]$, keeps the entropy updates bounded, and yields consistently lower ECE across datasets in our experiments.

Finally, we emphasize that the heteroscedastic interpretation of $\mathcal{L}_{\mathrm{ent}}$ (Sec. 3.3.1) is primarily conceptual: $\mathbf{U}_{\mathrm{norm}}(\mathbf{x})$ serves as a surrogate "difficulty" indicator that modulates the effective update strength, analogous to an inverse-variance weight in a heteroscedastic regression objective. Our empirical ablations support that this uncertainty-aware reweighting leads to significantly improved calibration compared to uniform entropy minimization.

## G   Hutchinson Approximation for Jacobian Sensitivity

In UC-TPT, the Jacobian-based uncertainty $\mathbf{U}_{\mathrm{jac}}$ (Eq. 13) captures the sensitivity of the predicted class probabilities $p_c(\mathbf{x})$ to perturbations of the prompt-token matrix $\mathbf{P} \in \mathbb{R}^{m \times d}$. Computing the full Jacobian norm $\left\| \frac{\partial p_c(\mathbf{x})}{\partial \mathbf{P}} \right\|_2^2$ for all classes requires $O(C)$ backpropagations and becomes computationally prohibitive at test time. To avoid this, we adopt a *Hutchinson trace estimator*, providing an unbiased stochastic estimate of the Jacobian norm using a single vector–Jacobian product.

**Full Jacobian Form.** Let the flattened prompt parameter matrix be $\tilde{\mathbf{p}} = \mathrm{vec}(\mathbf{P}) \in \mathbb{R}^{md}$. For any class $c$, its sensitivity is

$$S_c(\mathbf{x}) = \left\| \frac{\partial p_c(\mathbf{x})}{\partial \tilde{\mathbf{p}}} \right\|_2^2 = \mathrm{Tr}\big( J_c(\mathbf{x})^\top J_c(\mathbf{x}) \big), \tag{33}$$

where

$$J_c(\mathbf{x}) = \frac{\partial p_c(\mathbf{x})}{\partial \tilde{\mathbf{p}}}$$

is the class-wise Jacobian with respect to the prompt matrix $\mathbf{P}$.

**Hutchinson Trace Estimator.** For any PSD matrix $A$, the Hutchinson identity is:

$$\mathrm{Tr}(A) = \mathbb{E}_{\mathbf{r}}\big[ \mathbf{r}^\top A \mathbf{r} \big], \qquad \mathbf{r} \sim \mathcal{N}(\mathbf{0}, I) \text{ or Rademacher.} \tag{34}$$

Applying Eq. 34 to $A = J_c^\top J_c$ yields the stochastic estimate:

$$S_c(\mathbf{x}) \approx \mathbf{r}^\top \big( J_c(\mathbf{x})^\top J_c(\mathbf{x}) \big) \mathbf{r} = \left\| J_c(\mathbf{x})\, \mathbf{r} \right\|_2^2. \tag{35}$$

Here, $J_c(\mathbf{x})\,\mathbf{r}$ is evaluated via a single *vector–Jacobian product*, computed with one backward pass in modern autodiff frameworks.

**Top–$\widetilde{C}$ Class Restriction.**  Following Sec. 3.2, the Jacobian term is computed only for the $\widetilde{C}$ most probable classes:

$$\mathbf{U}_{\mathrm{jac}}(\mathbf{x}) = \sum_{c \in \mathrm{Top}\text{-}\widetilde{C}} \big(1 - \cos(z, v_c(\mathbf{x}))\big) \, \|J_c(\mathbf{x})\,\mathbf{r}\|_2^2. \tag{36}$$

The cosine-weighted factor amplifies gradients along semantically misaligned directions, ensuring that $\mathbf{U}_{\mathrm{jac}}(\mathbf{x})$ reflects both representational and semantic fragility.

**Why Top-$\tilde{C}$ Selection Does Not Affect ECE.**  In UC-TPT, the Jacobian-based uncertainty $\mathbf{U}_{\mathrm{jac}}(\mathbf{x})$ serves solely as a normalization factor inside the entropy-based adaptation loss and therefore does not alter the softmax probabilities used for prediction. Since ECE depends exclusively on the predicted label $\hat{y}$ and its corresponding confidence $\mathrm{conf}(\mathbf{x})$ (Eq. 23), restricting the Jacobian computation to the top-$\tilde{C}$ classes cannot influence bin assignments or confidence calibration.

In practice, we select $\tilde{C}$ such that more than 99.9% of the predictive class mass is retained. This choice is motivated by empirical evidence across all datasets showing that the cumulative softmax probability of a small subset of high-confidence classes consistently exceeds 0.99. As a result, the remaining low-probability classes contribute negligibly to gradient-based sensitivity, while restricting computation to the top-$\tilde{C}$ classes reduces the Jacobian cost by over $5\times$ without affecting predictions or ECE.

We further empirically validate this design on the EuroSAT dataset. When computing $\mathbf{U}_{\mathrm{jac}}(\mathbf{x})$ using all classes (full Jacobian), the batch-wise min–max normalized uncertainty has a mean of 0.38 and a standard deviation of 0.202. Applying the Hutchinson approximation together with the top-$\tilde{C}$ class restriction yields a closely matching distribution (mean 0.39, std. 0.211). This close agreement indicates that the Hutchinson–top-$\tilde{C}$ estimate preserves the relative ordering of uncertain samples within a batch—precisely the quantity utilized by the monotonic gating function $\alpha(\cdot)$—while avoiding the prohibitive cost of full Jacobian computation. Consequently, restricting to the selected top-$\tilde{C}$ classes maintains calibration behavior and leaves ECE unaffected.

## H  Complexity Analysis

We analyze the computational complexity of the proposed Uncertainty-Calibrated Test-Time Prompt Tuning (UC-TPT) framework. Let $N$ denote the batch size, $C$ the total number of classes, $C_B \ll C$ the number of classes active within a batch, $d$ the embedding dimension, and $K$ the number of perturbations or Hutchinson samples.

**Lemma H.1** (Visual Encoding Complexity). *Computing CLIP image embeddings for a batch of $N$ samples incurs a computational cost of $\mathcal{O}(Nd)$.*

**Lemma H.2** (Perturbation-Based Uncertainty Complexity). *Estimating uncertainty via $K$ perturbation-consistent logits over $N$ samples and $C$ classes incurs a cost of $\mathcal{O}(KNC)$.*

**Lemma H.3** (Full Jacobian Trace Complexity). *Explicit computation of the Jacobian trace of a $C$-dimensional logit vector with respect to $d$ prompt parameters over $N$ samples incurs a cost of $\mathcal{O}(NdC)$.*

**Lemma H.4** (Hutchinson Trace Approximation). *Using a Hutchinson estimator with $K$ samples approximates the Jacobian trace with computational cost $\mathcal{O}(KNC)$, eliminating the $\mathcal{O}(NdC)$ term.*

**Lemma H.5** (Batch-Restricted Class Interaction). *Restricting class–class regularization to batch-active classes reduces the complexity from $\mathcal{O}(C^2 d)$ to $\mathcal{O}(C_B^2 d)$.*

**Theorem H.6** (Overall UC-TPT Complexity). *The total computational complexity of UC-TPT is given by:*

- *UC-TPT (Full Jacobian):*

$$\mathcal{C}_{\mathrm{UC\text{-}TPT(FJ)}} = \Theta(NdC + KNC + C_B^2 d + Nd),$$

  *which is dominated by the $\Theta(NdC)$ term when $C$ is large.*

**Table 9: Computational complexity, time, and memory usage of TPT variants** on ImageNet-V2 (ViT-B/16). $C$: total classes, $\tilde{C} \ll C$: selected classes capturing more than 99% predictive mass, $C_B$: active classes in batch, $N$: batch size, $d$: embedding dim, $K$: Hutchinson perturbations. Memory denotes peak GPU memory per batch.

| Method | Complexity | Time (s)/batch | Memory (MiB) | ECE ($\downarrow$) |
|---|---|---|---|---|
| A-TPT Ahamed et al. (2026) | $\mathcal{O}(C^2 d)$ | 0.82 | 21840 | 8.11 |
| O-TPT Sharifdeen et al. (2025) | $\mathcal{O}(C^2 d)$ | 0.90 | 23740 | 4.01 |
| UC-TPT (Full Jacobian) | $\mathcal{O}(Nd) + \mathcal{O}(KNC)$ $+ \mathcal{O}(NdC) + \mathcal{O}(C_B^2 d)$ | 1.55 | 25410 | **3.06** |
| UC-TPT (Hutchinson) | $\mathcal{O}(Nd) + \mathcal{O}(KN\tilde{C})$ $+ \mathcal{O}(C_B^2 d)$ | **1.19** | **22650** | **3.06** |

- **_UC-TPT (Hutchinson):_**

$$\mathcal{C}_{\text{UC-TPT(H)}} = \Theta(KNC + C_B^2 d + Nd),$$

  *which is dominated by $\Theta(KNC)$ under the practical regime $C_B \ll C$ and small $K$.*

*Proof.* The result follows by summing the costs established in Lemmas H.1–H.5. The Hutchinson estimator removes the explicit Jacobian term while preserving an unbiased trace estimate, yielding the stated reduction in asymptotic complexity. □

**Empirical Validation and Deployment Feasibility.** Table 9 reports theoretical complexity alongside empirical runtime and memory usage measured on ImageNet-V2 with a ViT-B/16 backbone. Consistent with Theorem H.6, A-TPT and O-TPT incur a global $\mathcal{O}(C^2 d)$ overhead due to full class–class interactions. For larger class spaces, such as the full ImageNet vocabulary ($C = 1000$), this quadratic scaling becomes a severe latency bottleneck, demanding up to a million interaction computations per adaptation step.

In contrast, UC-TPT restricts regularization to batch-active classes and replaces the Jacobian trace with a Hutchinson approximation, substantially reducing both runtime and memory overhead. By decoupling the computational complexity from the total class count $C$, UC-TPT ensures that memory and inference latency scale gracefully with the batch size ($B \ll C$) rather than the global vocabulary.

As a result, UC-TPT introduces only a modest latency increase while achieving significantly improved calibration, reducing ECE from 4.01/8.11 to 3.06. This explicitly quantifies the operational trade-off: UC-TPT secures its robust calibration profile and Pareto-optimal accuracy with only a fractional, constant-bound latency cost, confirming its feasibility for real-time deployment even in massive-scale recognition tasks.

## I  Theoretical insight (safe sharpening as inverse-variance weighting)

We formalize Eq. 18 as a heteroscedastic regression problem and show that (i) the optimal weight is inverse variance, and (ii) the resulting gradient update is provably attenuated for high-uncertainty samples, yielding safer test-time sharpening.

**Setup.** For each test sample $\mathbf{x}$, define the *sharpening residual*

$$r(\mathbf{x}; \mathbf{P}) = \mathbf{H}(p(\mathbf{x}; \mathbf{P})) - \phi(\hat{p}_{\max}(\mathbf{x}; \mathbf{P})), \tag{37}$$

where $\mathbf{H}(\cdot)$ is predictive entropy (Eq. 15) and $\phi(\cdot)$ is the confidence-shaped target (Eq. 16). We interpret $r(\mathbf{x}; \mathbf{P})$ as a noisy observation of a latent "desired residual" 0 with sample-dependent noise:

$$r(\mathbf{x}; \mathbf{P}) = \epsilon(\mathbf{x}), \qquad \mathbb{E}[\epsilon(\mathbf{x})] = 0, \qquad \text{Var}[\epsilon(\mathbf{x})] = \sigma^2(\mathbf{x}). \tag{38}$$

Under distribution shift, $\sigma^2(\mathbf{x})$ is larger for ambiguous/OOD samples; UC-TPT uses uncertainty $\mathbf{U}(\mathbf{x})$ as a monotone proxy for $\sigma^2(\mathbf{x})$.

**Proposition 1 (ML derivation of inverse-variance weighting).** Assume $\epsilon(\mathbf{x}) \sim \mathcal{N}(0, \sigma^2(\mathbf{x}))$ independent across samples. The negative log-likelihood over a batch $\mathcal{B}$ is

$$\mathcal{L}_{\mathrm{NLL}} = \sum_{\mathbf{x} \in \mathcal{B}} \left( \frac{r(\mathbf{x}; \mathbf{P})^2}{2\sigma^2(\mathbf{x})} + \frac{1}{2} \log \sigma^2(\mathbf{x}) \right) + \text{const.} \tag{39}$$

If $\sigma^2(\mathbf{x})$ is treated as fixed w.r.t. $\mathbf{P}$ during prompt updates (or the log-term is absorbed into a constant), minimizing $\mathcal{L}_{\mathrm{NLL}}$ is equivalent to minimizing the weighted least-squares objective

$$\sum_{\mathbf{x} \in \mathcal{B}} \alpha(\mathbf{x}) \, r(\mathbf{x}; \mathbf{P})^2, \qquad \text{with} \quad \alpha(\mathbf{x}) \propto \frac{1}{\sigma^2(\mathbf{x})}. \tag{40}$$

Thus the statistically optimal weight is inverse variance. In UC-TPT we instantiate $\alpha(\mathbf{x}) = \boldsymbol{\alpha}(\mathbf{U}_{\mathrm{norm}}(\mathbf{x}))$ (Eq. 17), enforcing a monotone mapping from uncertainty to (approx.) inverse variance.

**Proposition 2 (Gauss–Markov / minimum-variance aggregation).** Consider estimating a shared parameter update direction from noisy per-sample residuals with heteroscedastic noise as in Eq. 38. Among all unbiased linear combinations of residual-based gradients, inverse-variance weighting minimizes the estimator variance (classical weighted least squares / Gauss–Markov). Concretely, for scalars $g_i$ with $\mathbb{E}[g_i] = g$ and $\mathrm{Var}(g_i) = \sigma_i^2$, the minimum-variance unbiased estimator of $g$ is

$$\hat{g} = \frac{\sum_i \sigma_i^{-2} g_i}{\sum_i \sigma_i^{-2}}. \tag{41}$$

Hence down-weighting high-uncertainty samples is not heuristic: it is the variance-optimal way to aggregate noisy test-time signals.

**Proposition 3 (safe sharpening via gradient attenuation).** Let the UC-TPT entropy loss for one sample be

$$\mathcal{L}_{\mathrm{ent}}(\mathbf{x}) = \alpha(\mathbf{x}) \, r(\mathbf{x}; \mathbf{P})^2. \tag{42}$$

Its gradient w.r.t. prompt parameters $\mathbf{P}$ is

$$\nabla_{\mathbf{P}} \mathcal{L}_{\mathrm{ent}}(\mathbf{x}) = 2 \, \alpha(\mathbf{x}) \, r(\mathbf{x}; \mathbf{P}) \, \nabla_{\mathbf{P}} r(\mathbf{x}; \mathbf{P}) + r(\mathbf{x}; \mathbf{P})^2 \nabla_{\mathbf{P}} \alpha(\mathbf{x}), \tag{43}$$

where the second term is typically small in practice when $\alpha$ depends on $\mathbf{U}$ through weakly varying proxies or is stop-gradient (either choice is valid). Ignoring $\nabla_{\mathbf{P}} \alpha$ (or treating it as bounded), we obtain the key attenuation property:

$$\left\| \nabla_{\mathbf{P}} \mathcal{L}_{\mathrm{ent}}(\mathbf{x}) \right\| \leq 2 \, \alpha(\mathbf{x}) \, |r(\mathbf{x}; \mathbf{P})| \, \left\| \nabla_{\mathbf{P}} r(\mathbf{x}; \mathbf{P}) \right\|. \tag{44}$$

Since $\alpha(\mathbf{x})$ is monotone decreasing in uncertainty, high-uncertainty samples provably contribute smaller gradient magnitudes, preventing aggressive sharpening driven by unreliable/OOD inputs.

**Corollary (risk-aware sharpening).** Under the heteroscedastic model, $\alpha(\mathbf{x}) \propto 1/\sigma^2(\mathbf{x})$ is the maximum-likelihood (and minimum-variance) choice, and Eq. 44 shows it yields conservative updates for samples with high uncertainty (large $\sigma^2$). Therefore, compared to uniform entropy minimization ($\alpha \equiv 1$), UC-TPT performs *risk-aware* sharpening: only samples that are simultaneously (i) aligned with the target entropy (small residual) and (ii) stable (large $\alpha$) meaningfully drive adaptation, which empirically reduces confident errors and stabilizes confidence dynamics (Fig. 4(a), Fig. 15(a,b)).

## J  Topology-Aware Prompt Geometry: Interpretive Analysis

This section analyzes the geometric structure induced by the proposed topology-aware diversity regularizer in UC-TPT, and its relation to the frozen semantic space of CLIP.

**Frozen CLIP Semantic Geometry.** Let $\mathbf{v}_c^0 \in \mathbb{R}^d$ denote the frozen CLIP text embedding corresponding to class $c$, obtained from the manual prompt `"a photo of a <class>"`. The pretrained class–class cosine similarity is defined as

$$\mathbf{S}_{c'c''}^0 = \cos(\mathbf{v}_{c'}^0, \mathbf{v}_{c''}^0), \qquad -1 \le \mathbf{S}_{c'c''}^0 \le 1, \tag{45}$$

which characterizes the semantic relations encoded by the frozen CLIP text encoder.

**Instance-Conditioned Prompt Geometry.** For a test sample $\mathbf{x}$, UC-TPT produces instance-conditioned prompt embeddings $\hat{\mathbf{v}}_c(\mathbf{x})$ for each class $c$. The adapted similarity between two classes for a given sample is

$$\hat{\mathbf{S}}_{c'c''}(\mathbf{x}) = \cos\big(\hat{\mathbf{v}}_{c'}(\mathbf{x}), \hat{\mathbf{v}}_{c''}(\mathbf{x})\big). \tag{46}$$

The diversity regularizer assigns topology-aware weights

$$\mathbf{w}_{c'c''} = 1 - \mathbf{S}_{c'c''}^0, \tag{47}$$

so that class pairs that are semantically close under frozen CLIP receive weaker repulsion, while distant pairs receive stronger repulsion.

The resulting per-sample diversity objective is

$$\mathcal{L}_{\mathrm{div}}(\mathbf{x}) = \sum_{c'<c''} \mathbf{w}_{c'c''} \, \hat{\mathbf{S}}_{c'c''}(\mathbf{x}), \tag{48}$$

which biases the adapted prompt geometry toward respecting relative semantic relations encoded in $\mathbf{S}^0$.

**Class-Averaged Prompt Structure.** For analysis, we consider the class-averaged adapted prompt

$$\hat{\mathbf{v}}_c = \frac{1}{|\mathcal{X}_c|} \sum_{\mathbf{x} \in \mathcal{X}_c} \hat{\mathbf{v}}_c(\mathbf{x}), \qquad \|\hat{\mathbf{v}}_c\|_2 = 1, \tag{49}$$

and stack them as

$$\hat{\mathbf{V}} = [\hat{\mathbf{v}}_1, \dots, \hat{\mathbf{v}}_C] \in \mathbb{R}^{d \times C}. \tag{50}$$

The corresponding Gram matrix

$$\hat{\mathbf{S}} = \hat{\mathbf{V}}^\top \hat{\mathbf{V}} \tag{51}$$

summarizes the average inter-class geometry induced by adaptation. The deviation $\|\hat{\mathbf{S}} - \mathbf{S}^0\|_F$ serves as a quantitative measure of how much the adapted prompts depart from the frozen CLIP semantic structure.

**Comparison with Other TPT Regularizers.** O-TPT enforces orthogonality among adapted class prompts, while A-TPT promotes uniform angular separation. Both objectives impose geometry that is independent of the pretrained semantic relations encoded in $\mathbf{S}^0$. In contrast, UC-TPT modulates inter-class repulsion according to frozen CLIP similarity, leading to an adapted geometry that more closely reflects the original semantic topology.

**Logit Sensitivity Perspective.** Given a visual feature $\mathbf{z} \in \mathbb{R}^d$, the TPT logit map is

$$g(\mathbf{z}) = \tau \, \mathbf{z}^\top \hat{\mathbf{V}}, \tag{52}$$

where $\tau > 0$ is the temperature. The sensitivity of logits to perturbations in $\mathbf{z}$ is influenced by the spectral norm $\|\hat{\mathbf{V}}\|_2$, which in turn depends on the geometry of the adapted prompts. By discouraging excessive distortion of inter-class relations, the proposed regularizer induces smoother variations in the logit space, aligning with the observed improvements in calibration reported in the empirical results.

# K   Analysis of Hyperparameters

To select our global hyperparameters while strictly preventing domain overfitting, we conducted a heavily constrained grid search. Rather than tuning on full validation splits, we restricted our development phase to an extremely limited **5-shot subset** (5 labeled images per class) sampled from five designated development datasets: DTD, Flowers, Pets, EuroSAT, and Cars.

In Fig. 10, we present the variation in ECE with respect to different model parameters: the gating scalars $k_0$ and $u_0$ (Eq. 17), the calibration weight $\lambda$ (Eq. 21), and $K$ (Eq. 10), shown in Fig. 10(a), (b), (c), and (d), respectively. The plots report the average performance across this restricted 5-shot subset using the CLIP ViT-B/16 backbone. While ablating one parameter, we kept all the remaining parameters fixed at their optimal values to ensure a fair comparison.

From these figures, we observe that the best calibration performance (ECE $\downarrow$) is achieved at $k_0 = 10$, $u_0 = 0.01$, $\lambda = 80$, and $K = 5$. Crucially, once these optimal values were identified via this minimal few-shot subset, **we rigidly locked these parameters for all our analysis**. Consequently, the remaining 9 datasets in our broader evaluation suite (as well as 11 datasets in BioMedCLIP analysis and 7 datasets in RemoteCLIP analysis) were evaluated using these exact fixed values. This ensures our test-time adaptation remains strictly validation-free and completely label-free when encountering novel, out-of-distribution domains.

We performed an ablation study by replacing the sigmoid-based gating function $\alpha(\cdot)$ (Eq. 17) in our uncertainty-aware update rule with several alternative gating mechanisms. The objective of this analysis was to evaluate the effectiveness and robustness of different gates across six datasets—Pets, DTD, Flowers, Aircraft, Caltech, and UCF—using CLIP ViT-B/16 as the backbone. Specifically, we evaluated the following gating functions:

- **Tanh-based gate** — a smoother and symmetric alternative to sigmoid:

$$\alpha(\mathbf{U}) = 0.5 \left( 1 - \tanh \left( k_0 \left( \mathbf{U}_{\text{norm}} - u_0 \right) \right) \right). \tag{53}$$

- **Inverse-square gate** — a polynomial attenuation mechanism:

$$\alpha(\mathbf{U}) = \frac{1}{1 + \left( k_0 (\mathbf{U}_{\text{norm}} - u_0) \right)^2}. \tag{54}$$

- **Exponential decay gate** — a monotonic exponential suppression:

$$\alpha(\mathbf{U}) = e^{-k_0 (\mathbf{U}_{\text{norm}} - u_0)}. \tag{55}$$

As illustrated in Fig. 11, the sigmoid-based gate achieves the best calibration performance (ECE$\downarrow$) on Aircraft, Caltech, and DTD. While it produces slightly higher ECE on Pets, Flowers, and UCF compared to certain alternatives, the **overall trend consistently favors sigmoid gating**. When averaged across all six datasets, the sigmoid gate achieves the **lowest mean ECE of 3.28**, outperforming the Tanh-based gate (3.49), Inverse-square gate (3.52), and Exponential decay gate (3.50).

Given its superior average calibration performance and stable behavior across diverse datasets, we adopt the **sigmoid gating function** as the default choice in our method.

# L   Detailed experimental analysis

In this section, we provide detailed tables for better clarity. Table 10 presents a comparison of evaluation metrics across all baseline methods on all datasets using the CLIP RN50 backbone. We observe that UC-TPT achieves the best (lowest) average ECE ($\downarrow$) while maintaining competitive accuracy compared to the baselines.

The detailed results for natural domain shift on the ImageNet variants are presented in Table 11 for the CLIP-RN50 backbone and Table 12 for the CLIP-ViT B/16 backbone. From both tables, we can observe that UC-TPT, our proposed method, outperforms all other baseline approaches and achieves the best average ECE.

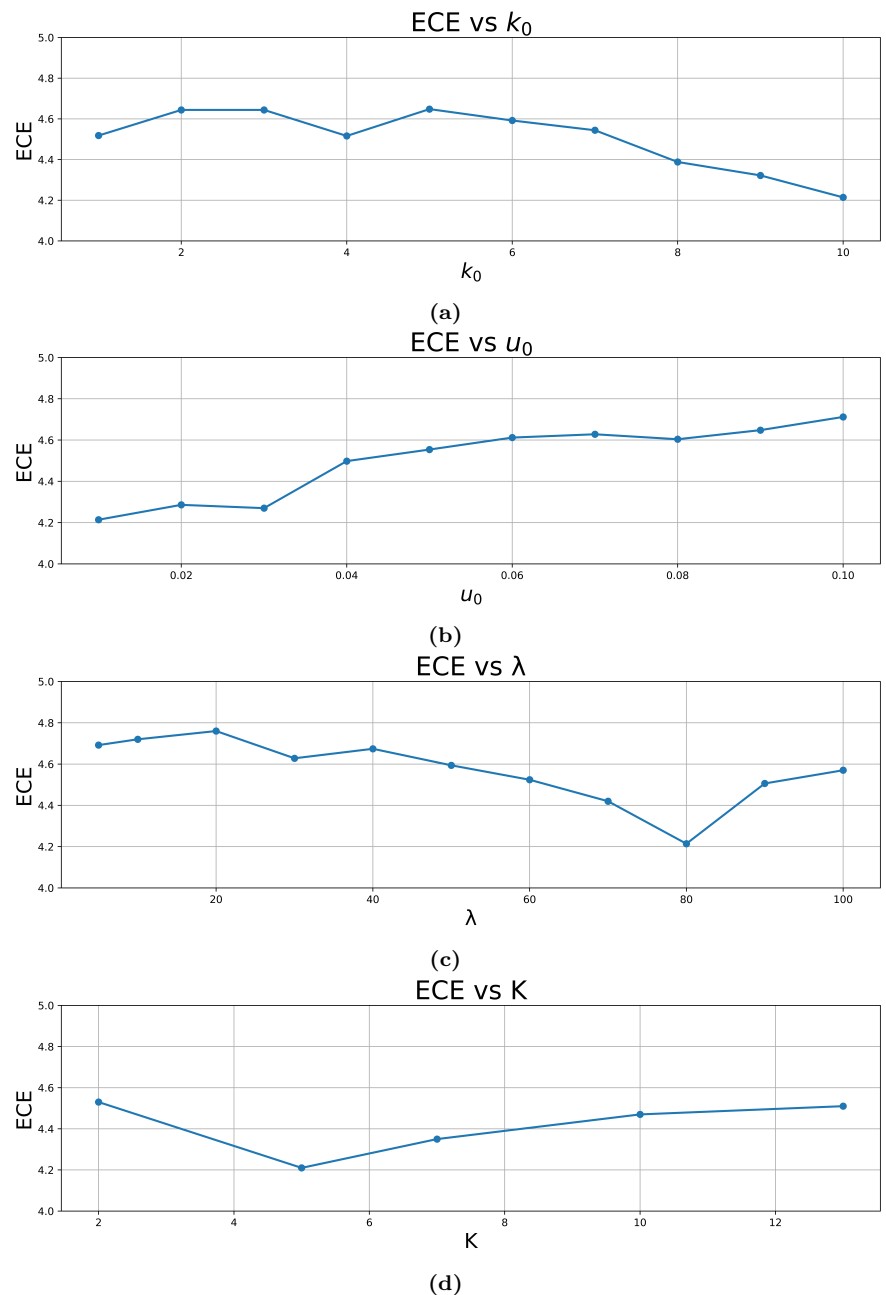

**Figure 10:** Ablations on model parameters with CLIP-ViT B/16 backbone averaged over DTD, Flower, Pets, Eurosat, Stanfordcar datasets (a) ECE vs $k_0$. (b) ECE vs $u_0$. (c) ECE vs $\lambda$. (d) ECE vs $K$.

In Table 13, we present the baseline methods—O-TPT Sharifdeen et al. (2025), C-TPT Yoon et al. (2024), R-TPT Sheng et al. (2025), and A-TPT Ahamed et al. (2026)—with and without our uncertainty modeling. The results clearly show that incorporating uncertainty estimation into test-time prompt tuning is effective not only for our method but also for all baseline approaches. A significant reduction in ECE is consistently observed when uncertainty estimation is applied, demonstrating that it is a powerful tool for achieving better-calibrated and more trustworthy models in test-time prompt tuning.

In Table 14, we report the standard deviation across three random seeds for A-TPT Ahamed et al. (2026), O-TPT Sharifdeen et al. (2025), and UC-TPT (Ours) using the CLIP-ViT B/16 backbone on the DTD, Flower, Food, Caltech, and Cars datasets. We observe that all methods exhibit similar standard deviation

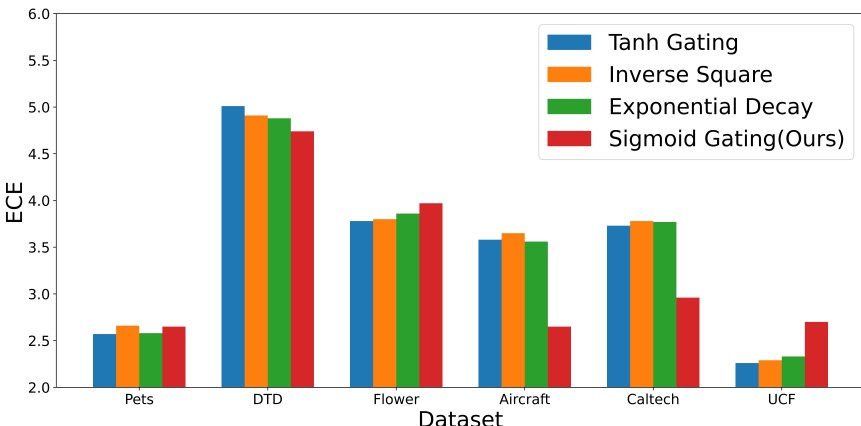

**Figure 11: Comparison of ECE (↓) across datasets** using CLIP-ViT B/16 as the backbone with different gating functions.

**Table 10: Comparison of calibration performance** with CLIP-RN50 backbone. Best average result is in **bold**.

| Method | Metric | ImageNet | DTD | Flowers | Food | Aircraft | Pets | Caltech | UCF | EuroSAT | Car | Avg |
|---|---|---|---|---|---|---|---|---|---|---|---|---|
| Zero-Shot | **Acc.** | 58.20 | 39.95 | 60.94 | 73.80 | 15.66 | 83.73 | 85.88 | 58.42 | 24.21 | 55.68 | 55.65 |
| | **ECE** | 2.08 | 9.66 | 3.12 | 2.48 | 6.31 | 6.01 | 4.22 | 3.03 | 14.42 | 4.62 | 5.60 |
| TPT | **Acc.** | 60.66 | 41.49 | 62.40 | 75.00 | 16.98 | 84.79 | 86.65 | 59.53 | 28.53 | 58.38 | 57.44 |
| | **ECE** | 11.41 | 25.72 | 13.59 | 5.24 | 16.15 | 3.55 | 4.80 | 12.35 | 22.46 | 3.60 | 11.88 |
| R-TPT | **Acc.** | 58.20 | 39.83 | 61.02 | 75.76 | 15.63 | 83.65 | 85.80 | 58.37 | 24.20 | 55.68 | 55.81 |
| | **ECE** | 11.48 | 24.83 | 12.36 | 5.20 | 12.37 | 4.04 | 4.94 | 9.64 | 30.93 | 1.33 | 11.71 |
| C-TPT | **Acc.** | 60.00 | 42.20 | 65.08 | 74.67 | 16.86 | 83.67 | 86.73 | 59.56 | 27.57 | 56.14 | 57.25 |
| | **ECE** | 3.02 | 20.75 | 4.04 | 1.86 | 10.99 | 2.67 | 2.14 | 4.05 | 15.19 | 1.95 | 6.66 |
| A-TPT | **Acc.** | 58.2 | 40.7 | 61.2 | 72.4 | 15.7 | 83.3 | 84.5 | 57.7 | 22.6 | 54.3 | 55.1 |
| | **ECE** | 2.46 | 6.05 | 6.02 | 7.46 | 7.47 | 7.20 | 7.09 | 3.10 | 16.02 | 12.16 | 7.50 |
| O-TPT | **Acc.** | 58.76 | 41.55 | 65.65 | 74.54 | 16.77 | 83.21 | 86.65 | 59.00 | 27.80 | 55.29 | 56.92 |
| | **ECE** | 3.19 | 16.70 | 2.64 | 1.13 | 8.42 | 3.15 | 3.35 | 2.21 | 15.72 | 1.88 | 5.84 |
| **UC-TPT (Ours)** | **Acc.** | 58.19 | 41.62 | 65.88 | 74.31 | 16.95 | 82.31 | 87.00 | 58.39 | 28.30 | 55.15 | 56.81 |
| | **ECE** | 2.07 | 11.62 | 3.05 | 1.22 | 7.23 | 3.74 | 2.76 | 2.71 | 11.33 | 1.85 | **4.76** |

**Table 11: Calibration performance** on the ImageNet suite (CLIP-RN50) for natural distribution shifts. Best average result is in **bold**.

| Method | Metric | I-A | I-V2 | I-R | I-S | Avg |
|---|---|---|---|---|---|---|
| CLIP-RN50 | **Acc.** | 21.69 | 51.44 | 55.94 | 33.33 | 40.60 |
| | **ECE** | 21.30 | 3.35 | 1.95 | 3.14 | 7.43 |
| TPT | **Acc.** | 25.17 | 54.58 | 59.10 | 35.26 | 43.53 |
| | **ECE** | 31.04 | 13.18 | 9.12 | 13.67 | 16.75 |
| R-TPT | **Acc.** | 21.64 | 51.51 | 55.95 | 33.30 | 40.60 |
| | **ECE** | 29.98 | 13.54 | 9.79 | 12.99 | 16.57 |
| C-TPT | **Acc.** | 22.20 | 53.37 | 56.87 | 34.34 | 41.70 |
| | **ECE** | 22.78 | 5.09 | 1.34 | 6.46 | 8.92 |
| A-TPT | **Acc.** | 23.74 | 54.50 | 58.42 | 35.09 | 42.94 |
| | **ECE** | 27.87 | 11.08 | 6.20 | 11.74 | 14.22 |
| O-TPT | **Acc.** | 22.57 | 53.15 | 57.11 | 34.07 | 41.72 |
| | **ECE** | 24.25 | 3.82 | 2.58 | 5.06 | 8.93 |
| **UC-TPT (Ours)** | **Acc.** | 24.80 | 52.81 | 55.97 | 33.92 | 41.88 |
| | **ECE** | 18.25 | 3.19 | 1.85 | 4.23 | **6.88** |

**Table 12: Calibration performance** on the ImageNet suite (CLIP-ViT B/16) for natural distribution shifts. Best average result is in **bold**.

| Method | Metric | I-A | I-V2 | I-R | I-S | Avg |
|---|---|---|---|---|---|---|
| CLIP-ViT B/16 | **Acc.** | 47.73 | 60.79 | 74.01 | 46.12 | 57.16 |
| | **ECE** | 8.40 | 2.79 | 3.59 | 4.84 | 4.90 |
| TPT | **Acc.** | 52.85 | 63.12 | 76.91 | 47.93 | 60.20 |
| | **ECE** | 16.37 | 11.14 | 4.36 | 14.94 | 11.70 |
| R-TPT | **Acc.** | 47.75 | 60.75 | 73.98 | 46.18 | 57.16 |
| | **ECE** | 13.95 | 11.52 | 4.58 | 14.02 | 11.02 |
| C-TPT | **Acc.** | 49.37 | 61.99 | 74.82 | 47.29 | 58.37 |
| | **ECE** | 6.45 | 4.51 | 2.87 | 7.23 | 5.26 |
| A-TPT | **Acc.** | 51.02 | 62.46 | 76.20 | 47.68 | 59.34 |
| | **ECE** | 10.23 | 8.11 | 1.91 | 11.58 | 7.95 |
| O-TPT | **Acc.** | 49.94 | 61.69 | 75.26 | 47.08 | 58.49 |
| | **ECE** | 7.14 | 4.01 | 2.15 | 6.93 | 5.06 |
| **UC-TPT (Ours)** | **Acc.** | 47.82 | 60.89 | 73.48 | 46.46 | 57.16 |
| | **ECE** | 7.11 | 3.06 | 2.08 | 6.28 | **4.63** |

**Table 13:** ECE comparison across different Test-Time Prompt Tuning methods with and without uncertainty modeling.

| Method | Metric | DTD | FLW | Food | Air. | Pets | C101 | UCF | SAT | Cars | Avg. |
|---|---|---|---|---|---|---|---|---|---|---|---|
| O-TPT | ECE | 8.08 | 3.87 | 4.63 | 3.97 | 1.96 | 4.64 | 2.28 | 13.80 | 1.61 | 4.98 |
| O-TPT+Uncertainty | ECE | 4.74 | 3.66 | 4.79 | 3.45 | 2.41 | 4.22 | 2.36 | 10.55 | 1.48 | **4.18** |
| C-TPT | ECE | 12.45 | 5.13 | 3.72 | 4.33 | 1.83 | 4.34 | 2.40 | 13.25 | 1.56 | 5.45 |
| C-TPT+Uncertainty | ECE | 5.39 | 4.01 | 5.24 | 3.74 | 2.56 | 3.70 | 2.85 | 12.49 | 1.64 | **4.62** |
| R-TPT | ECE | 18.79 | 10.84 | 3.30 | 12.65 | 5.40 | 3.60 | 12.07 | 22.02 | 1.93 | 10.07 |
| R-TPT+Uncertainty | ECE | 3.72 | 5.34 | 4.50 | 2.55 | 5.51 | 7.19 | 3.70 | 9.57 | 10.11 | **5.80** |
| A-TPT | ECE | 6.15 | 4.19 | 7.08 | 6.32 | 5.85 | 10.10 | 3.63 | 4.46 | 5.52 | 5.92 |
| A-TPT+Uncertainty | ECE | 5.08 | 4.34 | 3.57 | 4.77 | 2.48 | 4.59 | 2.51 | 15.03 | 1.43 | **4.87** |

**Table 14: Standard deviation across 3 random seeds** for A-TPT, O-TPT, and UC-TPT (Ours) with the CLIP-ViT B/16 backbone. Lower is better ($\downarrow$).

| Method | Metric | DTD | Flower | Food | Caltech | Car | Avg |
|---|---|---|---|---|---|---|---|
| A-TPT | **ACC** | 0.3014 | 0.0732 | 0.0516 | 0.1189 | 0.1020 | 0.12942 |
| | **ECE** | 0.2459 | 0.1103 | 0.0941 | 0.1293 | 0.1944 | 0.15480 |
| O-TPT | **ACC** | 0.1514 | 0.1529 | 0.0416 | 0.1189 | 0.1205 | 0.11706 |
| | **ECE** | 0.0983 | 0.1980 | 0.0163 | 0.2170 | 0.1702 | 0.13996 |
| **UC-TPT (Ours)** | **ACC** | 0.1316 | 0.1541 | 0.1225 | 0.1537 | 0.0612 | 0.12462 |
| | **ECE** | 0.0902 | 0.1287 | 0.0300 | 0.2614 | 0.1225 | **0.12656** |

values. Compared to the others, UC-TPT shows a slightly lower average ECE standard deviation, indicating more stable and reproducible results.

## M Domain shift analysis

For further analysis in domain shifts, we choose PACS Li et al. (2017) and DomainNet Peng et al. (2019) datasets.

The PACS Li et al. (2017) dataset is a widely used benchmark for domain-shift and domain-generalization studies. It contains four distinct domains—**Photo, Art Painting, Cartoon, and Sketch**—that exhibit large variations in texture, style, and abstraction. Despite sharing the same set of object categories, the

visual appearance differs drastically across domains, making PACS an effective testbed for evaluating a model's robustness to distribution shifts. Its diverse domain composition helps assess how well a method can generalize from one visual style to another, especially in real-world scenarios where models often encounter unseen or shifted data distributions.

The DomainNet Peng et al. (2019) dataset is one of the largest and most challenging benchmarks for domain-shift and domain-adaptation research. It spans six highly diverse domains—**Clipart, Infograph, Painting, Quickdraw, Real, and Sketch**—covering over 300 object categories. The dataset exhibits significant variations in drawing style, abstraction level, texture, and visual complexity, making cross-domain generalization particularly difficult. Due to its large scale and strong inter-domain discrepancies, DomainNet provides a rigorous testbed for evaluating the robustness, scalability, and adaptability of models under substantial distribution shifts.

The analysis on the PACS dataset using the CLIP-ViT B/16 backbone, comparing O-TPT Sharifdeen et al. (2025) with UC-TPT (ours), is presented in Fig. 12. We observe that UC-TPT consistently outperforms O-TPT across all domain-shift variants of PACS. Similarly, the results on the DomainNet dataset are shown in Fig. 13. Here as well, UC-TPT achieves better performance than O-TPT across all six domain-shift variants, demonstrating its robustness under diverse and challenging distribution shifts.

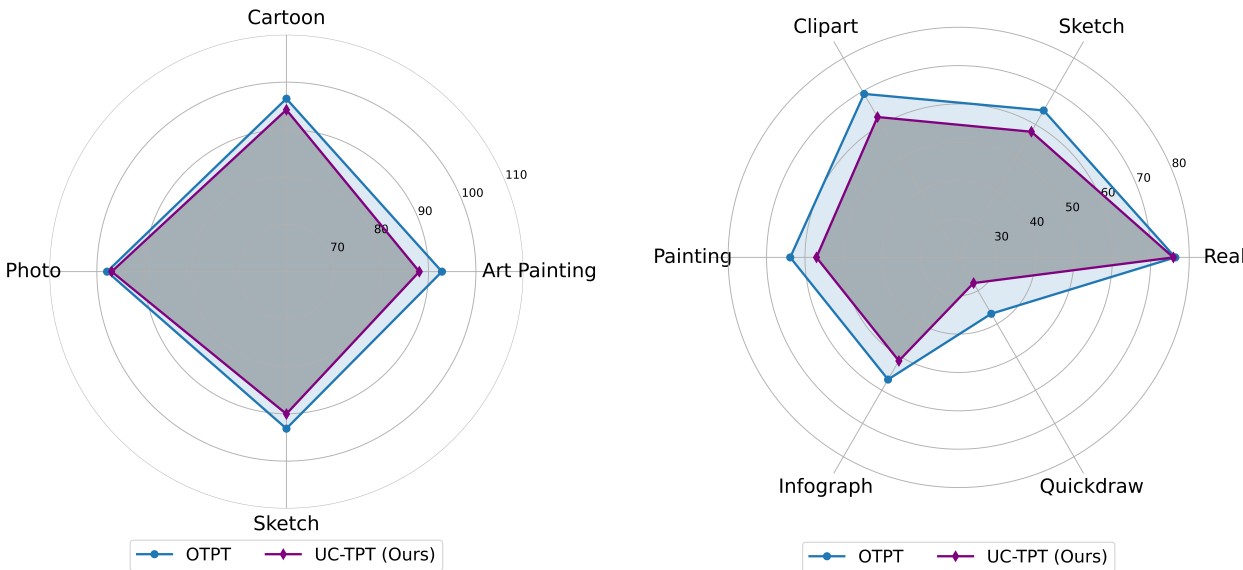

**Figure 12: Comparison of ECE (↓) on** 4 **Domain shift datasets of PACS** using CLIP-ViT B/16 as the backbone.

**Figure 13: Comparison of ECE (↓) on** 6 **Domain shift datasets of DomainNet** using CLIP-ViT B/16 as the backbone.

## N    TPT with prompt learning backbone

The analysis of test-time prompt tuning methods—O-TPT Sharifdeen et al. (2025), A-TPT Ahamed et al. (2026), and UC-TPT (ours)—combined with prompt-learning backbones such as CoOp Zhou et al. (2022) and MaPLe Khattak et al. (2023) is presented in Table 15. This evaluation is conducted on DTD, Flower, Food, Aircraft, Pets, Caltech101, UCF, EuroSAT, and Cars datasets using the CLIP-ViT B/16. From the table, we observe that UC-TPT achieves the best performance within the CoOp framework, as reflected by its lower average ECE.

For the MaPLe backbone, UC-TPT attains average ECE performance comparable to O-TPT and clearly outperforms A-TPT, while maintaining competitive accuracy across datasets. This behavior highlights an important insight: methods like MaPLe, which integrate multi-layer text–visual fusion, tend to distort the natural CLIP embedding space, negatively affecting calibration. We observed a similar effect when replacing

the shallow visual conditioning in our method with deeper visual fusion—the ECE values increased, indicating that excessive fusion alters the inherent CLIP semantic structure and consequently degrades calibration performance.

**Table 15:** Comparison of MaPLe and CoOp based prompt learning backbones combined with different TPT variants across datasets.

| Method | Metric | DTD | FLW | Food | Air. | Pets | C101 | UCF | SAT | Cars | Avg. |
|---|---|---|---|---|---|---|---|---|---|---|---|
| **MaPLe + A-TPT** | Acc | 42.38 | 66.63 | 83.33 | 22.65 | 86.07 | 93.67 | 65.29 | 48.22 | 63.28 | 63.50 |
| | ECE | 11.39 | 3.88 | 1.84 | 6.88 | 3.48 | 3.56 | 3.99 | 2.73 | 2.87 | 4.51 |
| **MaPLe + O-TPT** | Acc | 42.44 | 67.07 | 83.38 | 22.74 | 86.24 | 93.51 | 64.98 | 48.63 | 62.87 | 63.54 |
| | ECE | 11.77 | 3.23 | 2.06 | 6.10 | 3.58 | 3.50 | 3.26 | 2.45 | 3.07 | 4.34 |
| **MaPLe + UC-TPT** | Acc | 42.43 | 67.03 | 83.36 | 22.71 | 86.26 | 93.55 | 64.68 | 49.47 | 62.75 | 63.58 |
| | ECE | 10.31 | 3.40 | 2.11 | 6.04 | 3.59 | 3.67 | 3.17 | 3.81 | 3.05 | 4.35 |
| **CoOp + A-TPT** | Acc | 45.69 | 68.33 | 83.56 | 18.90 | 89.04 | 93.23 | 65.93 | 40.51 | 62.78 | 63.11 |
| | ECE | 16.10 | 9.03 | 3.94 | 20.12 | 1.57 | 1.15 | 10.84 | 12.83 | 2.55 | 8.68 |
| **CoOp + O-TPT** | Acc | 45.10 | 68.37 | 83.55 | 18.66 | 89.02 | 93.83 | 65.66 | 40.41 | 62.53 | 63.01 |
| | ECE | 16.41 | 6.96 | 3.56 | 16.87 | 2.10 | 0.97 | 9.11 | 13.71 | 2.78 | 8.05 |
| **CoOp + UC-TPT** | Acc | 44.86 | 68.58 | 83.47 | 17.70 | 88.96 | 93.63 | 64.63 | 40.56 | 62.18 | 62.73 |
| | ECE | 10.96 | 5.60 | 3.16 | 13.53 | 2.22 | 1.23 | 8.26 | 10.95 | 2.71 | 6.51 |

## O    Reliability analysis

Further reliability analysis on selected datasets—Pets, Aircraft, and DTD—using A-TPT Ahamed et al. (2026), O-TPT Sharifdeen et al. (2025), and UC-TPT (ours) with the CLIP-ViT B/16 backbone is presented in Fig. 14. In Fig. 14(a), we show the results on the Pets dataset. Both A-TPT and O-TPT exhibit strong overconfidence in the low-confidence regions, whereas UC-TPT demonstrates noticeably reduced overconfidence.

Fig. 14(b) illustrates the analysis on the Aircraft dataset. A-TPT remains largely overconfident across the confidence spectrum, while O-TPT performs slightly better but becomes underconfident at higher confidence levels. In contrast, UC-TPT shows significantly reduced underconfidence and overconfidence compared to both baselines.

Finally, Fig. 14(c) presents the results for the DTD dataset. Here, both A-TPT and O-TPT display strong overconfidence, whereas UC-TPT offers a more balanced reliability curve with substantially lower miscalibration.

For the analysis of incorrect confidences, we compared UC-TPT (ours) against A-TPT Ahamed et al. (2026) and O-TPT Sharifdeen et al. (2025) using the CLIP-ViT B/16 backbone on the Aircraft and Caltech datasets. These results are presented in Fig. 15.

In Fig. 15(a), we show the results for the Aircraft dataset. As illustrated, UC-TPT exhibits a higher density of incorrect samples in the low-confidence region and a much lower density in the high-confidence region. This behavior is desirable, as it indicates that the model assigns low confidence to its mistakes. In contrast, both A-TPT and O-TPT display the opposite trend: they produce fewer incorrect predictions in the low-confidence region but noticeably more incorrect predictions in the high-confidence region. Such patterns reflect overconfident errors, making those methods less reliable than UC-TPT.

Fig. 15(b) presents the analysis on the Caltech dataset. Even under this setting, UC-TPT shows a more favorable trend, with incorrect sample density peaking in the mid-confidence region and remaining lower in the high-confidence region compared to A-TPT and O-TPT. This again demonstrates that UC-TPT is better calibrated and more trustworthy, as it avoids producing overly confident incorrect predictions.

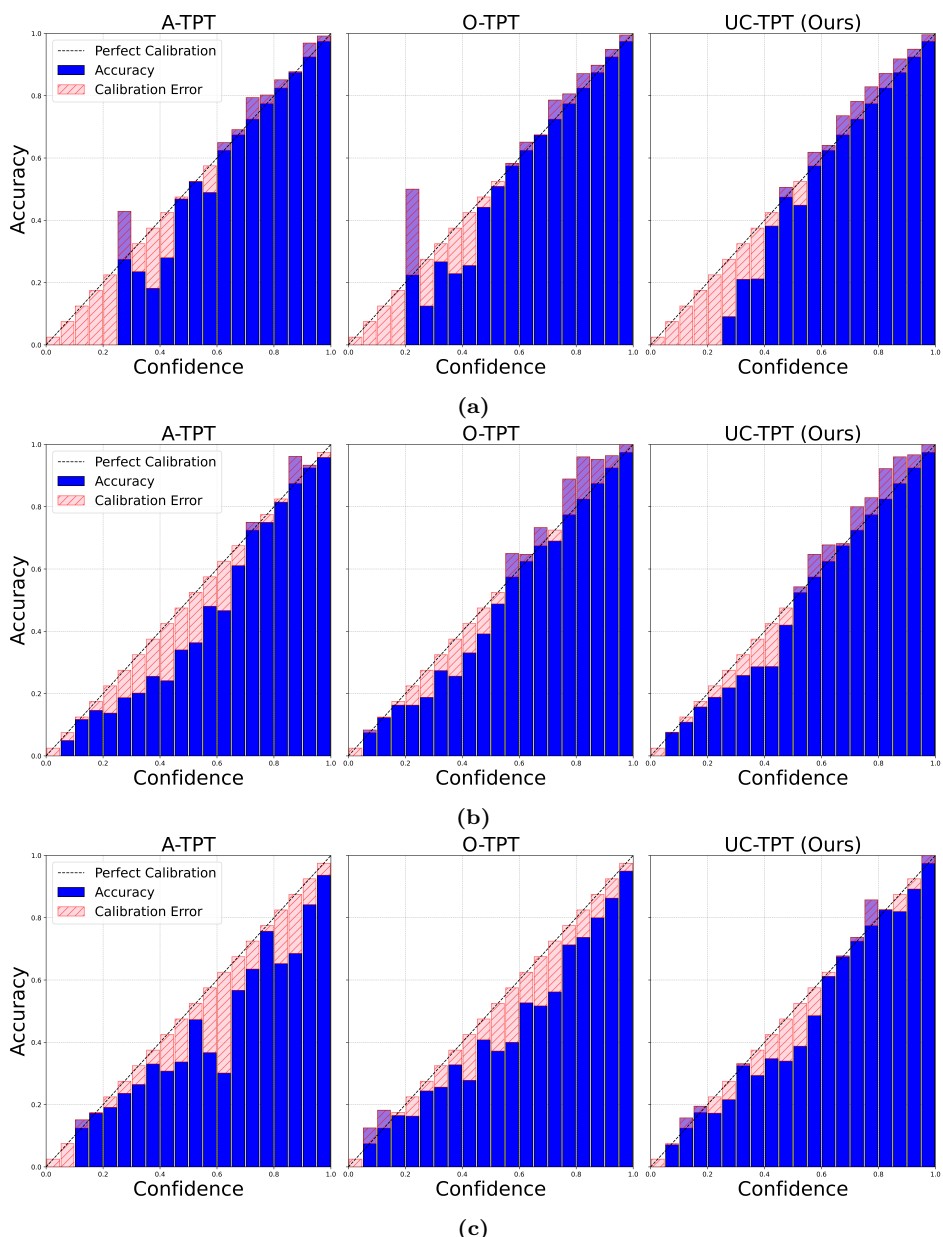

**Figure 14:** Reliability plots with CLIP-ViT B/16 backbone. (a) Pets. (b) Aircraft. (c) DTD.

## P   Analysis on combination of regularizers

In this section, we investigate how our UC-TPT behaves when combined with the regularizers used in other test-time prompt tuning methods, namely A-TPT Ahamed et al. (2026), O-TPT Sharifdeen et al. (2025), and C-TPT Yoon et al. (2024). This analysis is conducted using the CLIP-ViT B/16 backbone across six datasets: Pets, DTD, Flower, Aircraft, Caltech, and UCF. We report the average performance across these datasets to understand the overall effect of combining different regularizers. The complete results are presented in Table 16.

From the table, we observe that UC-TPT alone consistently achieves the lowest average ECE, while still maintaining competitive accuracy compared to all other combinations. This highlights the importance

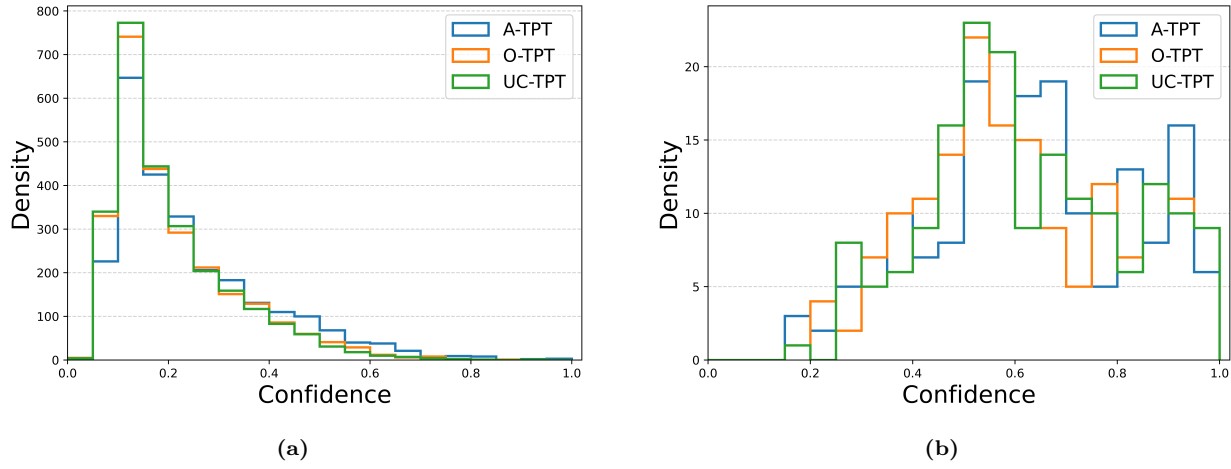

**Figure 15:** Incorrect confidence plots with CLIP-ViT B/16 backbone. (a) Aircraft. (b) Caltech.

**Table 16:** Performance comparison when regularizers from other methods are combined with UC-TPT (ours) across multiple datasets using CLIP-ViT-B/16 as the backbone. The overall best results are highlighted in **bold**.

| Method | Metric | Pets | DTD | FLW | Air. | C101 | UCF | Avg |
|---|---|---|---|---|---|---|---|---|
| Ours + O-TPT | **Acc.** | 87.94 | 43.79 | 68.55 | 23.10 | 92.49 | 63.97 | 63.31 |
| | **ECE** | 2.34 | 4.60 | 3.37 | 3.51 | 4.76 | 2.40 | 3.50 |
| Ours + A-TPT | **Acc.** | 87.62 | 43.61 | 67.21 | 23.64 | 92.25 | 63.20 | 62.92 |
| | **ECE** | 2.29 | 5.59 | 4.78 | 5.23 | 5.51 | 2.33 | 4.29 |
| Ours + C-TPT | **Acc.** | 87.84 | 44.08 | 67.83 | 23.25 | 92.53 | 63.50 | 63.17 |
| | **ECE** | 2.46 | 5.01 | 3.77 | 3.67 | 4.87 | 2.19 | 3.66 |
| **UC-TPT (Ours)** | **Acc.** | 88.40 | 44.30 | 67.80 | 24.00 | 93.40 | 63.70 | 63.60 |
| | **ECE** | 2.65 | 4.74 | 3.97 | 2.65 | 2.96 | 2.70 | **3.28** |

of preserving the semantic structure of textual embeddings in the frozen CLIP space—a central property maintained by our topology-preserving diversity regularizer.

The next best-performing combination is UC-TPT + O-TPT, followed by combinations with C-TPT and A-TPT. Although some of these combinations yield improvements on specific datasets (especially Pets and UCF), they fail to generalize across the remaining datasets. This suggests that applying aggressive transformations such as orthogonal projections, angular dispersion, or excessive separation of embeddings may distort the natural semantic relationships between closely related classes. Such distortions often lead to degraded calibration performance.

In contrast, UC-TPT alone maintains the intrinsic CLIP semantics, ensuring that class relationships remain meaningful while still enabling instance-wise uncertainty-aware prompt adaptation. This balanced behavior results in superior overall calibration performance compared to all the combined regularization approaches.

## Q   Analysis on different prompt initializations

In this section, we present a detailed analysis of different prompt initialization strategies. The experiments are conducted on the Pets, DTD, Flowers, Aircraft, Caltech, and UCF datasets using both CLIP-ViT-B/16 and RN50 backbones. Table 17 reports the results obtained with random prompt initialization. For both CLIP-ViT-B/16 and RN50, the average ECE achieved by our UC-TPT method is consistently lower compared to O-TPT Sharifdeen et al. (2025) and A-TPT Ahamed et al. (2026). This demonstrates that the uncertainty estimates used in our approach make the model more robust across different prompt initializations, allowing

it to generalize better than the competing methods. In contrast, A-TPT is highly sensitive to prompt initialization, showing a significant increase in ECE, followed by O-TPT in both backbones.

Table 18 summarizes the analysis using the prompt template "the photo of the cool $< class >$". A similar trend is observed here: our method continues to perform reliably across initialization schemes, maintaining low ECE values, while A-TPT again exhibits strong sensitivity to prompt choices, followed by O-TPT. The consistently lower ECE of UC-TPT across both initialization scenarios highlights its robustness and stability in comparison to existing approaches.

Table 17: **Comparison of accuracy and calibration performance** across datasets for the random prompt initialization. Best average performance is in **bold**.

| Method | Metric | Pets | DTD | FLW | Air. | C101 | UCF | Avg |
|---|---|---|---|---|---|---|---|---|
| ViT-B/16 O-TPT | **Acc.** | 80.32 | 34.87 | 59.72 | 22.11 | 86.97 | 55.62 | 56.60 |
| | **ECE** | 4.53 | 23.38 | 6.64 | 8.58 | 4.72 | 10.81 | 9.78 |
| ViT-B/16 A-TPT | **Acc.** | 77.10 | 34.51 | 53.59 | 21.18 | 86.53 | 50.17 | 53.85 |
| | **ECE** | 9.19 | 22.92 | 11.22 | 10.33 | 3.55 | 16.80 | 12.34 |
| ViT-B/16 UC-TPT (Ours) | **Acc.** | 82.72 | 32.68 | 61.59 | 21.54 | 85.43 | 53.89 | 56.31 |
| | **ECE** | 2.29 | 16.40 | 5.65 | 8.29 | 4.63 | 9.73 | **7.83** |
| RN 50 O-TPT | **Acc.** | 68.36 | 26.12 | 52.37 | 10.74 | 80.73 | 49.91 | 48.04 |
| | **ECE** | 10.19 | 10.67 | 7.89 | 16.34 | 6.60 | 6.21 | 9.65 |
| RN 50 A-TPT | **Acc.** | 61.29 | 26.54 | 49.74 | 10.68 | 79.11 | 49.70 | 46.18 |
| | **ECE** | 14.96 | 17.85 | 12.27 | 24.56 | 3.93 | 9.72 | 13.88 |
| RN 50 UC-TPT (Ours) | **Acc.** | 71.35 | 21.51 | 51.81 | 10.44 | 80.32 | 49.38 | 47.47 |
| | **ECE** | 5.53 | 6.87 | 6.91 | 15.34 | 7.67 | 5.53 | **7.98** |

Table 18: **Comparison of accuracy and calibration performance** across datasets with the prompt "the photo of the cool $< class >$". Best average performance is in **bold**.

| Method | Metric | Pets | DTD | FLW | Air. | C101 | UCF | Avg |
|---|---|---|---|---|---|---|---|---|
| ViT-B/16 O-TPT | **Acc.** | 88.55 | 47.17 | 70.36 | 23.88 | 91.44 | 65.71 | 64.51 |
| | **ECE** | 2.31 | 4.05 | 6.22 | 8.17 | 11.71 | 5.66 | 6.35 |
| ViT-B/16 A-TPT | **Acc.** | 85.85 | 46.57 | 63.52 | 23.55 | 91.12 | 66.08 | 62.78 |
| | **ECE** | 2.71 | 14.75 | 10.25 | 14.52 | 2.28 | 8.03 | 8.76 |
| ViT-B/16 UC-TPT (ours) | **Acc.** | 88.12 | 45.98 | 68.62 | 22.95 | 92.25 | 64.63 | 63.76 |
| | **ECE** | 2.63 | 8.71 | 4.88 | 8.12 | 2.21 | 4.45 | **5.17** |
| RN 50 O-TPT | **Acc.** | 82.23 | 39.18 | 67.15 | 16.74 | 86.69 | 59.55 | 58.58 |
| | **ECE** | 2.29 | 14.73 | 3.61 | 10.84 | 2.21 | 4.91 | 6.43 |
| RN 50 A-TPT | **Acc.** | 83.56 | 39.83 | 62.08 | 16.41 | 85.06 | 59.42 | 57.82 |
| | **ECE** | 2.10 | 10.48 | 6.33 | 11.68 | 3.06 | 6.32 | 6.66 |
| RN 50 UC-TPT (ours) | **Acc.** | 82.04 | 36.94 | 66.83 | 16.58 | 87.50 | 59.25 | 58.08 |
| | **ECE** | 2.09 | 13.15 | 3.66 | 9.32 | 2.62 | 2.95 | **5.63** |

# R    Full Results on Specialized Domains (RemoteCLIP and BioMedCLIP)

In addition to the natural, fine-grained, and biomedical domains discussed in the main text, we further evaluate the calibration performance of our method on specialized aerial and remote sensing imagery. For this analysis, we utilize the domain-specific **RemoteCLIP** Liu et al. (2024) backbone across 7 diverse remote sensing datasets: MLRSNet, PatternNet, RESISC45, AID, UCM, EuroSAT, and RSICD.

To provide a comprehensive view of the adaptation trade-offs as requested, Table 19 reports both the classification accuracy and Expected Calibration Error (ECE) across all datasets.

Our proposed **UC-TPT** achieves the lowest average ECE (**3.32%**) across the 7 datasets. It successfully suppresses the severe overconfidence degradation introduced by standard TPT (15.53%) and A-TPT (7.88%), while also slightly improving upon the highly conservative Zero-Shot baseline (3.36%). While the adaptation process introduces a slight accuracy trade-off compared to the overconfident standard TPT (43.79% vs. 45.84%), this demonstrates a crucial safety benefit of our approach: even in high-difficulty scenarios, the uncertainty-aware objective acts as a strong safeguard. It ensures the tuned model does not collapse into unwarranted overconfidence, keeping its predictive probabilities faithfully aligned with its actual capabilities.

**Table 19: Full Results on Remote Sensing Datasets (RemoteCLIP).** Classification Accuracy (%) and ECE (%) for zero-shot and various test-time prompt tuning methods.

| Method | Metric | MLRSNet | PatternNet | RESISC45 | AID | UCM | EuroSAT | RSICD | Average |
|---|---|---|---|---|---|---|---|---|---|
| Zero-Shot | Acc ↑ | 41.81 | 43.91 | 48.63 | 48.62 | 49.80 | 33.96 | 48.05 | 44.97 |
|  | ECE ↓ | 2.37 | 3.76 | 5.10 | 1.65 | 3.95 | 3.42 | 3.27 | 3.36 |
| TPT | Acc ↑ | 42.12 | 46.22 | 51.22 | 48.72 | 50.20 | 36.10 | 46.31 | 45.84 |
|  | ECE ↓ | 16.55 | 16.88 | 13.10 | 17.51 | 17.86 | 8.68 | 18.15 | 15.53 |
| A-TPT | Acc ↑ | 41.05 | 46.00 | 49.90 | 47.61 | 49.45 | 32.64 | 46.98 | 44.80 |
|  | ECE ↓ | 11.24 | 11.59 | 15.80 | 3.74 | 5.89 | 3.74 | 3.15 | 7.88 |
| C-TPT | Acc ↑ | 41.77 | 42.26 | 49.54 | 46.54 | 49.45 | 33.26 | 44.51 | 43.90 |
|  | ECE ↓ | 2.66 | 4.82 | 2.03 | 7.60 | 3.96 | 2.86 | 9.25 | 4.74 |
| O-TPT | Acc ↑ | 41.60 | 42.31 | 48.91 | 46.14 | 49.75 | 33.37 | 43.98 | 43.72 |
|  | ECE ↓ | 1.36 | 3.70 | 0.78 | 6.26 | 3.85 | 3.14 | 8.23 | 3.90 |
| **UC-TPT (Ours)** | Acc ↑ | 40.10 | 42.90 | 47.80 | 45.84 | 48.30 | 35.20 | 46.42 | 43.79 |
|  | ECE ↓ | **1.35** | 6.34 | 2.62 | 1.69 | **2.93** | 6.12 | **2.19** | **3.32** |

## R.1    Full Results on Biomedical Datasets (BioMedCLIP)

To supplement the aggregated calibration findings presented in the main text, Table 20 details the full per-dataset accuracy and Expected Calibration Error for the 11 biomedical datasets using the specialized BioMedCLIP backbone Zhang et al. (2024). Notably, UC-TPT not only provides the most calibrated predictions (achieving the lowest average ECE of 45.08%) but also yields the highest average classification accuracy (34.43%) compared to standard TPT (33.02%) and the next best regularized baseline, A-TPT (33.59%). This demonstrates that in certain highly specialized domains, our topology-preserving and uncertainty-gated framework successfully improves both reliability and discriminative performance simultaneously.

# S    Batch-Wise Normalization, Sample Efficiency, and Adaptation Steps

**The Adaptation Batch and Normalization.**    Following the standard TPT protocol Shu et al. (2022), UC-TPT processes a single test image at a time by generating a batch of $N = 64$ augmented views. In this context, our batch-wise min-max normalization (described in Appendix F) is strictly well-defined: it computes the *relative reliability* across these 64 variations of the same underlying instance. This allows the gating function to dynamically guide selective sharpening, suppressing updates for highly distorted or uninformative views while aggressively adapting on reliable ones.

**Table 20: Full Results on Biomedical Datasets (BioMedCLIP).** Classification Accuracy (%) and ECE (%) for various test-time prompt tuning methods. UC-TPT achieves both the highest average accuracy and the lowest average ECE.

| Method | Metric | BTMRI | BUSI | CHMNIST | COVID | CTKidney | Derma | Knee | Kvasir | LungCol | OCT | RETINA | Avg |
|---|---|---|---|---|---|---|---|---|---|---|---|---|---|
| TPT | Acc ↑ | 14.79 | 45.34 | 34.31 | 30.47 | 45.99 | 18.30 | 19.20 | 54.33 | 55.64 | 27.30 | 17.59 | 33.02 |
| | ECE ↓ | 81.88 | 52.85 | 57.83 | 66.55 | 46.17 | 65.98 | 73.25 | 39.59 | 33.82 | 69.41 | 66.65 | 59.45 |
| C-TPT | Acc ↑ | 15.02 | 46.18 | 32.78 | 30.45 | 46.25 | 18.90 | 19.44 | 54.08 | 55.61 | 29.40 | 18.45 | 33.32 |
| | ECE ↓ | 80.71 | 51.20 | 58.69 | 66.48 | 45.94 | 64.97 | 71.08 | 39.63 | 33.74 | 64.91 | 65.36 | 58.43 |
| O-TPT | Acc ↑ | 14.97 | 46.61 | 32.11 | 30.47 | 46.20 | 19.15 | 20.23 | 54.00 | 55.57 | 29.90 | 19.00 | 33.47 |
| | ECE ↓ | 80.59 | 49.69 | 59.07 | 66.43 | 45.95 | 64.08 | 70.30 | 39.37 | 33.67 | 64.24 | 64.24 | 57.97 |
| A-TPT | Acc ↑ | 14.61 | **49.57** | 30.64 | **30.85** | 32.10 | 36.05 | **26.63** | 53.33 | 46.25 | 28.60 | 20.82 | 33.59 |
| | ECE ↓ | 76.08 | 37.63 | 54.39 | 53.44 | 46.34 | 27.34 | 48.62 | 33.12 | **25.92** | 66.43 | 62.03 | 48.30 |
| **UC-TPT (Ours)** | Acc ↑ | **15.02** | 47.03 | **34.31** | 30.47 | **46.25** | **41.38** | 26.40 | **54.67** | **55.64** | **30.00** | **21.60** | **34.43** |
| | ECE ↓ | **74.91** | **34.19** | **44.81** | **52.46** | **38.13** | **22.55** | **45.25** | **31.70** | 29.29 | **63.01** | **59.59** | **45.08** |

**Sample Efficiency and Batch-Size Generalizability.** To evaluate the sample efficiency of UC-TPT, we ablate the number of augmented views per test instance. Table 21 reports the average performance across six representative datasets (DTD, Pets, Caltech, UCF, Flowers, Aircraft) using the ViT-B/16 backbone.

**Table 21: Ablation on the number of augmented views** ($N$). Average Accuracy and ECE across six datasets (DTD, Pets, Caltech, UCF, Flowers, Aircraft) with the ViT-B/16 backbone.

| Number of Views ($N$) | Avg. Acc ($\uparrow$) | Avg. ECE ($\downarrow$) |
|---|---|---|
| 4 | 62.95 | 4.91 |
| 16 | 63.06 | 3.79 |
| 32 | 63.18 | 3.58 |
| **64 (Default)** | **63.60** | **3.28** |

A known limitation of the general TPT paradigm is that decreasing the number of augmented views starves the model of the diverse distribution necessary to compute stable entropy gradients. To confirm that our normalization strategy does not introduce unique fragility, we compared UC-TPT against O-TPT under extreme low-view conditions. When reduced to 4 views, O-TPT degrades from 4.13 to 5.05 ECE. While UC-TPT similarly converges toward this noisy gradient floor at 4 views (4.91 ECE), it remains strictly superior to the baseline. This confirms that the degradation stems from the inherently noisy low-view estimates of test-time prompt tuning, not from our uncertainty-calibrated framework.

**Strict Batch Size 1.** In a strict batch-size 1 regime where augmentations are entirely removed, batch-wise normalization becomes mathematically undefined. To adapt our method to this setting, we substitute the batch-wise min-max scaling with running statistics normalization (tracking a moving average of the minimum and maximum uncertainty scores observed during test time). Under this augmentation-free protocol, UC-TPT achieves 62.10% Accuracy and 5.12 ECE, demonstrating that the method can still function safely, though it sacrifices the performance gains unlocked by standard TPT augmentations. Furthermore, we emphasize that we operate under a strictly globally-locked hyperparameter protocol; the exact parameters identified during our initial few-shot development phase (Appendix K) remain entirely fixed across all datasets and batch-size variations, ensuring a completely label-free test-time adaptation process.

**Test-Time Adaptation (TTA) Steps.** We also analyze the sensitivity of UC-TPT to the number of gradient steps taken during adaptation. Table 22 details the performance on ImageNet-V2 using the ResNet-50 (RN50) backbone as the number of adaptation steps is varied.

As shown in Table 22, increasing the number of adaptation steps beyond a single update strictly degrades calibration. While a second step yields a negligible accuracy bump (52.81% to 52.93%), the Expected Calibration Error (ECE) steadily worsens from 3.19 to 6.28 by the fifth step. This empirical trend confirms that extended optimization causes the prompt to overfit to the noisy test-time entropy objective, destabilizing the well-calibrated geometry of the pretrained embedding space.

**Table 22: Ablation on the number of Test-Time Adaptation steps.** Evaluated on ImageNet-V2 with the RN50 backbone.

| TTA Steps | Accuracy ($\uparrow$) | ECE ($\downarrow$) |
|:---:|:---:|:---:|
| **1 (Default)** | 52.81 | **3.19** |
| 2 | **52.93** | 3.54 |
| 3 | 52.76 | 3.98 |
| 4 | 52.56 | 4.54 |
| 5 | 52.49 | 6.28 |

Furthermore, extending adaptation to multiple steps is highly impractical for real-world deployment. Test-time prompt tuning is specifically designed for immediate, on-the-fly adaptation; multi-step TTA linearly multiplies both computational complexity and inference latency per sample. By restricting UC-TPT to a single, uncertainty-tempered step, we ensure that the framework remains lightweight and deployment-friendly without sacrificing model reliability.

## T Comparison with Post-Hoc Temperature Scaling

To explicitly isolate the calibration benefits of our representational adaptation from simple confidence softening, we benchmarked our framework against a **Zero-Shot CLIP + Temperature Scaling (TS)** baseline, as recommended during the review process.

To maintain strict adherence to our minimal-data and globally locked hyperparameter protocol, we optimized the temperature scalar using the exact same 5-shot subset of the DTD dataset utilized for our primary hyperparameter tuning. A parameter sweep identified $T = 1.2$ as the optimal temperature scalar, which was subsequently applied uniformly across the evaluation suite.

Crucially, because Temperature Scaling acts as a global, post-hoc scalar division of the logits, it strictly preserves the original argmax ordering. Consequently, the classification accuracy of the Temperature Scaled model remains identically locked to the base Zero-Shot CLIP model. To avoid redundancy, Table 23 omits classification accuracy to focus strictly on the Expected Calibration Error (ECE) trade-offs.

As shown in Table 23, while applying the optimal temperature scalar effectively softens the baseline's overconfidence, **UC-TPT** still yields a significantly lower average ECE (3.62%) compared to the Zero-Shot + TS baseline (6.59%). This confirms that the calibration gains achieved by UC-TPT stem from genuine, instance-aware adaptation of the underlying text features and decision boundaries, rather than merely acting as a static post-hoc confidence flattener.

**Table 23: Calibration Comparison: Temperature Scaling vs. UC-TPT.** Expected Calibration Error (ECE % $\downarrow$) across 9 fine-grained datasets. Accuracy is omitted as Temperature Scaling does not alter the underlying Zero-Shot classification accuracy.

| Method | DTD | UCF101 | Aircraft | Pets | Caltech | Cars | Food | Flowers | EuroSAT | Average |
|:---|:---:|:---:|:---:|:---:|:---:|:---:|:---:|:---:|:---:|:---:|
| Zero-Shot + TS ($T = 1.2$) | **4.29** | 6.25 | **2.53** | 8.19 | 9.74 | 10.02 | 6.14 | 7.00 | **5.15** | 6.59 |
| **UC-TPT (Ours)** | 4.74 | **2.70** | 2.65 | **2.65** | **2.96** | **1.35** | **3.23** | **3.97** | 8.36 | **3.62** |

## U Qualitative Analysis: Mitigating Catastrophic Overconfidence

To explicitly demonstrate the theoretical differences between standard entropy minimization, uniform geometric regularization (O-TPT), and our proposed framework (UC-TPT), we present a qualitative failure analysis using high-ambiguity samples from the EuroSAT remote sensing dataset with ViT-B/16.

As established main paper, standard test-time geometric regularizers are topology-agnostic. When dealing with fine-grained or visually similar classes—such as satellite imagery of a "River" and a "Highway"—forcing the semantic prompts to be completely orthogonal violently distorts the local embedding space. This semantic destruction forces the model to make highly overconfident predictions on the wrong class.

Figure 16 illustrates three "supporting cases" where both baseline methods suffer from this exact catastrophic failure, while UC-TPT successfully corrects the prediction.

Crucially, UC-TPT not only corrects the prediction but also provides *calibrated* confidence. For instance, in the first case (True Label: AnnualCrop), the visual ambiguity is exceptionally high. Standard TPT and O-TPT confidently misclassify the image as a Highway (>95% confidence). By gating the entropy updates using our adaptation risk metric ($U(x)$) and preserving the semantic topology, UC-TPT correctly identifies the image while outputting a calibrated, lower confidence score (27.5%). This prevents the model from injecting unwarranted certainty into downstream decision-making pipelines.

| Input Image | TPT (Baseline) | O-TPT (Geometry) | UC-TPT (Ours) |
|:---:|:---:|:---:|:---:|
|  | ✗ Highway
Conf: 99.6% | ✗ Highway
Conf: 95.0% | ✓ **AnnualCrop**
**Conf: 27.5%** |
|  | ✗ Highway
Conf: 99.3% | ✗ Highway
Conf: 94.5% | ✓ **River**
**Conf: 65.8%** |
|  | ✗ Residential
Conf: 92.0% | ✗ Residential
Conf: 94.9% | ✓ **Industrial**
**Conf: 46.9%** |

**Figure 16: Qualitative Failure Recovery on EuroSAT.** Highly ambiguous remote sensing images where geometric regularizers (O-TPT) and standard TPT yield catastrophic, high-confidence misclassifications. UC-TPT preserves the underlying semantic topology and gates risky updates, resulting in correct predictions with properly calibrated uncertainty.

## V Empirical Validation of Adaptation Risk Proxy $U(x)$

To explicitly validate the proxy assumption of our adaptation risk metric $U(x)$, we provide a quantitative stratification analysis. The core hypothesis of our uncertainty-gated mechanism is that a high $U(x)$ score correlates strongly with actual predictive failure, allowing the model to detect and suppress destructive test-time updates.

Table 24 validates this assumption empirically on the ImageNet-A dataset. By stratifying the test samples into tertiles based on their normalized uncertainty scores ($U_{norm}$), we observe a strict monotonic relationship with classification accuracy. Samples isolated in the lowest risk tertile maintain a high accuracy of 66.50%, whereas accuracy collapses to 25.76% for samples in the highest risk tertile.

This statistical stratification confirms that $U(x)$ is a highly calibrated proxy for true predictive risk. Furthermore, it demonstrates the smooth attenuation of our sigmoid gating function: it applies maximal relative weight ($\alpha \approx 0.515$) to safe, low-risk samples while gracefully decaying to near-zero ($\alpha \approx 0.019$) to prevent catastrophic updates on the highest-risk samples.

**Table 24: Empirical Validation of Adaptation Risk** $U(x)$**.** Performance stratified by uncertainty tertiles on ImageNet-A (ViT-B/16). As the predicted risk $U_{norm}$ increases, the actual classification accuracy drops significantly. Furthermore, our sigmoid gating function ($k_0 = 10, u_0 = 0.01$) smoothly attenuates the update magnitude, applying maximal relative weight to confident samples while gracefully decaying to near-zero for the high-risk tertile.

| Uncertainty Tertile | Mean $U_{norm}$ | Mean Gate ($\alpha$) | Accuracy (%) |
|---|---|---|---|
| Low Risk (Bottom 33%) | 0.004 | 0.515 | 66.50 |
| Medium Risk (Middle 33%) | 0.193 | 0.138 | 51.20 |
| High Risk (Top 33%) | 0.402 | 0.019 | 25.76 |

# W  Comparison with Concurrent Work: Semantic Orthogonal Calibration (SoC)

To further contextualize our contributions against the most recent advancements in the field, we provide an empirical comparison with the concurrent work Semantic Orthogonal Calibration (SoC) Fillioux et al. (2026). SoC similarly targets the calibration of test-time prompt tuning by addressing the semantic proximity of classes.

**Implementation and Validation Protocol:**
At the time of writing, the official codebase for SoC is publicly unavailable. Given its structural similarities to the O-TPT and C-TPT pipelines, we meticulously re-implemented the SoC methodology within our evaluation framework. To guarantee the fidelity of our re-implementation, we benchmarked it against the exact configuration reported in the original SoC paper: utilizing the CLIP ViT-L/14 backbone on the DTD dataset.

The original authors reported an Accuracy of 54.4% and an ECE of 10.9%. Our faithful re-implementation yielded an Accuracy of 54.48% and an ECE of 10.82%. This near-perfect alignment confirms the validity of our implementation and ensures a strictly fair comparison.

**Comparative Results:**
Having validated the baseline, we evaluated SoC alongside UC-TPT across our standard evaluation suite using both the ViT-B/16 and ResNet-50 backbones. The results are detailed in Tables 25, 26, and 27.

On the 10 base datasets, UC-TPT consistently outperforms SoC in calibration error across both architectures, achieving an average ECE of 3.41% (vs. 4.03% for SoC) on ViT-B/16 and 4.76% (vs. 5.03% for SoC) on ResNet-50. On the ImageNet natural distribution shifts using a ViT-B/16 backbone, UC-TPT similarly maintains a superior calibration profile (4.63% vs. 5.00% ECE).

Interestingly, under the weaker ResNet-50 backbone on ImageNet shifts, SoC yields a slightly lower average ECE (6.48% vs. 6.88% for UC-TPT). However, this marginal calibration advantage for SoC comes at a cost to raw performance. UC-TPT achieves a higher average classification accuracy on these shifts (41.87% vs. 40.59%), highlighted by consistent improvements across the majority of the distribution shifts, most notably on the challenging ImageNet-A (+3.61% accuracy) and ImageNet-K (+0.88% accuracy). Furthermore, UC-TPT still achieves superior calibration (lower ECE) on specific severe shifts like ImageNet-R (1.85% vs 2.26%). This demonstrates that while SoC achieves tight confidence bounds on average, it does so conservatively, whereas UC-TPT successfully optimizes for competitive discriminative power while maintaining robust calibration across the majority of evaluations.

**Table 25: Comparison with SoC (ViT-B/16).** Accuracy (%) and ECE (%) across 10 base datasets. SoC results are based on our validated re-implementation, while UC-TPT results are compiled from our main evaluation suite.

| Method | Metric | INet | DTD | FLW | Food | Air. | Pets | C101 | UCF | SAT | Car | Avg |
|---|---|---|---|---|---|---|---|---|---|---|---|---|
| SoC Fillioux et al. (2026) | Acc ↑ | 67.28 | 43.74 | 68.17 | 82.42 | 23.13 | 87.87 | 93.02 | 63.71 | 40.38 | 64.62 | 63.43 |
| | ECE ↓ | 1.89 | 5.21 | 3.36 | 5.43 | 3.54 | 2.61 | 4.35 | 2.52 | 9.95 | 1.43 | 4.03 |
| **UC-TPT (Ours)** | Acc ↑ | **68.20** | **44.30** | **67.80** | **83.30** | **24.00** | **88.40** | **93.40** | **63.70** | **42.40** | **64.40** | **64.00** |
| | ECE ↓ | **1.53** | **4.74** | **3.97** | **3.23** | **2.65** | **2.65** | **2.96** | **2.70** | **8.36** | **1.35** | **3.41** |

**Table 26: Comparison with SoC (ResNet-50).** Accuracy (%) and ECE (%) across 10 base datasets.

| Method | Metric | INet | DTD | FLW | Food | Air. | Pets | C101 | UCF | SAT | Car | Avg |
|---|---|---|---|---|---|---|---|---|---|---|---|---|
| SoC Fillioux et al. (2026) | Acc ↑ | 58.73 | 41.31 | 65.57 | 74.36 | 16.98 | 82.97 | 86.57 | 58.21 | 27.00 | 55.12 | 56.68 |
| | ECE ↓ | 3.05 | 12.13 | 3.12 | 1.21 | 7.25 | 3.72 | 2.86 | 2.41 | 12.67 | 1.88 | 5.03 |
| **UC-TPT (Ours)** | Acc ↑ | **58.19** | **41.62** | **65.88** | **74.31** | **16.95** | **82.31** | **87.00** | **58.39** | **28.30** | **55.15** | **56.81** |
| | ECE ↓ | **2.07** | **11.62** | **3.05** | **1.22** | **7.23** | **3.74** | **2.76** | **2.71** | **11.33** | **1.85** | **4.76** |

**Table 27: Comparison with SoC on ImageNet Distribution Shifts.** Accuracy (%) and ECE (%) evaluated on IN-A, IN-V2, IN-R, and IN-S using both ViT-B/16 and ResNet-50 backbones.

| Backbone | Method | Metric | I-A | I-V2 | I-R | I-K | Average |
|---|---|---|---|---|---|---|---|
| **ViT-B/16** | SoC Fillioux et al. (2026) | Acc ↑ | 47.84 | 61.28 | 73.68 | 46.65 | 57.36 |
| | | ECE ↓ | 6.72 | 3.45 | 4.01 | 5.81 | 5.00 |
| | **UC-TPT (Ours)** | Acc ↑ | **47.82** | **60.89** | **73.48** | **46.46** | **57.16** |
| | | ECE ↓ | **7.11** | **3.06** | **2.08** | **6.28** | **4.63** |
| **ResNet-50** | SoC Fillioux et al. (2026) | Acc ↑ | 21.19 | 52.11 | 56.00 | 33.04 | 40.59 |
| | | ECE ↓ | 18.52 | 2.38 | 2.26 | 2.75 | 6.48 |
| | **UC-TPT (Ours)** | Acc ↑ | **24.80** | **52.81** | **55.97** | **33.92** | **41.87** |
| | | ECE ↓ | **18.25** | **3.19** | **1.85** | **4.23** | **6.88** |

## X  Accuracy-Calibration Pareto Analysis

To explicitly quantify the trade-off between the calibration gains of our proposed framework and the test-time adaptation accuracy costs, we present Accuracy-ECE Pareto frontiers across our evaluation suites.

In test-time adaptation, standard entropy minimization inherently risks overconfidence. As visualized in Figure 17, plotting Average Accuracy (where higher is better) against Expected Calibration Error (where lower is better) reveals a strict trade-off frontier.

The ideal deployment state lies in the top-left quadrant (high accuracy, minimal calibration error). Across both the ViT-B/16 and ResNet-50 backbones—and extending across both the 10 base domains and the ImageNet distribution shifts—standard TPT consistently push the accuracy boundary higher but suffer from severe miscalibration, falling far to the right of the plots (ECE > 11%).

Conversely, **UC-TPT** successfully anchors the high-reliability end of the Pareto frontier. By preserving the semantic topology and gating risky updates, UC-TPT achieves the lowest ECE in our evaluations. We explicitly acknowledge that this robust calibration profile involves a measured compromise in peak adaptation accuracy compared to standard TPT. However, the Pareto curves demonstrate that this trade-off is highly efficient; UC-TPT secures substantial calibration improvements while maintaining competitive accuracy, strictly dominating several intermediate baselines (e.g., C-TPT and O-TPT) that often fall entirely below the optimal frontier curve.

This visual analysis confirms that UC-TPT offers a balanced and reliable deployment trade-off for real-world, out-of-distribution applications where predictive certainty is as critical as raw accuracy.

## Y  Dataset-Level Analysis: Isolating Uncertainty Modeling vs. Loss Shape

In Table 5 of the main text, we demonstrated that our dynamic uncertainty gate $\alpha(\mathbf{U}_{\text{norm}})$ is the primary driver of calibration in the UC-TPT framework, while the confidence-shaped target $\phi$ serves as a secondary, synergistic refinement. To provide deeper insight into the interaction between these components—and specifically to analyze the representation collapse observed in the "Shape-Only" setting—we provide a detailed, dataset-level breakdown in Table 28.

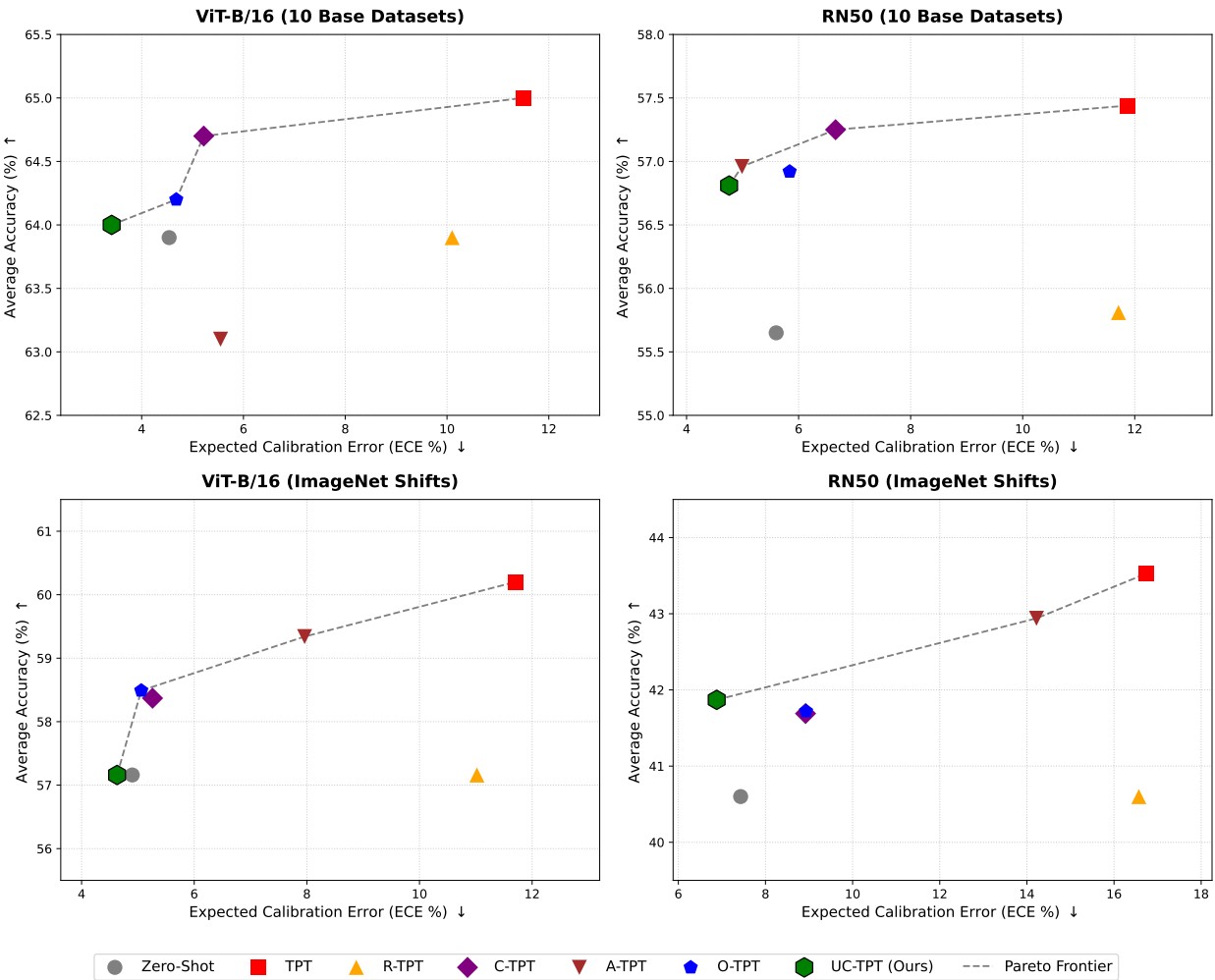

**Figure 17: Accuracy vs. Expected Calibration Error (ECE) Pareto Frontiers.** The ideal model behavior resides in the top-left quadrant (high accuracy, low ECE). Dashed lines trace the Pareto optimal frontier. Standard TPT achieves peak accuracy at the cost of severe miscalibration. UC-TPT consistently anchors the high-reliability region of the frontier across architectures and datasets, significantly reducing ECE alongside an explicitly visualized, minor trade-off in accuracy.

**Analysis of the "Shape-Only" Failure Mode:**
As observed in Table 28, replacing standard entropy minimization with the quadratic shaped loss $(\mathbf{H}(p) - \phi)^2$ *without* the uncertainty gate causes a catastrophic drop in accuracy across all six domains (e.g., Accuracy on DTD collapses from 46.00% to 23.16%).

This failure mode occurs due to the aggressive nature of uniform quadratic penalties on noisy data. Standard entropy minimization pushes prediction probabilities linearly toward one-hot extremes. In contrast, the shaped loss introduces a quadratic penalty that rigidly forces the model's entropy to match the target $\phi$. When the model encounters highly ambiguous or out-of-distribution samples, its initial predictive entropy is naturally high. Forcing a sudden, quadratic gradient update to anchor these noisy samples to $\phi$ results in violent gradient oscillations. This aggressively distorts the underlying semantic topology of the prompt embeddings, resulting in catastrophic forgetting of the pre-trained zero-shot features.

**The Synergistic Role of the Uncertainty Gate:**
The addition of the dynamic uncertainty gate $\boldsymbol{\alpha}$ perfectly resolves this instability. By effectively zeroing out the gradient magnitude for highly ambiguous samples, the gate acts as a "safety brake." This protects the

prompt topology from destructive updates while allowing the shaped loss to safely and precisely refine the boundaries of confident, well-calibrated samples. Consequently, the "Full UC-TPT" framework recovers the discriminative accuracy while driving the Average ECE down to a state-of-the-art 3.28%.

**Table 28: Dataset-Level Ablation of Entropy Objective Components.** Accuracy (%) and ECE (%) across six domains. The topology-weighted diversity regularizer ($\mathcal{L}_{\text{div}}$) remains active across all settings to ensure a strictly controlled isolation of the entropy formulation.

| Entropy Objective Formulation | Metric | DTD | Pets | Caltech | UCF | Flowers | Aircraft | Avg. |
|---|---|---|---|---|---|---|---|---|
| Base: $\mathbf{H}(p)$ | Acc ↑ | 46.00 | 88.47 | 93.83 | 67.11 | 69.38 | 24.51 | 64.88 |
| | ECE ↓ | 21.00 | 1.43 | 3.49 | 5.34 | 8.07 | 10.49 | 8.30 |
| Shape-Only: $(\mathbf{H}(p) - \phi)^2$ | Acc ↑ | 23.16 | 65.60 | 72.45 | 43.14 | 44.86 | 13.35 | 43.76 |
| | ECE ↓ | 7.05 | 6.21 | 4.44 | 6.32 | 6.48 | 6.60 | 6.18 |
| Uncertainty-Only: $\boldsymbol{\alpha} \cdot \mathbf{H}(p)$ | Acc ↑ | 43.97 | 88.08 | 93.18 | 63.52 | 67.27 | 23.10 | 63.19 |
| | ECE ↓ | 5.05 | 2.60 | 3.78 | 2.68 | 3.69 | 3.66 | 3.58 |
| Full: $\boldsymbol{\alpha} \cdot (\mathbf{H}(p) - \phi)^2$ | Acc ↑ | 44.30 | 88.40 | 93.40 | 63.70 | 67.80 | 24.00 | 63.60 |
| | ECE ↓ | **4.74** | **2.65** | **2.96** | **2.70** | **3.97** | **2.65** | **3.28** |

