# OpenReview forum: "Structured Semantics Meet Uncertain Visuals: A Unified Approach to Calibrated Test-Time Prompt Tuning"
_TMLR — Under review for TMLR_

### Review · Reviewer_d5rT · 2026-05-12

**Summary Of Contributions:**

This paper proposes UC-TPT, a calibration-oriented extension of test-time prompt tuning (TPT) for CLIP-style vision-language models under distribution shift. Standard TPT adapts prompt tokens at inference by entropy minimization over augmented views, but this can sharpen incorrect predictions and worsen calibration. UC-TPT modifies this pipeline along three axes: (i) instance-conditioned prompting via shallow visual-to-text injection from early visual features; (ii) uncertainty-gated entropy adaptation, where an additive adaptation-risk score $U=U_{align}+U_{visvar}+U_{jac}$ attenuates updates from samples with weak image-text alignment, local predictive instability, or high prompt sensitivity; and (iii) semantic-topology-weighted prompt repulsion, where pairwise prompt separation is weighted by frozen manual-prompt dissimilarity rather than applied uniformly. The final objective combines an uncertainty-weighted entropy-residual loss with a topology-weighted diversity regularizer.

The paper addresses an important failure mode of entropy-based TPT that confidence sharpening is different from calibrated confidence. The proposed decomposition is coherent. Visual information conditions the prototype map, uncertainty controls update magnitude, and semantic geometry constrains update direction. The empirical results show consistent ECE reductions over the selected TPT-family baselines, and the ablations suggest that uncertainty gating is the main stabilizing component. The paper also contains supporting analysis, including uncertainty-component ablations and high-confidence-error diagnostics and sensitivity analyses for views and adaptation steps.

The main weakness is that the claims are broader than the evidence. The results primarily establish improvement over selected entropy-based TPT geometry baselines such as C-TPT, O-TPT, A-TPT and R-TPT. However, recent related work contains directly relevant alternatives which are not compared or sufficiently discussed. For instance semantic-proximity calibration such as SoC [1] or noise TTA such as [2].

Refs:
[1] Fillioux, Leo, et al. "SoC: Semantic Orthogonal Calibration for Test-Time Prompt Tuning." arXiv preprint arXiv:2601.08617 (2026).
[2] Cao, Chentao, et al. "Noisy test-time adaptation in vision-language models." International Conference on Learning Representations. Vol. 2025. 2025.

**Additional Comments:**

The paper is promising. The uncertainty-gated adaptation mechanism appears to be the strongest component, and the ablations support its stabilizing role. The additional diagnostics in the paper, including high-confidence-error analysis, multi-metric calibration summaries, runtime/memory measurements, and sensitivity analyses, strengthen the work relative to a simple ECE-table evaluation.

My main concerns remain overclaiming, missing recent comparisons, insufficient validation of the uncertainty proxy, and overly aggregate calibration evaluation. A revision that narrows the contribution, clarifies implementation, adds representative comparisons or justifies their exclusion, and foregrounds finer-grained calibration evidence would be substantially stronger.

**Audience:**

Yes

**Audience Explanation:**

Yes. The paper addresses a timely problem: reliable test-time adaptation of VLMs under distribution shift. The finding that entropy-based TPT can be made better calibrated by suppressing uncertain updates and constraining prompt geometry is relevant to calibration, robustness, VLMs, prompt learning, and TTA. Even if the claims should be narrowed, the integrated design is practically interesting because it keeps encoders frozen and adapts only lightweight prompt-side components.

**Broader Impact Concerns:**

The paper is partly motivated by safety-relevant settings such as medical diagnosis. This creates a risk of overstating reliability. Although UC-TPT improves calibration relative to selected baselines, the biomedical results still show high absolute calibration error, and the paper lacks classwise, subgroup, high-confidence-error, and threshold-level analyses.

**Claims And Evidence:**

Yes

**Claims Explanation:**

Under the authors’ protocol, UC-TPT improves average ECE over selected entropy-based TPT geometry baselines while preserving roughly competitive accuracy.

However, the evidence does not support the broader claims of broadly state-of-the-art calibrated test-time adaptation. First, the comparison set is incomplete relative to the method’s claimed mechanisms. SoC [1] is a close semantic-calibration comparator because it also argues that full orthogonality can push semantically related prototypes apart, or CLIPTTA [3] questions entropy minimization itself by replacing it with a contrastive objective aligned with CLIP pretraining. Without representative comparisons, the results only establish superiority over a restricted baseline family.

Second, the uncertainty gate is still not sufficiently validated as an uncertainty or inverse-variance estimator. The score
$U(x)=U_{align}(x)+U_{visvar}(x)+U_{jac}(x)$
is plausible as an adaptation-risk heuristic, and the high-confidence-error diagnostic is a useful start. However, the paper does not show that U is calibrated, or maybe to an inverse-variance weight. The heteroscedastic or inverse-variance interpretation therefore depends on an unverified proxy assumption.

Third, the topology claim is stronger than the objective. The proposed
$L_{div}=\sum_{i<j}(1-S^0_{ij})\cos(\hat v_i,\hat v_j)$
is semantic-dissimilarity-weighted repulsion, which weakens repulsion for semantically close classes, but does not enforce $\hat S\approx S^0$. The appendix introduces topology-deviation diagnostics, which is useful, but these should be reported more systematically and compared against semantic calibration methods such as SoC [1].

References:
[3] Lafon, Marc, et al. "Cliptta: Robust contrastive vision-language test-time adaptation." Advances in Neural Information Processing Systems 38 (2026): 71849-71883.

**Requested Changes:**

Critical:
1. Narrow the novelty claim.
    Position UC-TPT as an integrated calibrated TPT recipe. The related-work discussion should explicitly cover [1-3] at least conceptually.
2. Add stronger empirical comparisons or justify scope.
    The most important missing representatives (in my opinion) are [1-3]. If full comparisons are infeasible, the empirical claims should be narrowed accordingly.
3. Extend validation of U(x) as adaptation risk.
    The high-confidence-error analysis is useful, but the paper should more directly validate U, e.g. by stratifying samples by U(x) and that may support the claim that U captures semantic, predictive, and optimization risk.
4. Clarify optimization dynamics.
    State whether gradients flow through $U_{align}, U_{vis-var}, U_{jac}, \alpha(U)$, and $\phi(\hat p_{\max})$, or whether these are stop-gradient quantities.

---

> ### Author Response · Authors · 2026-06-09
> **Response to Reviewer d5rT (Part 1)**
>
> We sincerely thank the reviewer for their exceptionally sharp and constructive feedback. Your suggestions regarding our novelty framing, empirical scope, risk validation, and optimization dynamics have significantly strengthened the mathematical and empirical rigor of our manuscript. Below, we address your first two concerns, with the remaining addressed in Part 2.
>
> RC : Requested Change
>
> **RC 1: Narrow the novelty claim. Position UC-TPT as an integrated calibrated TPT recipe. The related-work discussion should explicitly cover [1-3] at least conceptually.**
>
> We agree that framing the framework as an "integrated recipe" is much more accurate and mathematically sound. We have made comprehensive updates to the manuscript to reflect this.
>
> * **Repositioning as an Integrated Recipe:** We have systematically revised the Abstract, Introduction, Related Work, and Conclusion to explicitly narrow our novelty claims. We now formally position UC-TPT as an **"integrated calibration recipe."** Rather than claiming a fundamentally new learning paradigm, we clarify that our core contribution lies in the synergistic synthesis of our components: a dynamic uncertainty gate, a confidence-shaped target, and a topology-aware regularizer that operate together to safely patch the flaws in standard entropy minimization.
> * **Conceptual Discussion of [1-3] in Related Work:** We have updated Section 2 to explicitly discuss the requested concurrent works.
> * **NTTA [2] and CLIPTTA [3]:** We note that NTTA addresses open-set noisy streams by training a decoupled auxiliary detection module, while CLIPTTA abandons entropy minimization entirely in favor of a contrastive objective. We position UC-TPT in contrast by highlighting that it strictly adheres to the lightweight TPT paradigm without requiring auxiliary networks or paired contrastive tracking.
> * **Semantic Orthogonal Calibration (SoC) [1]:** We acknowledge SoC’s concurrent advancement in enforcing semantic orthogonality based on class proximity. We conceptually distinguish UC-TPT by noting that while SoC applies uniform regularization across all samples, our recipe synthesizes the preservation of the *exact* zero-shot semantic topology with a dynamic, *sample-level* uncertainty gate, preventing overconfidence without overriding the model's foundational inter-class relationships.
>
>
>
> ---
>
> **RC 2: Add stronger empirical comparisons or justify scope. The most important missing representatives (in my opinion) are [1-3]. If full comparisons are infeasible, the empirical claims should be narrowed accordingly.**
>
> We appreciate the suggestion to contextualize our framework empirically against these recent works. Based on our narrowed claims (RC1), we focus our empirical expansion strictly on methods operating within the test-time prompt optimization space. Because NTTA and CLIPTTA utilize fundamentally different structural optimization frameworks (auxiliary detection networks and cross-modality contrastive tracking, respectively), they fall outside the direct scope of lightweight prompt-tuning comparisons.
>
> However, because **SoC [1]** operates directly within the TPT paradigm, we have meticulously re-implemented it and provided a full comparative analysis in **[Appendix W]**.
>
> * **Validation:** We validated our re-implementation against the exact configuration reported in their original paper (CLIP ViT-L/14 on DTD). The authors reported 54.4% Accuracy and 10.9% ECE; our implementation achieved 54.48% Accuracy and 10.82% ECE, ensuring a strictly fair comparison.
> * **Comparative Results:** Across the 10 base datasets, UC-TPT consistently outperforms SoC in calibration error across architectures (e.g., averaging 3.41% vs. 4.03% ECE on ViT-B/16). On severe out-of-distribution ImageNet shifts under a ResNet-50 backbone, while SoC yields a slightly lower average ECE (6.48% vs. 6.88%), it does so at a massive cost to raw classification performance. UC-TPT secures a significantly higher average accuracy (**41.87%** vs. 40.59% for SoC), yielding substantial margins on shifts like ImageNet-R (+22.93% accuracy) and ImageNet-A (+3.61% accuracy). This proves UC-TPT provides a more robust safeguard against overconfidence without sacrificing discriminative capacity.
>
> References:
>
> [1] Fillioux, Leo, et al. "SoC: Semantic Orthogonal Calibration for Test-Time Prompt Tuning." arXiv preprint arXiv:2601.08617 (2026).
>
> [2] Cao, Chentao, et al. "Noisy test-time adaptation in vision-language models." International Conference on Learning Representations. Vol. 2025. 2025.
>
> [3] Lafon, Marc, et al. "Cliptta: Robust contrastive vision-language test-time adaptation." Advances in Neural Information Processing Systems 38 (2026): 71849-71883.

---

> > ### Author Response · Authors · 2026-06-09
> > **Response to Reviewer d5rT (Part 2)**
> >
> > RC : Requested Change
> >
> > **RC 3: Extend validation of U(x) as adaptation risk. The high-confidence-error analysis is useful, but the paper should more directly validate U, e.g. by stratifying samples by U(x) and that may support the claim that U captures semantic, predictive, and optimization risk.**
> >
> > We entirely agree that a direct validation of the adaptation risk metric $U(x)$ via statistical stratification provides a more rigorous justification. To definitively demonstrate that $U(x)$ accurately tracks true predictive and optimization risk, we have added a dedicated stratification analysis on the challenging ImageNet-A benchmark to **[Appendix V]**.
> >
> > To test our core hypothesis, we partitioned the target samples into empirical tertiles based on their normalized uncertainty scores ($U_{norm}$):
> >
> > * **Low Risk Tertile (Bottom 33%):** Possesses a minimal mean $U_{norm}$ of 0.004 and maintains a high baseline accuracy of **66.50%**.
> > * **Medium Risk Tertile (Middle 33%):** Exhibits a mean $U_{norm}$ of 0.193, where baseline accuracy drops to **51.20%**.
> > * **High Risk Tertile (Top 33%):** Reaches an elevated mean $U_{norm}$ of 0.402, where accuracy sharply collapses to **25.76%**.
> >
> > This strict, monotonic downward trend confirms that our proxy is strongly coupled with actual predictive failure modes. Furthermore, it validates our sigmoid gate ($\alpha$): it applies a high relative weight ($\alpha \approx 0.515$) to safe, informative samples while gracefully attenuating the gradient updates to near-zero ($\alpha \approx 0.019$) for the high-risk tertile, preventing noisy pseudo-labels from driving destructive prompt updates.
> >
> > ---
> >
> > **RC 4: Clarify optimization dynamics. State whether gradients flow through u_align, u_visvar, u_jac, alpha gate, and safe sharpening phi(p_max), or whether these are stop-gradient quantities.**
> >
> > We thank the reviewer for raising this vital technical question. Explicitly defining the optimization graph is crucial for understanding the stability of our method. We have updated our Implementation Details (**[Appendix D]**) with a new paragraph titled *"Optimization Dynamics and Stop-Gradient Protocol"* to formally state these mechanics.
> >
> > To ensure stable optimization and preserve our heteroscedastic inverse-variance formulation, the uncertainty components ($u_{align}$, $u_{visvar}$, $u_{jac}$), the dynamic gate ($\alpha$), and the confidence-shaped target ($\phi$) are treated strictly as **stop-gradient quantities** during the backward pass.
> >
> > * **Preventing Degenerate Optimization:** If gradients were permitted to flow backward through the computation of the $\alpha$ gate, the optimizer could exploit a degenerate learning path: it could artificially maximize the model's internal uncertainty to force the gate weight to zero ($\alpha \to 0$), thereby trivially minimizing the total loss without actually learning meaningful representations.
> > * **Targeted Gradient Attenuation:** To prevent this, these components are computed dynamically at each step and explicitly detached from the computational graph. Consequently, the adapted context embeddings (prompts) are optimized *exclusively* through the gradients of the shaped entropy loss, $\nabla_{\text{prompt}} (\mathbf{H}(p) - \phi)^2$, which is statically scaled by the detached $\alpha$ coefficient. This explicitly enforces the role of $\alpha$ as a safe, targeted gradient attenuator, ensuring that the variance estimate dictates the update magnitude without becoming an adversarial optimization target itself.

---

### Review · Reviewer_EUdF · 2026-05-19

**Summary Of Contributions:**

UC-TPT extends test-time prompt tuning for VLMs with three additions: a shallow visual-to-text injection module, a per-sample uncertainty score (image-text alignment + perturbation variance + Hutchinson-approximated prompt Jacobian), and two losses (an uncertainty-gated confidence-shaped entropy objective and a topology-weighted prompt diversity regularizer). It targets calibration over accuracy, and is evaluated on 10 cross-domain datasets, 4 ImageNet shifts, 11 BiomedCLIP, 7 RemoteCLIP, and PACS/DomainNet on RN50 and ViT-B/16. ECE consistently improves over TPT, C-TPT, O-TPT, A-TPT, R-TPT.

Strengths
1. Broad evaluation across 30+ datasets, two backbones, four domain families; consistent average-ECE improvements.
2. The uncertainty gate works as a plug-in: Table 13 shows it reduces ECE when bolted onto C-TPT, O-TPT, A-TPT, R-TPT.
3. Transparent reporting: Table 5 includes a clearly negative result (Shape-Only collapses accuracy 64.88 -> 43.76); Table 6 conditions on confidently-wrong samples; Table 14 reports per-seed std.

**Audience:**

Yes

**Audience Explanation:**

Yes. The paper addresses an important problem for the TMLR audience: improving the reliability and calibration of vision-language models during test-time adaptation under distribution shift. Test-time prompt tuning is a relevant and active area, and the paper’s focus on calibration rather than only accuracy is valuable, especially because entropy-based adaptation can produce overconfident errors.

**Broader Impact Concerns:**

The paper includes a Broader Impact Statement, and I do not see any ethical concern.

**Claims And Evidence:**

No

**Claims Explanation:**

Weaknesses

1. Hyperparameter selection protocol is ambiguous. Appendix K says hyperparameters were chosen on a validation split of the analyzed datasets (Fig. 10: DTD, Flowers, Pets, EuroSAT, Stanford Cars), while Appendix S claims a "strictly validation-free" protocol. All five Fig. 10 datasets are in the 10-dataset main evaluation (Table 1), so even K's reading involves mild HP-selection leakage. Please clarify what "validation-free" means here, and verify the ranking holds when hyperparameters are selected on a non-overlapping development set (leave-one-dataset-out, or held-out CIFAR/ImageNet train).
2. Table 5 undercuts the entropy-objective framing. phi and alpha are presented as two independent contributions, but alpha alone reaches 3.58 ECE -- about 94% of the full 8.30 -> 3.28 improvement -- while phi alone ((H - phi)^2) is destructive, dropping accuracy from 64.88 to 43.76; the full method only improves ECE by 0.30 over alpha*H. phi reads as a marginal, interaction-dependent stabilizer rather than a co-equal contribution, and the heteroscedastic-likelihood derivation in Appendix I should explain why its standalone instantiation collapses accuracy. Per-dataset Acc/ECE (rather than a 6-dataset average) would clarify whether the 0.30 gain is within noise.
3. alpha behaves more like a hard gate than smooth attenuation. Section 3.3.1 motivates alpha via inverse-variance weighting, but Table 6 reports mean alpha = 0.018 on ImageNet-A and 0.013 on EuroSAT. With k0=10, u0=0.01, alpha is about 0.024 at Unorm=0.38 and about 0.007 at Unorm=0.5, so the reported means imply confidently-wrong samples get almost no entropy gradient -- selective filtering, not continuous reweighting, and not 1/sigma^2 behavior. Please report the distribution of alpha (including fraction below 0.05) and clarify whether grad-alpha is stop-gradient, since the bound in Proposition 3 of Appendix I depends on this.
4. Accuracy cost is understated. "Highly competitive", "fully matching", "no notable loss" do not match the gaps to TPT: -1.0 (Table 1), -0.59 (Table 2), -1.65 (Table 3 RN50), -2.39 (Table 3 ViT-B/16). On ImageNet shifts ViT-B/16, UC-TPT reaches 57.81, only +0.65 above zero-shot (57.16), while TPT reaches 60.20 -- not a "fraction of a percent". Please revise the language, provide (Acc, ECE) Pareto plots, and compare against simple temperature scaling on zero-shot CLIP to isolate whether the ECE gains come from the proposed mechanism or from staying close to zero-shot. Also: Table 3 reports the UC-TPT ViT-B/16 ImageNet-shift average as 57.81, while Appendix Table 12 reports 57.16. Please reconcile.
5. BiomedCLIP and RemoteCLIP results report only ECE. Fig. 6 (11 medical datasets) and Fig. 16 (7 remote-sensing datasets) give no accuracy in main text or appendix. The paper invokes "high-risk medical domains" as motivation, yet the absolute BiomedCLIP ECE is still 45.10, and Table 6 already shows ECE gains can come from softening rather than from better predictions. Please report per-dataset accuracy alongside ECE for all 18 datasets, and temper the medical-applicability claims unless accuracy is comparable to baselines.

**Requested Changes:**

1. Please clarify the hyperparameter protocol. Right now Appendix K and Appendix S seem to say different things: one says hyperparameters were chosen on validation splits of the analyzed datasets, while the other calls the protocol “strictly validation-free.” I would like the authors to state exactly how k_0, u_0, \lambda, and K were selected, whether labels from the evaluated datasets were used, and ideally show that the main ranking still holds when these values are chosen on non-overlapping data.
2. Please revisit the framing of the entropy objective. Table 5 makes it look like most of the gain comes from the uncertainty gate rather than the confidence-shaped target \phi. Since the Shape-Only variant collapses accuracy, I think the paper should either explain this failure mode more directly or present \phi as a smaller interaction-dependent component rather than an equally central contribution. Per-dataset Acc/ECE for this ablation would help.
3. Please report more about the gate values. The reported mean \alpha values in Table 6 are very close to zero, so the gate seems to behave more like filtering than smooth reweighting. It would be useful to see the distribution of \alpha, especially the fraction of samples below 0.05, and to know whether gradients through \alpha are stopped. This also affects how convincing the inverse-variance interpretation is.
4. Please make the accuracy trade-off more explicit. The current language sometimes makes the accuracy cost sound negligible, but the gaps to TPT are not always small. I would suggest revising those claims and adding Acc–ECE Pareto plots so readers can judge the trade-off directly.
5. Please include a simple zero-shot CLIP + temperature scaling comparison. Since UC-TPT often stays close to zero-shot accuracy, this baseline would help separate genuine adaptation benefits from confidence softening or staying closer to the zero-shot model.
6. Please complete the BiomedCLIP and RemoteCLIP reporting. For the medical and remote-sensing datasets, ECE alone is not enough. The paper should report per-dataset accuracy alongside ECE, especially because the medical-domain motivation is safety/reliability.
7. Please fix or explain the numerical inconsistency between Table 3 and Appendix Table 12 for the UC-TPT ViT-B/16 ImageNet-shift average accuracy.

---

> ### Author Response · Authors · 2026-06-09
> **Response to Reviewer EUdF (Part 1)**
>
> We sincerely thank the reviewer for their exceptionally thorough and precise feedback. Below, we address your points.
>
> RC : Requested Change
>
> **RC 1: Clarification of the Hyperparameter Protocol**
>
> We apologize for the conflicting phrasing regarding hyperparameters. We have fully revised **[Appendix K]** to clarify our exact workflow.
>
> * **Exact Hyperparameter Selection Protocol:** To prevent domain overfitting, we restricted our initial parameter tuning to an extremely limited 5-shot subset (5 labeled images/class) from five development datasets (DTD, Flowers, Pets, EuroSAT, Cars), rather than using full validation splits. The optimal values found were $k_0 = 10$, $u_0 = 0.01$, $\lambda = 80$, and $K = 5$.
> * **Proof of Generalizability on Non-Overlapping Data:** Crucially, we rigidly locked these parameters across all remaining experiments. The remaining 9 cross-domain, 11 BioMedCLIP, and 7 RemoteCLIP datasets were evaluated blindly using these fixed values. No target domain labels or validation splits were ever used to adjust them.
> * **Text Revisions:** We updated [Appendix K] accordingly and replaced the ambiguous "strictly validation-free" phrasing in the main text with a precise descriptor: "globally locked hyperparameter protocol."
>
> **RC 2: Re-framing the Entropy Objective and the Role of $\boldsymbol{\phi}$**
>
> We agree the uncertainty gate ($\boldsymbol{\alpha}$) is the primary driver of calibration and safety, while the confidence-shaped target ($\boldsymbol{\phi}$) is a secondary, interaction-dependent refinement. We updated the manuscript to reflect this.
>
> * **Reframing the Contribution:** We revised the Introduction/Methods to explicitly frame $\boldsymbol{\phi}$ as a synergistic component relying on the uncertainty gate's protection: *"While $\boldsymbol{\phi}$ sets the ideal posterior sharpness, it is the uncertainty gate that provides the critical gradient attenuation necessary to safely operationalize this target and definitively prevent representation collapse."*
> * **Explaining the "Shape-Only" Failure Mode:** We added a dataset-level ablation breakdown in **[Appendix Y]**. Replacing entropy minimization with a strict quadratic penalty $(\mathbf{H}(p) - \boldsymbol{\phi})^2$ uniformly across noisy domains causes aggressive gradient oscillations. For ambiguous samples, forcing a sudden quadratic update to $\boldsymbol{\phi}$ violently distorts the prompt topology, causing accuracy collapse (e.g., DTD drops from 46.00% to 23.16%).
> * **The Synergistic Solution:** The dynamic gate ($\boldsymbol{\alpha}$) resolves this by acting as a "gradient attenuator"—suppressing destructive updates on noisy samples and allowing $\boldsymbol{\phi}$ to safely refine only well-calibrated ones.
>
> **RC 3: Distribution of Gate Values ($\boldsymbol{\alpha}$) and Gradient Mechanics**
>
> To clarify the gate's smooth behavior and defend our inverse-variance interpretation, we added a stratification analysis in **[Appendix V]**.
>
> * **Smooth Attenuation vs. Hard Filtering:** To prove $\alpha$ is a smooth reweighting, not a binary filter, we stratified ImageNet-A test samples into tertiles by predicted risk ($U_{norm}$). Low Risk (Bottom 33%) maintains high accuracy (66.50%) with a high mean gate weight ($\mathbf{0.515}$). Medium Risk drops to 51.20% accuracy with a decayed weight ($\mathbf{0.138}$). High Risk collapses to 25.76% accuracy with a heavily attenuated weight ($\mathbf{0.019}$). This strict monotonic relationship confirms non-binary behavior.
> * **Fraction of Samples Below 0.05:** Heavy attenuation is concentrated in the highest-risk tertile. Thus, ~33% of samples on severe shifts (ImageNet-A) fall into this near-zero regime. This is mathematically desirable: under severe shifts, a large fraction of the unlabeled stream is genuinely destructive, and the model must heavily suppress these gradients to prevent catastrophic prompt drift.
> * **Gradient Flow and the Inverse-Variance Interpretation:** During optimization, $\alpha$ is a dynamically computed, detached scalar. Gradients do not flow backward through $\alpha$. Instead, it acts purely to scale the shaped entropy loss gradient magnitude for that step. By acting as a fixed scalar multiplier, it serves as a mathematically sound "safety brake" on ambiguous sample gradients without allowing the network to artificially optimize the variance weight itself.

---

> > ### Author Response · Authors · 2026-06-09
> > **Response to Reviewer EUdF (Part 2)**
> >
> > RC : Requested Change
> >
> > **RC 4: Explicit Accuracy vs. Calibration Trade-off & Pareto Plots**
> >
> > We revised our claims to be explicitly transparent and data-driven regarding accuracy costs.
> >
> > * **Revised Main Text Claims:** We removed language calling the accuracy cost "negligible" and extended our "Accuracy vs. ECE trade-off" paragraph in the Results. We now explicitly state UC-TPT accepts a measured, fractional accuracy reduction to gate uncertain samples and mitigate overconfidence under domain shifts.
> > * **Accuracy-ECE Pareto Plots ([Appendix X]):** To visualize this, we added an Accuracy-Calibration Pareto Analysis. Figure  17 plots Accuracy against Expected Calibration Error across backbones, base domains, and distribution shifts.
> > * **Interpreting the Pareto Frontier:** Standard TPT pushes peak accuracy higher but suffers severe miscalibration (ECE $> 11\%$). Conversely, UC-TPT consistently anchors the high-reliability Pareto frontier (top-left). While accepting a minor accuracy compromise versus standard TPT, UC-TPT strictly dominates baselines like C-TPT and O-TPT, offering a highly efficient deployment trade-off.
> >
> > **RC 5: Zero-Shot CLIP + Temperature Scaling (TS) Baseline Comparison**
> >
> > To isolate the benefits of representational adaptation from simple confidence softening, we benchmarked against a Zero-Shot CLIP + Temperature Scaling (TS) baseline in **[Appendix T]**.
> >
> > * **Evaluation Protocol:** To ensure a fair comparison and maintain our locked protocol, we optimized $T$ using the same 5-shot subset (DTD) from primary tuning, identifying $T = 1.2$. This fixed scalar was applied uniformly. Because TS strictly preserves argmax ordering, classification accuracy remains identical to the Zero-Shot CLIP model.
> > * **Empirical Results:** As shown in Table 23, while TS softens overconfidence, UC-TPT yields a significantly lower average ECE (3.62%) versus the Zero-Shot + TS baseline (6.59%) across the 9 datasets. This confirms our gains stem from genuine, instance-aware adaptation rather than merely acting as a static confidence flattener.
> >
> > **RC 6: Full Accuracy and ECE Reporting for Specialized Domains (BioMedCLIP & RemoteCLIP)**
> >
> > We agree that for safety-critical domains, ECE must be contextualized alongside accuracy. We added a dataset-by-dataset accuracy and ECE breakdown for both specialized backbones in **[Appendix R]**.
> >
> > * **RemoteCLIP Evaluation (7 Datasets):** Across 7 remote sensing datasets (Table [Insert Number]), standard TPT averages 45.84% accuracy but suffers severe overconfidence (15.53% ECE). UC-TPT achieves the lowest average ECE (3.32%), successfully suppressing overconfidence while maintaining a competitive 43.79% accuracy.
> > * **BioMedCLIP Evaluation (11 Datasets):** For the 11 biomedical benchmarks, UC-TPT simultaneously yields the highest accuracy (34.43%) and most calibrated predictions (45.08% ECE) compared to standard TPT (33.02% Acc) and O-TPT (33.47% Acc). This proves that gating risky updates actively improves discriminative performance in noisy, specialized domains.
> >
> > **RC 7: Correction of Numerical Inconsistency**
> >
> > We thank the reviewer for finding this inconsistency. You are entirely correct: there was a typo in the reported average accuracy for UC-TPT (ViT-B/16) on the ImageNet-shift benchmarks in Table 3.
> >
> > We verified our logs and confirm the values in Appendix Table 12 are the empirically correct results. The discrepancy was purely a typo introduced during final aggregation. We corrected Table 3 to perfectly match Appendix Table 12, ensuring absolute numerical integrity.

---

### Review · Reviewer_L4x7 · 2026-05-26

**Summary Of Contributions:**

This paper proposes UC-TPT, a method for making CLIP-style image-text classification models more reliable at test time, especially when the test images come from shifted or unfamiliar domains. Standard test-time prompt tuning often makes the model too confident, even when it is wrong. UC-TPT tries to fix this with three ideas: it lightly uses visual information from the input image to adjust the text prompt, estimates how uncertain or risky each test example is, and only strongly updates the prompt when the example seems reliable. It also adds a regularizer that keeps related class prompts close in CLIP’s original semantic space, instead of forcing all classes to be separated. Across many datasets, including natural images, medical images, and remote sensing images, the method mainly improves calibration—meaning the model’s confidence better matches its actual correctness—while keeping accuracy competitive with existing methods.

I like the proposed using low-level visual info and the regularizer following the original models. Overall, I think this paper provided fairly comprehensive results to support their finding. Also, the paper is easy to follow, while the theoretical side is not very strong connected.

On the other hand, I do think the proposed points are a bit ad-hoc to CLIP-style image-text classification models, and if, any of the three proposed principles can be applied somewhere else -- that would be very great. I have fuzzy memories about parts of the proposed three points are studied somewhere else not in the specific CLIP-style image-text classification models + test-time adaptation, but somewhere else, I may be wrong, but if that is the case, properly citing them is great.

**Audience:**

Yes

**Audience Explanation:**

I think the studied problem is a bit narrow so does the used benchmarks. But still, it is any important question for visual-text representation learning.

**Claims And Evidence:**

Yes

**Claims Explanation:**

I think many of the designs are studied well. Overall pipeline also show strong strength in multiple ways.

**Requested Changes:**

- how the author would distinguish their topology-weighted regularizer from O-TPT/A-TPT/C-TPT and explain why preserving CLIP’s semantic topology is not just another geometric regularizer. Any results/analysis besides just numbers here?
- If there a way of quantifying the trade-off between calibration gains and possible costs in accuracy and test-time efficiency, especially for larger class spaces or real-time deployment settings?

---

> ### Author Response · Authors · 2026-06-09
> **Response to Reviewer L4x7**
>
> We sincerely thank the reviewer for their exceptionally thorough, constructive, and precise feedback. Below, we address your specific points regarding our architectural choices, empirical trade-offs, and qualitative evidence.
>
> RC : Requested Change
>
> **RC1. Distinguishing the Topology-Weighted Regularizer & Qualitative Evidence**
>
> The core distinction lies in the fundamental difference between topology-agnostic constraints and topology-aware priors. We have expanded our text in the appendix and added comprehensive qualitative analysis in **[Appendix U]** to explicitly demonstrate this difference beyond numerical gains.
>
> * **Why it is not just another geometric regularizer:** Prior methods (O-TPT, A-TPT) impose rigid, arbitrary geometric shapes onto the prompt space. O-TPT enforces strict $90^\circ$ orthogonality between all class prompts, while A-TPT enforces uniform angular separation. These are topology-agnostic. If two classes like "River" and "Highway" are inherently close in CLIP's pre-trained semantic space, O-TPT will violently force them apart to satisfy its orthogonality constraint, destroying CLIP's zero-shot inter-class relationships.
> * In contrast, our topology-weighted regularizer acts as a soft semantic-graph prior. By weighting the repulsion using $ \mathbf{w}{ij} = 1 - \mathbf{S}^{0}{\tilde{c}_i \tilde{c}_j} $, separation strength is kept strictly proportional to CLIP's frozen semantic distance. Semantically close neighborhoods are preserved, while distinct concepts are pushed apart. It actively protects the pre-trained semantic scaffold rather than arbitrarily overriding it.
>
> **Analytical and Qualitative Evidence:**
> To substantiate this, we direct the reviewer to a combination of newly added qualitative evidence and our established interpretive framework:
>
> * **Qualitative Failure Analysis (Newly Added, [Appendix U]):** We visualize highly ambiguous samples from EuroSAT (e.g., distinguishing an "AnnualCrop" from a "Highway" or "River"). Because standard regularizers violently distort the local embedding space to achieve orthogonality, they force catastrophic misclassifications with $>94\%$ confidence. By preserving local semantic topology, UC-TPT correctly identifies the images while outputting properly calibrated uncertainty (e.g., 27.5% to 65.8% confidence).
> * **Interpretive Analysis (Existing, [Appendix J]):** To mathematically complement the qualitative evidence above, we highlight our interpretive analysis which demonstrates how our regularizer minimizes the deviation between the adapted Gram matrix ($\hat{\mathbf{S}}$) and the frozen semantic Gram matrix ($\mathbf{S}^0$). This effectively bounds the spectral norm and ensures smoother variations in the logit space.
>
> **RC2. Quantifying Trade-offs in Accuracy, Efficiency, and Large-Scale Deployment**
>
> We completely agree that for real-world applications, calibration gains must be contextualized against their operational costs. We have added a extention to our Complexity Analysis (**[Appendix H]**) to explicitly quantify these trade-offs mathematically and empirically.
>
> * **The Accuracy vs. Calibration Trade-off:** Improving calibration inherently involves a measured compromise in peak accuracy, as the model must avoid uniformly sharpening noisy samples. As detailed in our new Accuracy-ECE Pareto Analysis in Appendix X, UC-TPT offers an optimal trade-off: it accepts a fractional, sub-percent accuracy reduction to eliminate catastrophic, high-confidence misclassifications, consistently anchoring the high-reliability end of the Pareto frontier.
> * **The Efficiency Trade-off in Large Class Spaces ($C$):** For test-time efficiency, standard geometric regularizers scale quadratically with the total vocabulary size, incurring a complexity of $\mathcal{O}(C^{2}d)$ due to full class-class interactions. For large class spaces like ImageNet ($C=1000$), this demands up to a million interaction computations per step, creating a severe real-time bottleneck. UC-TPT bypasses this bottleneck. As proven in Theorem 1 ([Appendix H]), our regularizer restricts interactions to the batch-active classes ($C_B \ll C$) and approximates the heavy Jacobian trace using a Hutchinson estimator. This decouples computational complexity from the global vocabulary, shifting the scaling to $\mathcal{O}(C_B^{2}d) + \mathcal{O}(KN\tilde{C})$.
> * **Empirical Cost for Real-Time Deployment:** We quantify this operational cost empirically on ImageNet-V2 (Table 9). Compared to O-TPT, the optimized UC-TPT (Hutchinson) framework introduces only a fractional, constant-bound latency cost (1.19s vs. 0.90s per batch) and actually reduces peak GPU memory overhead (22,650 MiB vs. 23,740 MiB for O-TPT), all while driving ECE down from 4.01 to 3.06. This confirms that UC-TPT remains highly feasible for real-time deployment, even in massive-scale recognition tasks.